# Granular porous landslide tsunami modelling – the 2014 Lake Askja flank collapse

Matthias Rauter [1,2,3,4✉], Sylvain Viroulet [5], Sigríður Sif Gylfadóttir[6], Wolfgang Fellin [1] & Finn Løvholt[3]

Subaerial landslides and volcano flank collapses can generate tsunamis with devastating consequences. The lack of comprehensive models incorporating both the landslide and the wave mechanics represents a gap in providing consistent predictions of real events. Here, we present a novel three-dimensional granular landslide and tsunami model and apply it to the 2014 Lake Askja landslide tsunami. For the first time, we consistently simulate small-scale laboratory experiments as well as full scale catastrophic events with the same model. The model captures the complete event chain from the landslide dynamics to the wave generation and inundation. Unique and complete field data, along with the limited geographic extent of Lake Askja enabled a rigorous validation. The model gives deep insights into the physical landslide processes and improves our understanding and prediction capabilities of frequent and catastrophic landslide tsunamis.

[1] Unit of Geotechnical Engineering, University of Innsbruck, Innsbruck, Austria. [2] Department of Water Management, Office of the Tyrolean Regional Government, Innsbruck, Austria. [3] Norwegian Geotechnical Institute, Oslo, Norway. [4] Faculty of Mathematics and Natural Sciences, University of Oslo, Oslo, Norway. [5] Institut de Mécanique des Fluides de Toulouse (IMFT), Université de Toulouse, CNRS, Toulouse, France. [6] Icelandic Meteorological Office, Reykjavík, Iceland. ✉email: matthias@rauter.it

Tsunamis induced by giant landslides impacting water bodies can be catastrophic[1,2], as recently manifested by the 2018 Anak Krakatau volcanic sector collapse that caused 430 fatalities in the Sunda Strait[3–5]. More than six other events during the last 15 years with tsunami run-up as high as 150 m[6–10] show that such powerful tsunamis happen relatively frequently. Further, a variety of other fatal events from volcanic sector collapses (1792 Mount Unzen tsunami[11]) to subaerial land- (e.g. 1963 Vajont landslide[12]) and rockslides (e.g. three Norwegian events from 1904–1936[2]) underpin their destructive potential. Paleotsunami evidence from ancient volcano sector collapses shows that these events may scale up to an even larger magnitude[13,14] and that they might have oceanic reach[15].

Despite recent progress in modelling coupled landslide and tsunami dynamics[16–22], realistic landslide tsunami modelling is still a major hurdle in understanding and predicting future events. Fundamental properties of landslide tsunamis have been studied extensively in the laboratory[23], but this is yet to be matched with predictive mathematical models[16,17]. Simplifications of either landslide dynamics (e.g. simplified rheologies[19,24]) or wave generation (e.g. depth-integration[16,25]) have prevented real progress in this field for more than a decade and it has become clear that a paradigm shift is necessary[26]. A key issue has been the inability to explain both the landslide dynamics and the tsunami at various scales with a unified model and parametrisation. Without validated continuum-mechanical models, a deeper understanding of the process is hardly possible and predictions cannot be scaled up to forecast real events.

In this work, we provide a significant leap forward, and demonstrate, for the first time, a model that is equally applicable to laboratory and real case events. The novel multiphase granular flow and tsunami model is based on the $\mu(I),\phi(I)$-rheology, complemented by critical state theory and pore fluid flow[27]. This is a logical extension of the classic $\mu(I)$-rheology for dry granular material proposed by Jop et al.[28] and allows mixing of the granular material with fluids that represent the tsunami. Accuracy and scalability are demonstrated through laboratory data[23], but more importantly, through a unique dataset obtained from the 2014 Lake Askja (Iceland) landslide and tsunami (see Fig. 1), covering in detail both the landslide footprint and the tsunami run-up across the entire periphery of the lake[6]. The limited geographical extent of Lake Askja renders it reminiscent of a large-scale laboratory, suitable for testing models at a full-scale and appropriate resolution. The insight obtained from this event shows promise on how predictive granular multiphase models with very few assumptions are within reach for even the largest events (e.g. Anak Krakatau, Mount Unzen). We use the multiphase Navier–Stokes equations to simulate the porous landslide and the pore fluid (see the Methods section for more details). The pore fluid is composed of air and water and extends beyond the slide to represent the lake and the tsunami. The unknown variables of the model are the granular velocity $\mathbf{u}_g$, the combined velocity of water and air $\mathbf{u}_c$, and the volumetric phase fractions of granules $\phi_g$, water $\phi_w$ and air $\phi_a$. The phase fractions range from 0 to 100% and describe the local mixture of constituents as shown in Fig. 2. The granular rheology is described in terms of the $\mu(I),\phi(I)$-rheology[27] and the permeability is described by a drag model. While this model is computationally expensive, it is within reach even for full three-dimensional simulations of real cases and a good trade-off between simplicity and accuracy.

## Results and discussion

**Laboratory experiment and parameters.** The porous multiphase model is validated with tank experiments of Viroulet et al.[23] (setup shown in Fig. 4). We neglect variations across the tank width and approximate the experimental setup with a two-dimensional, width-averaged numerical simulation (see supplementary materials for validation of the approach). The landslide is represented by 2 kg of dry granules on a rough slope with an inclination of 45°, right above the water reservoir with a depth of 0.15 m. All constitutive parameters are determined a priori to the experiment: The granular material has a grain diameter $d$ of 0.004 m, a density $\rho_g$ of 2500 kg m$^{-3}$ and a quasi-static friction angle of 23.3°. The $\mu(I)$-parameters are chosen accordingly ($\mu_s = \sin(23°) = 0.39$[29]) and supplemented with data from the literature ($\mu_d \approx \mu_s + 0.25$, $I_0 \approx 0.3$[30]). The critical state parameters, determining the porosity of the granular material, $\phi_{rlp}$, $\phi_{rcp}$, $a$ and $\Delta\phi$ are the same as in a previous work[27]. Material parameters of water ($\rho_w$, $\nu_w$) and air ($\rho_a$, $\nu_a$) are well known. The complete set of parameters is listed in Table 1.

**Laboratory-scale model results.** A time series of the simulated granular slide and its impact into the water reservoir is shown in Fig. 3, along with pictures of the experiment. The numerical simulation is visualised in terms of fractions of air, water and granules (the full multidimensional colour map is shown in Fig. 2). Furthermore, the interface between air and water, defined by $\phi_w/(\phi_a + \phi_w) = 0.5$ is shown as a black line. The slide starts as a triangular block of dry granular material (Fig. 3f), hits the water surface immediately after its release and generates a tsunami that travels away from the slope (Fig. 3g). Water penetrates the porous landslide and saturates the granular material shortly after it is fully submerged (Fig. 3h). Some enclosed air bubbles leave the granular slide, break up and rise to the surface (Fig. 3i). The same phenomenon can be observed in the experiments (Fig. 3c, d), although the position and timing of bubbles differ. The slide

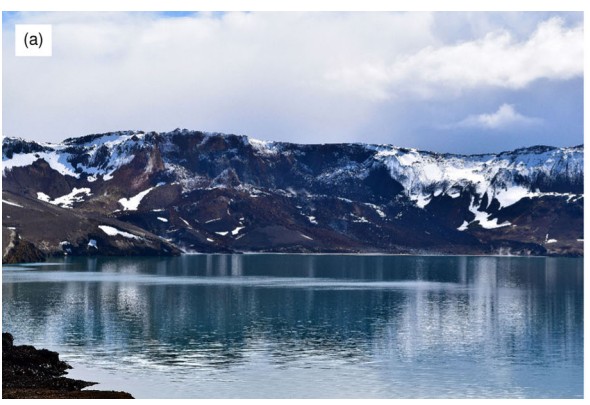
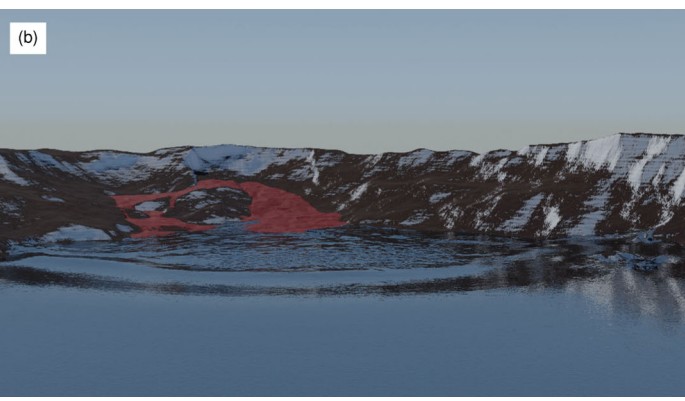

**Fig. 1 Lake Askja landslide tsunami. a** Photo of Lake Askja including the landslide scar and deposition from the 2014 landslide event (Picture: Kristinn I. Pétursson). **b** Rendered simulation of this event, showing the wave in the middle of the lake and the displaced landslide mass, highlighted in red.

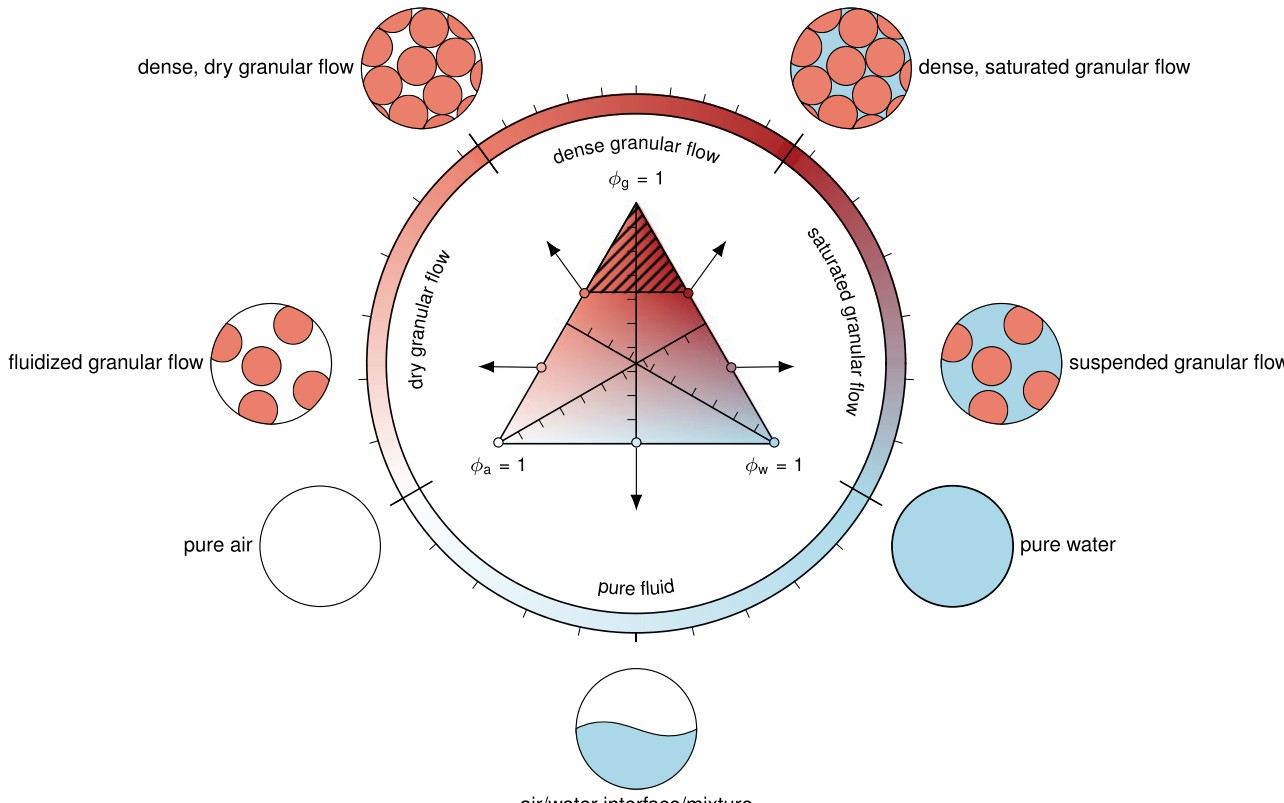

**Fig. 2 Phases and the corresponding flow regimes.** The multiphase Navier–Stokes equations allow the representation of various fluids and flow regimes by combining the phase fractions as shown in the diagram. Phases differ in densities and stress models and can thus represent the behaviour of the various flow regimes. The striped area is excluded, accounting for the pore space that is always present in granular material. The coloured ring represents a reduced colour map of limiting cases. The relevant part of this ring is used as colour map in this paper to visualise the local phase fractions.

**Table 1 Model parameters for the small-scale experiment of Viroulet et al.[23] and the Lake Askja case[6].**

| Phase/Component | parameter | description | value (experiment) | value (Askja) |
|---|---|---|---|---|
| air | $\rho_a$ | air density | $1\,\mathrm{kg\,m^{-3}}$ | $1\,\mathrm{kg\,m^{-3}}$ |
| | $\nu_a$ | air viscosity | $1.48 \cdot 10^{-5}\,\mathrm{m^2\,s^{-1}}$ | $1.48 \cdot 10^{-5}\,\mathrm{m^2\,s^{-1}}$ |
| water | $\rho_w$ | water density | $1000\,\mathrm{kg\,m^{-3}}$ | $1000\,\mathrm{kg\,m^{-3}}$ |
| | $\nu_w$ | water viscosity | $1 \cdot 10^{-6}\,\mathrm{m^2\,s^{-1}}$ | $1 \cdot 10^{-6}\,\mathrm{m^2\,s^{-1}}$ |
| grains | $d$ | particle diameter | $0.004\,\mathrm{m}$ | $0.01\,\mathrm{m}$ |
| | $\mu_s$ | quasi-static friction coefficient | 0.39 | 0.17 |
| | $\mu_d$ | dynamic friction coefficient | 0.64 | 0.37 |
| | $l_0$ | reference inertial number | 0.30 | 0.30 |
| | $\nu_{min}$ | lower viscosity threshold | $10^{-5}\,\mathrm{m^2\,s^{-1}}$ | $10^{-4}\,\mathrm{m^2\,s^{-1}}$ |
| | $\nu_{max}$ | upper viscosity threshold | $10^{0}\,\mathrm{m^2\,s^{-1}}$ | $10^{4}\,\mathrm{m^2\,s^{-1}}$ |
| | $\bar{\phi}$ | reference packing density | 0.60 | 0.60 |
| | $\rho_g$ | particle density | $2500\,\mathrm{kg\,m^{-3}}$ | $2500\,\mathrm{kg\,m^{-3}}$ |
| | $\phi_{rlp}$ | random loose packing density | 0.53 | 0.53 |
| | $\phi_{rcp}$ | random close packing density | 0.63 | 0.63 |
| | $a$ | critical state line parameter | $130\,\mathrm{Pa}$ | $50{,}000\,\mathrm{Pa}$ |
| | $\Delta\phi$ | dynamic loosening factor | 0.05 | 0.05 |

reaches its final run-out after ~1.5 s (Fig. 3j), similar as in the experiment (Fig. 3e), and forms a stable pile. Virtual wave gauges recorded the water surface at selected positions over time. The results are shown and compared with the experimental wave signals in Fig. 4.

**Laboratory-scale model performance.** Comparing pictures of the experiment (Fig. 3a–e) with snapshots of the numerical simulation (Fig. 3f–j) shows that the model captures the macroscopic dynamics of the event, from the landslide shape, its final

deposition to the generated wave. The model is able to produce the decisive behaviour of granular flows under dry and submerged conditions (as shown in separate simulations by Rauter[27]) without any fitting of parameters. Further, we notice many details, such as the propagation of the waterfront into the porous landslide, the escape of air in bubbles and the formation of a vortex above the slide. Capturing these details is only possible due to the granular-porous multiphase formulation. Notably, the same parameters achieve consistently good results in all cases (see supplementary materials for additional cases, varying the

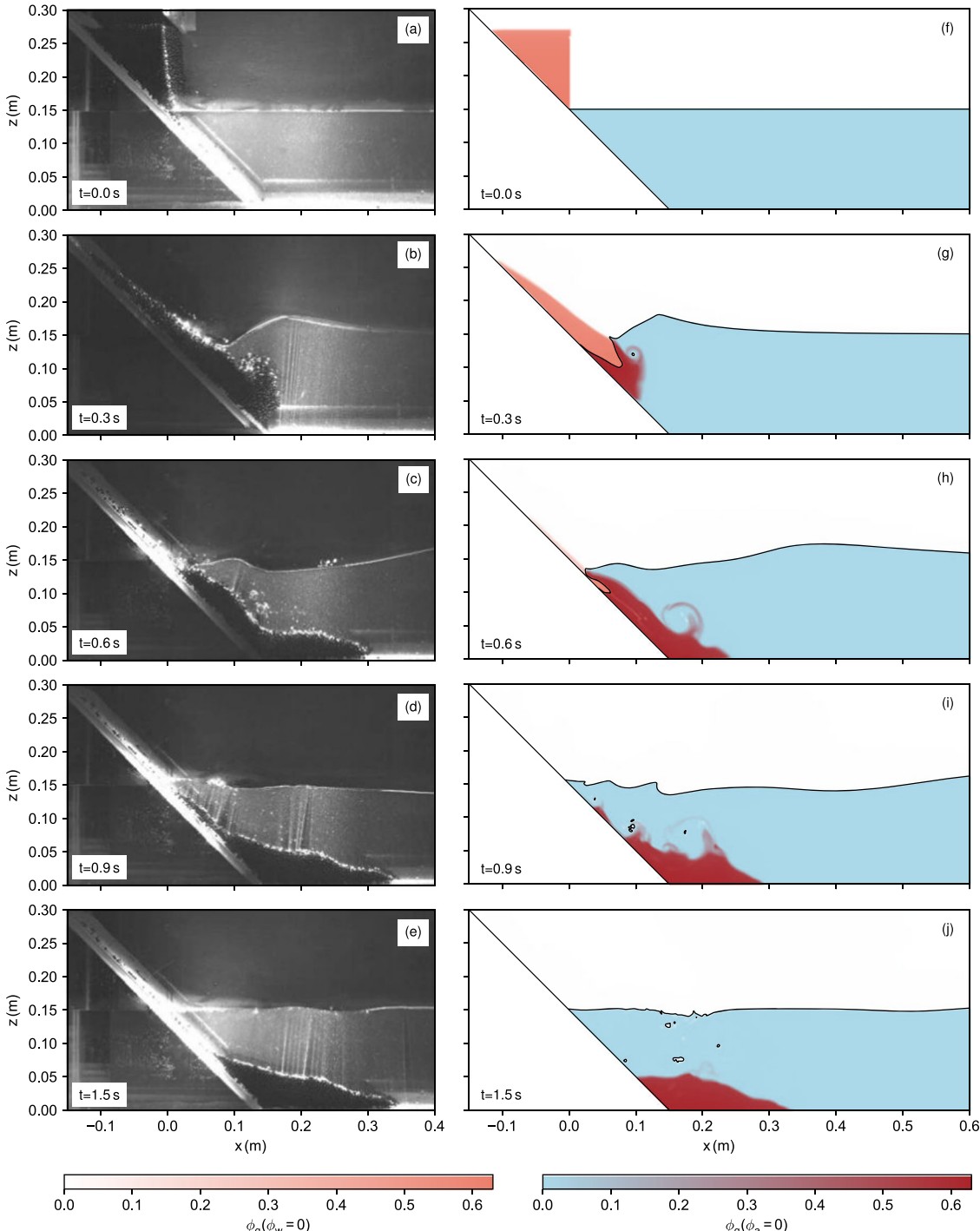

**Fig. 3 Small-scale model results.** Pictures of the experiments of Viroulet et al.[23] (**a–e**) and snapshots of the respective numerical simulation (**f–j**). The numerical results are visualised in terms of phase fractions. The interface between water and air is highlighted as a black line. The left colour map indicates the limiting case of dry regions, the right colour map the limiting state of water-saturated regions. Few regions are partially saturated due to the sharp interface between air and water. The full colour map is shown in Fig. 2.

landslide mass and slope angle). This could not be achieved with simplified rheologies[19,31,32] and highlights the value of a realistic rheological model.

**Further remarks on the model**. The front of the slide is slightly slower in the numerical simulation and the slide is shorter and more densely packed (compare Fig. 3c, h). This could be related to a mismatch of parameters or general modelling errors but also the

release mechanism of the experimental setup might be responsible. However, the final run-out distance is not affected by this discrepancy and fits the experiment well. The final deposition shows a slope angle close to the friction angle of 23°, which is consistent with the granular rheology. The bump in the middle of the deposition is more pronounced in the simulation, which can be traced back to an overestimation of the turbulence in this region. In fact, the deposition pattern would fit the experiment

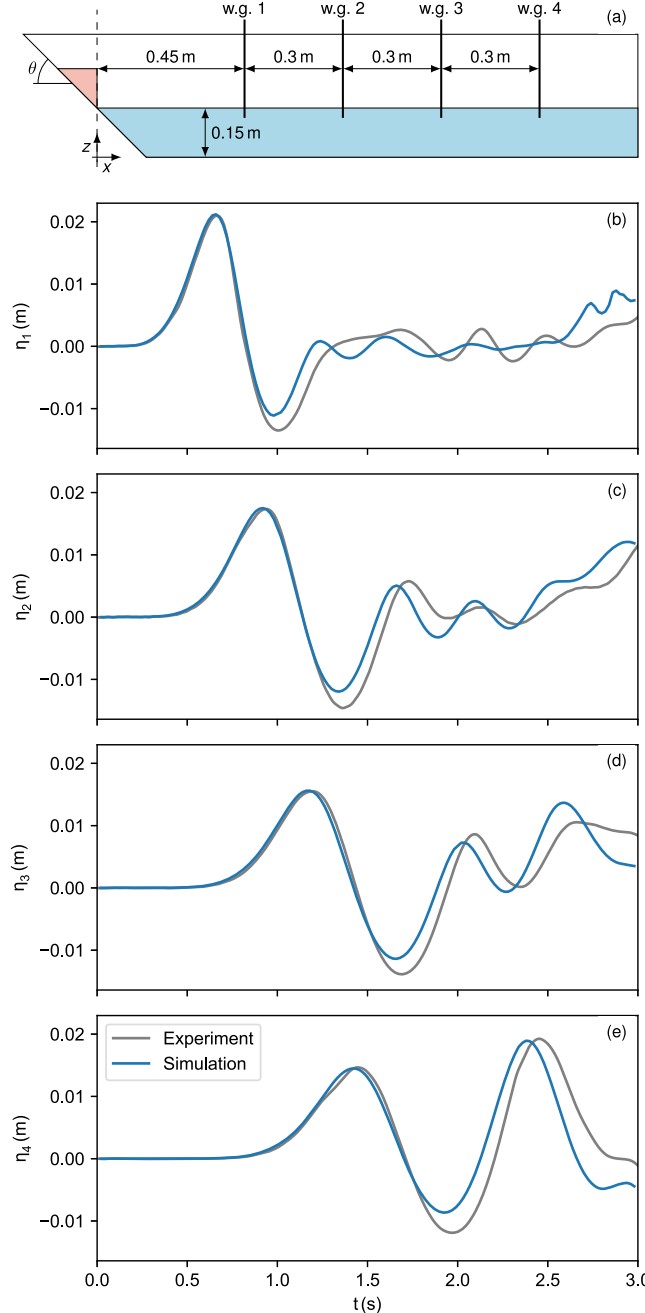

**Fig. 4 Wave signal in the small-scale model. a** The simulation setup with the position of the wave gauges (w.g.) at $x = 0.45, 0.75, 1.05, 1.35$ m. The slope angle $\theta$ is 45° in the presented case. **b–e** Water elevation over time at the four wave gauges in the simulation and the experiment. Note that there is some freedom concerning the definition of $t = 0$, i.e. the synchronisation between simulation and experiment. Here $t = 0$ is the time at which the gate is opened completely.

better with the application of a sub grid turbulence model (see supplementary materials), but the wave is mostly unaffected.

The porosity of the slide and its effects are clearly visible in the simulations. Porosity dampens the wave generation by reducing the displaced water volume, increasing the bulk density of the submerged slide and allowing us to distinguish between pore pressure and effective pressure. The correct determination of effective pressure is imperative for the slide dynamics[19,24] and it follows that porosity is equally important. The velocity with which water enters the slide is controlled by permeability. The

permeability also controls the migration and release of trapped air bubbles. We do not expect an accurate prediction of the bubble dynamics, because surface tension and a full three-dimensional resolution of bubbles are not included in the model. However, the influence of single bubbles on the generated wave is small and the approximation is reasonable if the macroscopic behaviour of the slide and the wave is of interest.

The simulated wave (Fig. 4) matches the measurements from the experiment well (mean error 1.83 mm, see supplementary materials). The first wave crest is reproduced accurately by the numerical model (error 0.1 mm or 0.5%) while the first trough is underestimated (error 2.1 mm or 15.5%), likely a result of the underestimated slide velocity.

Numerical diffusion, which unrealistically reduced the wave amplitude in the far-field in previous applications, see e.g. Løvholt et al.[15], is well controlled in this simulation and the wave crest at the last gauge is still equally well-matched as at the first gauge (compare Fig. 4b, e). The rest of the wave train matches the experimental wave signal well and some of the discrepancies in the near field are reduced in the far-field due to the wave kinematics, which is captured accurately.

**The Lake Askja landslide tsunami**. The 2014 Lake Askja flank collapse and the associated tsunami[6] is an outstanding example to demonstrate the application in complex, natural terrain at a large scale. The event is exceptionally well documented[6,33] and thus provides a good large-scale test and benchmark. The rockslide with a volume of $2 \cdot 10^7$ m³ was triggered 150 m above the lake and generated a considerable tsunami with inundations up to 70 m above the resting lake. The tsunami deposited dark sediments onto the snow-covered shores and the inundation could be accurately documented after the event. Terrain and bathymetry data before and after the event are available, which allows the location of the failure plain in the slope and the deposition of the slide in the lake. Constitutive parameters for the rockslide are highly uncertain but can be estimated from observations and comparable materials with sufficient accuracy (see Table 1 and section 3.3). Finally, the lake represents an enclosed water body and by simulating the whole lake we can avoid complex boundary conditions within the water body that would eventually lead to incorrect wave reflections.

**Lake Askja model results**. The simulated landslide dynamics is visualised in Fig. 5 in terms of the iso-surface for the granular phase fraction $\phi_g = 0.25$, which roughly corresponds to the dense core of the slide[30]. The dense core is surrounded by a fluidised and suspended particle cloud, which is illustrated through the iso-surface $\phi_g = 0.01$ in the same figure (slightly transparent). The elevation of the water surface due to the slide impact is shown in Fig. 6. A vertical transect through the slide and the region of impact is shown in Fig. 7, highlighting the mixture of granules, water and air in terms of the phase fractions. The different flow regimes, i.e. the dense core and the fluidised or suspended particle cloud can be recognised well in this figure. The maximum inundation is highlighted in Fig. 8, alongside the documented inundation and a posteriori optimised depth-integrated simulations (conducted by Gylfadóttir et al.[6]).

The slide accelerates quickly after its release and reaches velocities up to 60 m s⁻¹ before hitting the lake at $t = 20$ s (Fig. 5b). It should be noted that the shown velocity is not depth-averaged and that it cannot be compared with depth-averaged landslide models or block models (as used by Gylfadóttir et al.[6]). Assuming a Bagnold velocity profile[34], as common in granular flows, the depth-averaged velocity follows as $|\bar{\mathbf{u}}_g| = 0.6\,|\mathbf{u}_g| \approx 36$ m s⁻¹. The slide is slowed down abruptly

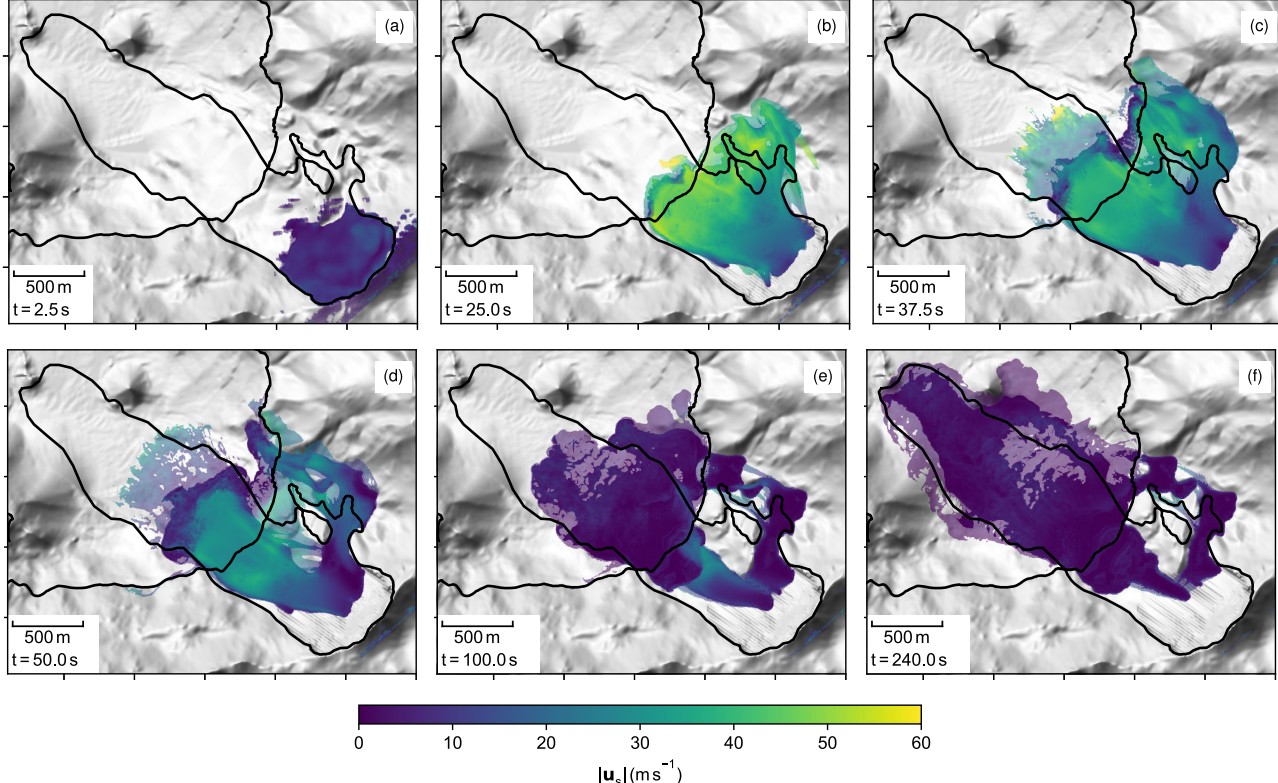

**Fig. 5 The simulated Lake Askja landslide. a–f** The dense core of the slide is represented by the $\phi_g = 0.25$ iso-surface and the surrounding dilute particle cloud by the $\phi_g = 0.01$ iso-surface. The colour shows the particle velocity and the dilute cloud is shown transparently. The black lines show the lake shoreline and the documented avalanche path for reference. For an animated version see supplemented movies 1 and 2.

when hitting the lake surface, as substantial kinetic energy is transferred into the water (Fig. 5c).

The impact of the slide into the lake is accompanied by rapid accelerations and strong turbulences. The impact angle of the slide is very shallow and the slide pushes the lake water horizontally, best seen in the vertical transect in Fig. 7. Parts of the slide are deflected by the water and flow above the water surface before sinking to the bottom of the lake. Large volumes of air are enclosed in the slide and can only escape slowly (Fig. 7d). Mixing of water and slide material takes primarily place at the front of the slide, where a diluted particle cloud, similar to a turbidity current[35], is formed. The turbidity current propagates far into the lake (Fig. 5d–f) and is only stopped by an elevation in the middle of the lake. The partially saturated dense core comes to rest near the shore where substantial parts of the slide remain above the water level (Fig. 7e).

The water that is rapidly displaced by the slide forms a turbulent wave that reaches up to 40 m above the still water level (Fig. 6b). At $t = 65$ s, roughly 45 s after the impact, the leading wave propagated ~1000 m from the impact area and escaped the turbulent region (Fig. 6c). At $t = 80$ s, the wave overflows the small island in the south of the lake and breaks in its wake (Fig. 6d). From here the wave travels across the lake until the first wave crest reaches the opposite shore at $t = 150$ s, roughly 130 s after the impact (Fig. 6e). The propagation of the wave across the lake is accompanied by run-ups on the south and north side of the lake, reaching well above 40 m inundation height (Fig. 8). The maximum inundation at the opposite shore is reached later as the second and third wave crests reach the shore. In some regions, the maximum inundation is reached very late (after $t = 240$ s) and we cannot exclude that additional areas would have been inundated after the simulation is terminated (at $t = 300$ s).

**Lake Askja model performance**. The macroscopic slide kinematics and final run-out pattern are represented well by the numerical simulation. The simulated slide moves within the documented slide path, except for the orographic right branch, which is substantially stronger in the numerical simulation. The reasons for this discrepancy could be manifold, but we suspect uncertainties in the initialisation of the landslide geometry or the packing density and the related initial stresses and weak layers. We find it likely that the initial failure occurred on the southern side of the failure area and that this asymmetric failure lead to an exaggerated flow of the slide towards the orographic left side. This effect is not included in the simulation and the right branch of the slide is respectively exaggerated. The subaquatic part of the slide flows very slowly ($|\mathbf{u}_g| < 10$ m s$^{-1}$) after the abrupt deceleration due to the impact in the lake. The slide keeps moving for an extended period and is only stopped by an elevation after ~3000 m flow distance. This behaviour can be traced back to the suspension of grains in water, reducing the packing density and contacts between grains substantially. The effective pressure is respectively small and the same follows for the viscosity and the basal friction of the slide. This indicates that the final run-out is not sensitive to the frictional properties of the slide and we anticipate that a landslide with a different friction angle would have reached the same run-out, limited only by the elevation in the lake bottom. A second simulation with a lower friction angle (see supplementary materials) reached the same run out and thus strengthened this hypothesis. These results render it unlikely that the deposition alone is a reliable indicator for the tsunamigenic potential. Scenarios with similar run out but e.g. different impact velocities will differ strongly in their tsunamigenic potential, which highlights once more the value of consistent models, describing the granular behaviour and water interaction more

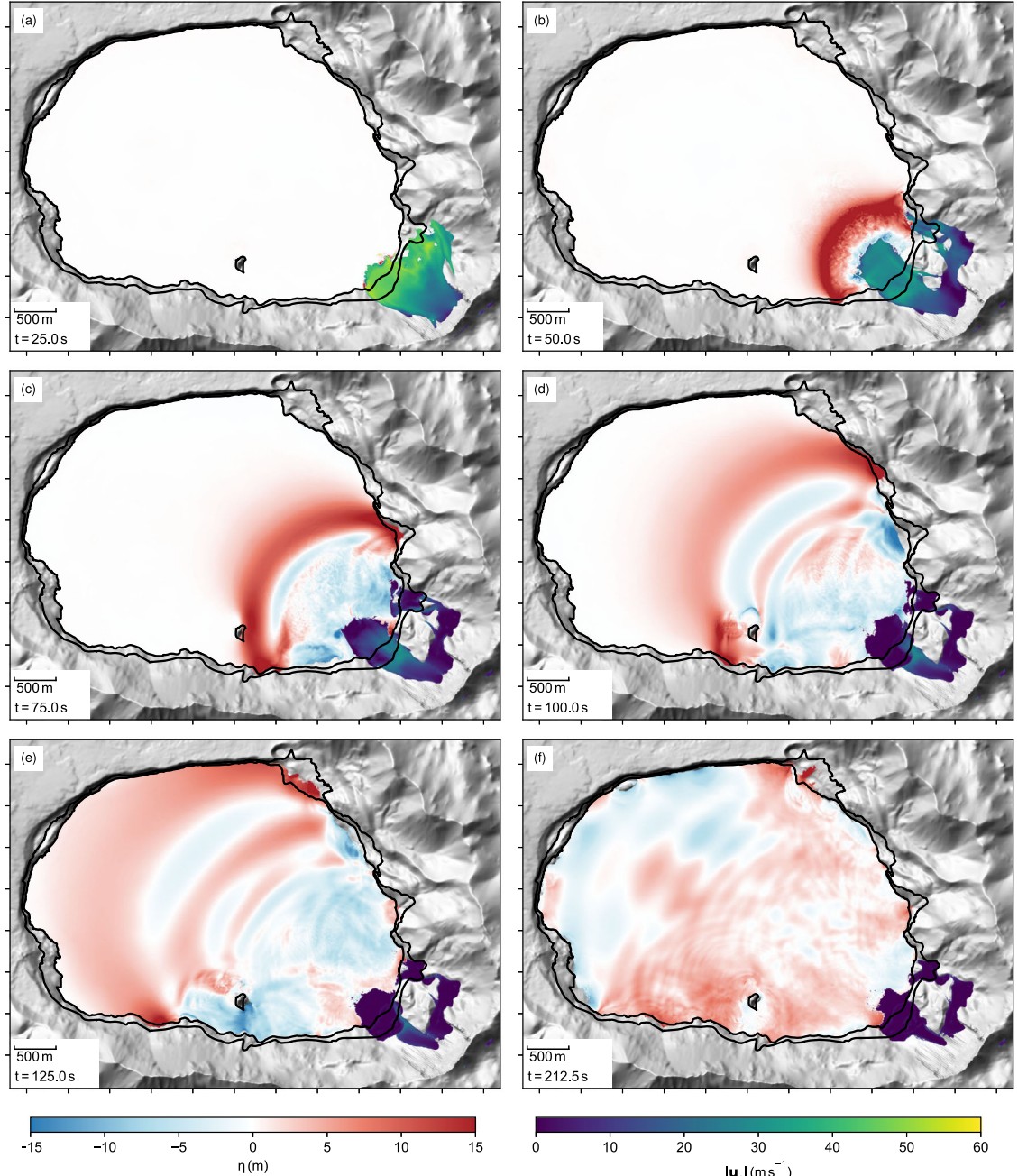

**Fig. 6 The simulated Lake Askja landslide tsunami. a–f** The water surface is represented by the iso-surface $\phi_w/(\phi_w + \phi_a) = 0.5$ and the surface elevation $\eta$ is calculated as the difference to the surface at rest at $z = 1058.25$ m. The dense core of the landslide is shown similar as in Fig. 5. For an animated version see supplemented movies 3 (top view) and 4 (perspective view).

realistically. It should be noted that such results would not have been possible with simpler nonporous slide models.

The simulation gives us detailed insights into the tsunami generation process and the interaction between the slide and lake water. The lake water is mostly pushed horizontally, building up a wave in front of the slide. Parts of the slide are deflected upwards (see Fig. 7b), before sinking into the lake and forming a turbidity current. This was not observed in the small-scale experiments of Viroulet et al.[23] due to the steeper slope. Bougouin et al.[36] present experiments that relate more closely to the observed wave generation mechanism, i.e. fast, fluidised, subaerial landslide impacts. Their experiments show a vertical granular jet at the impact and granular material flowing on the water surface before sinking into the lake and forming a turbidity current. This

mechanism fits the presented simulations well and gives us confidence that we capture the important physical processes. Further, this gives a hint that the wave generation mechanisms might be highly diverse and that complex models are required to cover multiple regimes, scales, and geometries.

The documented inundation is matched by the numerical model with an average error of 7.7 m across the periphery of the lake (see supplementary materials for more details), as shown in Fig. 8. The inundation tends to be overestimated on the northern shore and underestimated on the southern shore. This is consistent with the simulation error in the landslide, flowing overly strong towards the northern side. We conclude from this observation that the wave generation is represented well by the model and that the systematic error in the wave reflects, at least to

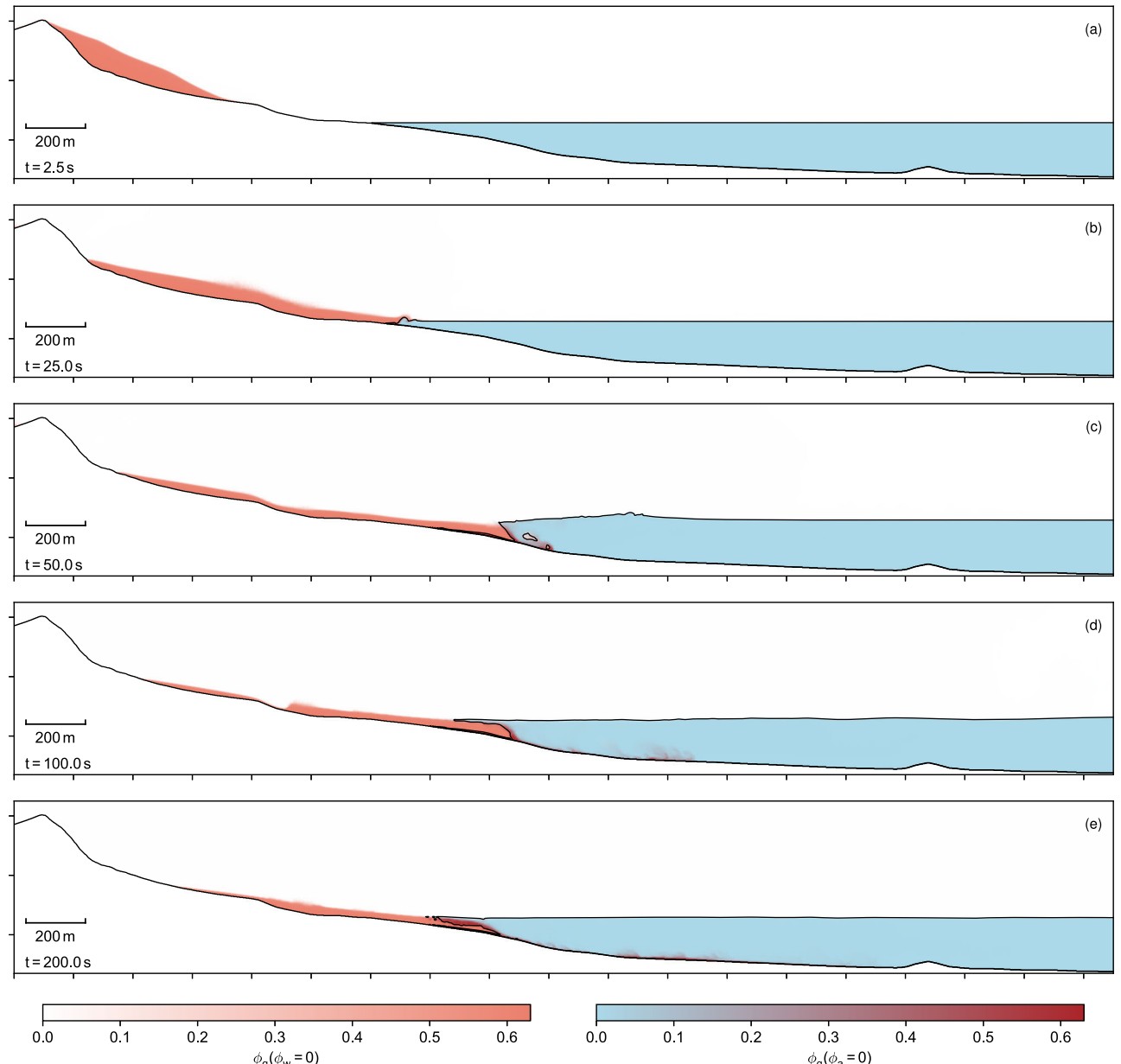

**Fig. 7 Vertical slice through the Lake Askja landslide tsunami. a–e** The colour represents the local phase fractions. The free water surface and the mesh boundary are highlighted as a black line. The left colour map indicates the limiting case of dry regions, the right colour map the limiting state of water-saturated regions. Few regions are partially saturated due the sharp interface between air and water. The total length of the transect is 3700 m and the axes are scaled equally. For an animated version see supplemented movie 5.

some degree, the error in the simulated slide. Further, the wave propagation might involve some degree of artificial damping due to numerical diffusion and thus an underestimation of the inundation on the opposite shore. However, the small-scale experiment and the mesh refinement study (see supplementary materials) suggest that the numerical diffusion is very small and we do not expect this aspect to play a significant role.

**Comparison with previous models**. The result of our three-dimensional model is compared with the depth-integrated simulations of Gylfadóttir et al.[6] in Fig. 8. The depth-integrated models utilise sliding blocks as wave sources and their kinematics were optimised to fit the observed inundation. The error of our model (mean error 7.7 m and maximum error of 32.24 m or 41% of the maximum inundation) is comparable with the error of the

depth-integrated models (mean errors of 8.16 and 6.04 m and maximum errors of 39.5 and 30.9 m for the shallow water and Boussinesq equations, respectively). Given that the presented model was not subject to any kind of optimisation (i.e. class A prediction[37]), we consider this fit with observations to be remarkably close. Interestingly, the maximum depth-integrated slide velocity (36 m s$^{-1}$) is just slightly higher than the back-calculated velocity of Gylfadóttir et al.[6] (31 m s$^{-1}$). This shows that evidence left by the wave, such as the inundation, allows remarkable conclusions on the generation event, i.e. the landslide. In fact, this suggests once more that indirect evidence may be more reliable than direct evidence, such as the deposition[38,39].

**Numerical uncertainty and mesh resolution**. The good match of the simulated slide with the documentation indicates that the mesh

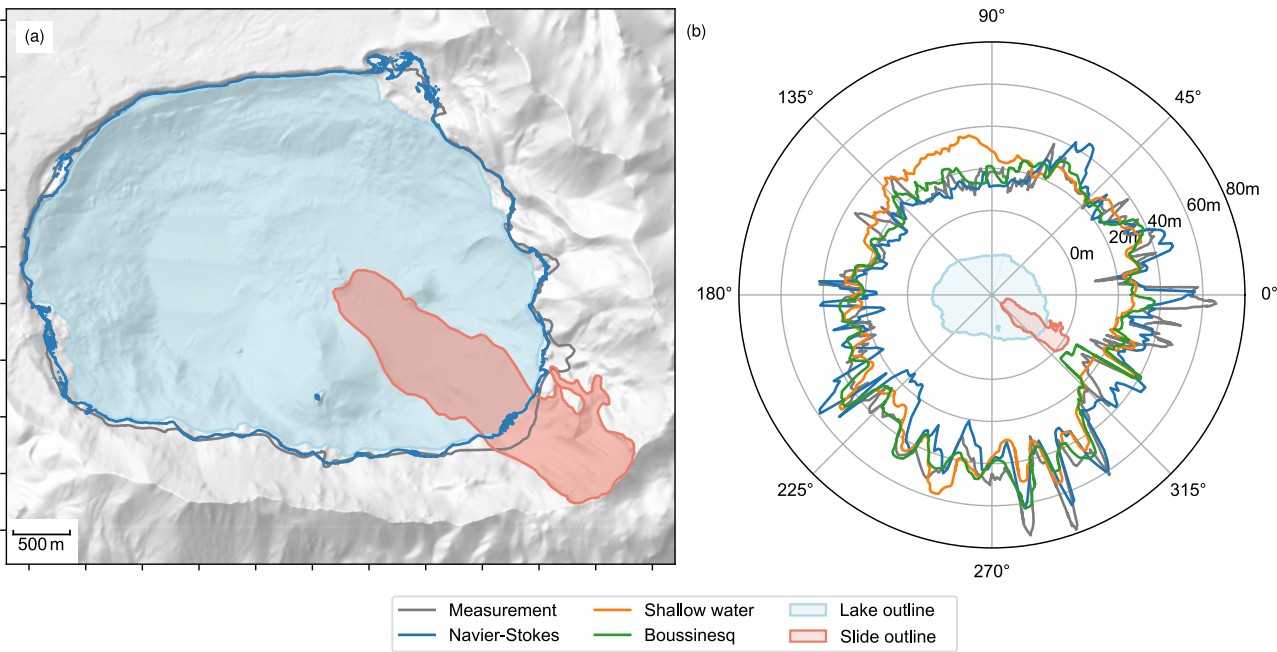

**Fig. 8 Comparison of modelled and measured inundation. a** Simulated maximum inundation (blue) compared to the measured maximum inundation (grey). The lake at rest and the documented slide outline are shown filled. **b** Maximum run-up as a function of the azimuth from the centre of the lake. The result of our model is shown in blue, posteriori optimised depth-integrated simulations are shown in green and orange and the measured maximum inundation is shown in grey. The lake and the slide path are shown for orientation in the centre.

resolution is sufficient to cover the macroscopic slide kinematics. A mesh coarsening study (see supplementary materials) confirmed that the wave-relevant slide kinematics (e.g. front velocity) are captured well and widely independent of the mesh resolution, while details and the slide tail (where the flow thickness is low) are lost in low-resolution simulations. It is clear that the limited numerical resolution of 2.5 m allows only the tracking of structures of similar length scales. Smaller structures (e.g. flow arms with flow thickness $\lesssim 1.25$ m, bubbles with diameter $\lesssim 2.5$ m) will not appear in the simulation, however, their macroscopic influence is limited. Accuracy is also compromised where only a few or a single computational point are available across the flow depth.

Numerical simulations, especially those with three-dimensional models, are always subject to numerical diffusion, over-proportionally dampening the wave in the far-field. Previous investigations[22,40] showed that the numerical resolution (i.e. cell size) must not be larger than 1/10 of the wave amplitude for an accurate simulation of wave propagation. The cell size of 2.5 m should thus be sufficient to accurately simulate the wave in the near field with wave amplitudes above 25 m. The wave amplitude is reduced in the far-field and covers only six cells, which might lead to numerical diffusion. However, neither the results nor the mesh coarsening study (see supplementary materials) revealed substantial traces of numerical diffusion or uncertainty and we conclude that the mesh resolution is appropriate. It should further be noted that the presented simulation is very well resolved in comparison with similar studies[32,41].

The impact area appears highly turbulent and the application of a sub grid turbulence model seems appropriate, as the mesh resolution is not sufficient to resolve all turbulent structures. However, the sub grid turbulence model is not expected to substantially influence the general macroscopic results (see supplementary materials). This aspect was neglected in this work and could be further investigated in the future.

The Lake Askja event is challenging for a three-dimensional landslide tsunami model because the wave amplitude is very shallow in relation to the lake size ($\eta_{max}/L \approx 0.004$). This means that a high amount of computational cells is required to capture the small water surface elevation across the whole lake. Other cases, e.g. the Vajont event ($\eta_{max}/L \approx 0.1$) might be better suited for three-dimensional models. However, the vast majority of landslide tsunami cases might show a relative wave amplitude similar or smaller to the Lake Askja case. Hence, this case may be more relevant and a better benchmark for three-dimensional and depth-integrated tsunami models. A solution to this issue might also be the coupling of a three-dimensional model in the near-field to a depth-integrated model for the far-field wave propagation[15].

**Discussion.** This work represents a major step forward in modelling subaerial landslide tsunamis. Hindcasting the small-scale experiment of Viroulet et al.[23] demonstrates remarkable accuracy of wave kinematics and landslide run out, without the need for optimised parameters. Other observable details, such as the water intrusion and the frontal vortex appeared naturally in the simulations. The numerical results further compare well with the 2014 Lake Askja landslide tsunami at full scale, predicting tsunami run-up heights and landslide run-out as deduced from field observations[6]. Again, phenomena such as vertical granular jets occurred naturally in the simulations. However, the simulations are computationally expensive (see method section) and uncertainties of various nature (numerical, geological, constitutive parameters) are present. Differences between observations and simulations are small and improved agreement can most likely be obtained with a more careful estimation of the constitutive parameters, higher grid resolutions or by including additional processes, such as turbulence or surface tension. A long-standing problem of granular flow models, the apparent reduction of the friction coefficient for very large events[42,43], remains unsolved in the presented model and it will be a major challenge to identify the responsible processes in the future. The presented model

might be helpful in this endeavour as well, providing a strong platform for further developments.

The model is sufficiently complex to accurately predict the landslide dynamics and wave generation, but still efficient enough to tackle full three-dimensional problems. It is presently too computationally expensive for parametric studies or probabilistic analyses[39], but provides fundamental new insight into the physics of landslide tsunamis. This model may hence be used operationally, to build an understanding of the involved processes or as a benchmark for a new generation of more efficient models. Calibrating and developing depth-integrated models with porous and granular character[7,44,45] represents a possible path forward, while dynamic coupling with depth-integrated tsunami models for the far-field propagation[15,46] represents another promising approach, deemed necessary for resolving problems where the tsunami propagates over long distances. Notably, the discrete element method (DEM)[47] provides an alternative to continuum-mechanical models, however, for a substantially higher computational cost that scales unfavourable with the size of the event.

Full three-dimensional methods are relatively rare in the tsunami community but their application and the respective publications have been rising consistently in the last few years. In the present paper, we show that such models may provide the necessary paradigm shift to understand and predict landslide tsunamis. The $\mu(I),\phi(I)$-rheology plays a central role in this endeavour, because it allows to include the granular and porous nature of the slide. With the increasing computational capabilities expected in the near future, utilisation of fully three-dimensional models can become mainstream. This represents a unique opportunity to improve the protection of coastal communities and our understanding of devastating landslide tsunamis.

## Method

**Mathematical model**. We start with the hypothesis that a sufficiently realistic model for landslide generated tsunamis needs to implement the following key elements (in addition to conservation of mass and momentum):

1. a sharp water–air interface with low diffusivity,
2. granular rheology for the landslide,
3. differentiation between effective pressure, pore pressure and total pressure,
4. porosity, dilatancy (i.e. porosity changes) and permeability.

The first point is imperative for an accurate wave description. Numerical schemes with poor properties can lead to artificial wave dampening and a systematic underestimation of the tsunami. Points two and three are important for the landslide kinematics and the interaction with the water. Simplified rheologies, e.g. Bingham fluids, are not capable of describing the landslide kinematics[31,32] and the same is the case for granular rheologies that do not account for hydrostatic and excess pore pressure[19,24,27]. Last but not least, the porosity can have a strong influence on the wave generation, absorbing substantial amounts of water and dampening the wave generation[36,48]. These aspects are strongly related to one another: granular rheologies require effective pressure which is related to porosity as described by the critical state theory[27,49]. Modelling one without the others is, to our understanding, not possible.

The applied model can be derived from the tri-phase (granules, water and air) Navier–Stokes equations and the assumption that air and water move with the same velocity field (for details see supplementary materials). This assumption is reasonable and has great advantages in terms of diffusivity and stability at the water–air interface[50]. Further, the model is closed with constitutive models for internal stresses and inter-phase momentum exchange (i.e. drag). Surface tension and capillary effects are not included in this study, although they might play a role in small-scale experiments[51]. The same is the case for sub grid turbulence that might play a role at high Reynolds numbers (see supplementary materials for a short investigation).

The model is defined in terms of phase fractions for granules $\phi_g(\mathbf{x}, t)$ and the pore fluid $\phi_c(\mathbf{x}, t)$, as well as the respective phase velocities $\mathbf{u}_g(\mathbf{x}, t)$ and $\mathbf{u}_c(\mathbf{x}, t)$ (see Fig. 9). The variables in brackets $(\mathbf{x}, t)$ indicate that these are fields that change in space $(\mathbf{x})$ and time $(t)$. The fluid phase is further split into two components, air and water, that are described by component indicator functions $\alpha_a(\mathbf{x}, t)$ and $\alpha_w(\mathbf{x}, t)$.

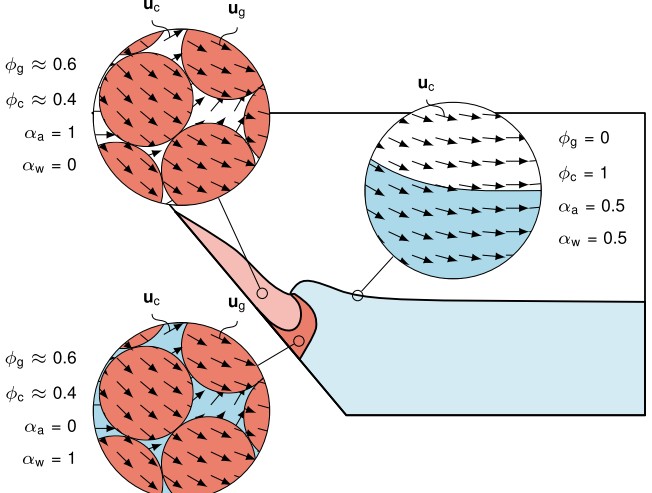

**Fig. 9 Sketch of the mathematical model of a dry subaerial landslide impacting a water reservoir and generating a tsunami.** The mathematical model of this process is defined in terms of phase fractions $\phi_g$ and $\phi_c$, component indicator functions $\alpha_a = \phi_a/(\phi_a + \phi_w)$ and $\alpha_w = \phi_w/(\phi_a + \phi_w)$ and phase velocities $\mathbf{u}_g$ and $\mathbf{u}_c$.

The phase fractions are defined as the volume occupied by the respective phase per unit Volume $V$,

$$\phi_i = \frac{V_i}{V}. \tag{1}$$

Phases move independently from one another and are only coupled by drag and pressure terms. Components $\alpha_j(\mathbf{x}, t)$ are defined as 1 if the respective component is present at position $\mathbf{x}$ and time $t$ and 0 otherwise,

$$\alpha_j(\mathbf{x}, t) = \begin{cases} 1 & \text{phase } j \text{ present at } \mathbf{x}, t, \\ 0 & \text{otherwise.} \end{cases} \tag{2}$$

Components differ from phases by moving with the same velocity as components of the same phase and by having a sharp interface between them. Component indicators between 0 and 1 are not intended, following the assumption of a sharp interface between components. However, such values will appear due to the numerical discretisation. The volume fraction of a specific component $j$ of phase $i$ can be calculated as

$$\phi_j = \phi_i \alpha_j. \tag{3}$$

We define two sets of mass and momentum conservation equations[44,45,50,52,53]. The first set describes the granular material in terms of the granular phase fraction $\phi_g(\mathbf{x}, t)$ and the grain velocity $\mathbf{u}_g(\mathbf{x}, t)$ (see Fig. 9) as

$$\frac{\partial \phi_g}{\partial t} + \nabla \cdot (\phi_g \mathbf{u}_g) = 0, \tag{4}$$

$$\frac{\partial \phi_g \rho_g \mathbf{u}_g}{\partial t} + \nabla \cdot (\phi_g \rho_g \mathbf{u}_g \otimes \mathbf{u}_g) = \\ \nabla \cdot (2 \phi_g \rho_g \nu_g \mathbf{S}_g) - \nabla p_s - \phi_g \nabla p + \phi_g \rho_g \mathbf{g} + \mathbf{m}_{gc}. \tag{5}$$

The gravitational acceleration $\mathbf{g}$ is assumed to be $(0, 0, -9.81)^T$ m s$^{-2}$. The interaction with the pore fluid is considered with the pore (or shared) pressure $p(\mathbf{x}, t)$ and the drag force per unit volume $\mathbf{m}_{gc}(\mathbf{x}, t)$ (see below). The granular viscosity $\nu_g(\mathbf{x}, t)$ follows from the $\mu(I)$-rheology[27,28,54,55] as

$$\nu_g = \mu(I) \frac{p_s}{2 \overline{\phi} \rho_g} \frac{1}{\|\mathbf{S}_g\|}, \tag{6}$$

with the friction coefficient

$$\mu(I) = \mu_s + \frac{\mu_d - \mu_s}{I_0/I + 1}, \tag{7}$$

and the inertial number

$$I = \frac{2 d \|\mathbf{S}_g\|}{\sqrt{p_s/\rho_g}}. \tag{8}$$

The granular viscosity is limited to an interval $[\nu_{min}, \nu_{max}]$ to avoid numerical issues[27,56]. The granular deviatoric shear rate tensor $\mathbf{S}_g(\mathbf{x}, t)$ is defined as

$$\mathbf{S}_g = \frac{1}{2}(\nabla \mathbf{u}_g + (\nabla \mathbf{u}_g)^T) - \frac{1}{3}\nabla \cdot \mathbf{u}_g \mathbf{I}, \tag{9}$$

with the identity matrix $\mathbf{I}$. The effective pressure or particle pressure $p_s(\mathbf{x}, t)$ follows from the combination of the $\phi(I)$-theory[30] and the critical state theory[27,49,57] as

$$p_s = a \frac{\phi_g - \phi_{rlp}}{\phi_{rcp} - \phi_g} + \rho_g \frac{\phi_g}{\overline{\phi}} \left( 2 \parallel \mathbf{S}_g \parallel d \frac{\Delta\phi}{\phi_{rcp} - \phi_g} \right)^2. \tag{10}$$

The first term in Eq. (10) is dropped at packing densities below $\phi_{rlp}$. For flows with a small Stokes number, slightly different scaling laws should be applied[58,59] and kinetic theory should be applied to diluted particle flows[52,53,60]. Material parameters for the granular phase are the grain density $\rho_g$, the grain diameter $d$, the friction coefficients $\mu_s$ and $\mu_d$, the reference inertial number $I_0$, the random loose packing density $\phi_{rlp}$, the random dense packing density $\phi_{rcp}$, the pressure scaling factor $a$, the dynamic dilatancy factor $\Delta\phi$ and the reference packing density $\overline{\phi}$. The reference packing density is required to make the compressible rheology consistent with the incompressible $\mu(I)$-rheology[27].

The second set of equations describes the fluid phase (i.e. water and air) in terms of the fluid phase fraction $\phi_c(\mathbf{x}, t) = 1 - \phi_g$ and fluid velocity $\mathbf{u}_c(\mathbf{x}, t)$ as

$$\frac{\partial \phi_c}{\partial t} + \nabla \cdot (\phi_c \, \mathbf{u}_c) = 0, \tag{11}$$

$$\frac{\partial \phi_c \rho_c \mathbf{u}_c}{\partial t} + \nabla \cdot (\phi_c \rho_c \mathbf{u}_c \otimes \mathbf{u}_c) = \\ \nabla \cdot (2 \phi_c \rho_c \nu_c \mathbf{S}_c) - \phi_c \nabla p + \phi_c \rho_c \mathbf{g} + \mathbf{m}_{cg}. \tag{12}$$

The composition of the fluid phase is described by component indicator functions $\alpha_a(\mathbf{x}, t)$ and $\alpha_w(\mathbf{x}, t)$ (see Eq. (2)) and the fluid density and viscosity follow as

$$\rho_c = \alpha_a \rho_a + \alpha_w \rho_w, \tag{13}$$

$$\nu_c = \frac{(\alpha_a \rho_a \nu_a + \alpha_w \rho_w \nu_w)}{\rho_c}, \tag{14}$$

with the densities and viscosities of air ($\rho_a$, $\nu_a$) and water ($\rho_w$, $\nu_w$). The deviatoric shear rate tensor of the fluid $\mathbf{S}_c(\mathbf{x}, t)$ is defined in analogy to Eq. (9). Component indicator functions are transported by the fluid phase velocity and are tracked with the advection equations,

$$\frac{\partial \phi_c \alpha_a}{\partial t} + \nabla \cdot (\phi_c \alpha_a \mathbf{u}_c) + \nabla \cdot (\phi_c \alpha_a \alpha_w \mathbf{u}_{aw}) = 0, \tag{15}$$

$$\frac{\partial \phi_c \alpha_w}{\partial t} + \nabla \cdot (\phi_c \alpha_w \mathbf{u}_c) + \nabla \cdot (\phi_c \alpha_w \alpha_a \mathbf{u}_{wa}) = 0, \tag{16}$$

that can be derived from the respective mass conservation equations. The third term in Eqs. (15) and (16) are artificial interface compression terms which allow an accurate representation of the surface wave[22,50]. The relative velocity $\mathbf{u}_{aw}(\mathbf{x}, t)$ is calculated to be of the same magnitude as the phase velocity and normal to the interface between air and water,

$$\mathbf{u}_{aw} = -\mathbf{u}_{wa} = c_{\alpha,aw} |\mathbf{u}_c| \frac{\alpha_w \nabla \alpha_a - \alpha_a \nabla \alpha_w}{|\alpha_w \nabla \alpha_a - \alpha_a \nabla \alpha_w|}. \tag{17}$$

The constant $c_{\alpha,aw} = [0, 4]$ can be used to adjust the interface compression but is usually chosen to be 1 (as is the case in this work). Note that the sum of the compressed fluxes of Eqs. (15) and (16) is used to solve the advective term of the momentum equation (12), to avoid diffusion of momentum across the component interface. This showed to be important for accuracy and stability.

The drag force of particles in the fluid per unit volume, $\mathbf{m}_{gc}(\mathbf{x}, t)$, couples the two phases and follows from the model of Ergun[61],

$$\mathbf{m}_{gc} = -\mathbf{m}_{cg} = \left( 150 \frac{\phi_g^2 \nu_c \rho_c}{\phi_c^2 d^2} + 1.75 \frac{\rho_c \phi_g}{\phi_c d} |\mathbf{u}_g - \mathbf{u}_c| \right) (\mathbf{u}_c - \mathbf{u}_g). \tag{18}$$

It combines the Kozeny–Carman relation[27,44] with a higher-order term that helped to improve the stability in this study.

The shared pressure $p(\mathbf{x}, t)$ follows from and enforces the incompressibility of phases,

$$\phi_g + \phi_c = 1, \tag{19}$$

and the respective constraint on the average velocity

$$\nabla \cdot (\phi_g \mathbf{u}_g + \phi_c \mathbf{u}_c) = 0. \tag{20}$$

The time-stepping is controlled by limiting the Courant–Friedrich–Lewy numbers[62] to well-defined values. The Courant–Friedrich–Lewy numbers for the presented model can be written in a simplified and approximated form as[22]

$$\text{CFL}_i^{\text{conv}} = \frac{u_i \Delta t}{\Delta x} \tag{21}$$

and

$$\text{CFL}_i^{\text{diff}} = \frac{\nu_i \Delta t}{\Delta x^2}, \tag{22}$$

where $\Delta x$ is the grid size and $\Delta t$ the time step duration.

The mathematical model is solved with the open-source toolkit OpenFOAM®[63,64] and its rich multiphase flow library[50,52]. OpenFOAM provides a wide range of functionalities, offering a stable and efficient implementation. Most notable is the MULES framework (multidimensional limiter for explicit solution), that provides the majority of the required multiphase and multicomponent functionality. Furthermore, the rich OpenFOAM infrastructure can be used for case setup and post processing. The applied solver is based on multiphaseEulerFoam, in particular the version of Rauter[27]. The subdivision of phases into components is accomplished by combining it with the code of the solver multiphaseInterFoam, as used and validated for wave generation and propagation by Romano et al.[41], Chen et al.[21] and Rauter et al.[22]. The granular rheology is implemented as a separate library which was validated in the previous publications[27,29,34].

**Setup of the laboratory-scale simulation.** The experiments were conducted in a tank with length $L = 2.2$ m, height $H = 0.4$ m and width $W = 0.2$ m. The two-dimensional mesh for this case (see supplementary materials) was generated using cartesian2DMesh, a mesh generator of the cfMesh toolbox[65]. The mesh has cell sizes down to 0.001 m close to the free surface and the granular slide and up to 0.032 m far away from important regions. Cells are mostly hexagonal and have an aspect ratio close to one. The finest resolution corresponds to 1/20 of the expected wave amplitude (following the experiment), which showed to be ideal in the previous investigations[22]. The time step was limited by $\text{CFL}_g^{\text{diff}} < 10$ in the granular phase, $\text{CFL}^{\text{diff}} < 1$ in the fluid phase and $\text{CFL}^{\text{conv}} < 0.5$ in both phases. These limits were found to be stable in previous investigations[22,27]. All boundaries are modelled as impenetrable walls ($\mathbf{u}_g = \mathbf{u}_c = 0$, $\mathbf{n} \cdot \nabla p = 0$, where $\mathbf{n}$ is the normal vector on walls).

The component indicator functions $\alpha_a$ and $\alpha_w$ are initialised to 0 and 1 in the respective regions of air and water. The granular phase fraction $\phi_g$ is initialised to match the total landslide mass and the critical state, such that the particle pressure $p_s$ is in equilibrium with the lithostatic pressure[27]. We simulate five different cases, varying the slope angle and the landslide mass. The case with $\theta = 45°$ and a landslide mass of 2 kg is highlighted in this paper, other cases with similar performance can be found in supplementary materials, alongside a rigorous sensitivity and mesh refinement study. The mesh for this case consisted of 173,288 cells, the time step duration varied between $10^{-5}$ and $10^{-4}$ s and the execution took 4.2 h on eight cores of an Intel Xeon E5-2690 v4.

**Setup of the real case simulation.** The model assumes a monodisperse granular material with constant diameter $d$ and constant friction properties. Natural rock-slides differ from this assumption by containing a large variety of grains with varying size and form[66]. The grain distribution can have substantial influences on the rheology, the packing density and the permeability of the slide[34]. Herein we neglect these influences and assume that uncertainties do not fully transfer to the macroscopic kinematics of the landslide and the tsunami. This assumption should be valid if segregation can be neglected[34]. We choose an effective diameter of $d = 0.01$ m to achieve a permeability that is higher than in the small-scale experiment but small enough to capture air for an extended period of time at the length scale of the real case event. This matches observation of rising air bubbles on the day after the event.

The quasi-static friction coefficient $\mu_s$ was chosen as $\mu_s = \sin(10°) = 0.17$, following friction parameters of similar events[42,43] but also to match the overall slope gradient from release to deposition. Notably, this friction coefficient is smaller than the intrinsic friction coefficient of the material would suggest. Reduced friction at high-pressure levels and in landslides with volumes above 500,000 m³ is a well-described effect[42,43] and it is usually considered by a reduction of the friction coefficient, similar as in this work.

The solid density was set to $\rho_g = 2500$ kg m⁻³, which results in a density of 2000 kg m⁻³ for the water-saturated dense core of the landslide at a packing density of 60%. Both parameters fit the observations and assumptions of Gylfadóttir et al.[6].

The critical state line parameters ($\phi_{rlp}$, $\phi_{rcp}$) match the small-scale experiment, except for a substantially higher scaling factor $a = 50,000$ Pa. This provides a reasonably low-pressure gradient $\partial p_s/\partial \phi_g$ at the very high-pressure level of $10^6$ Pa in the real case simulation, which improves stability. With a scaling factor of $a = 130$ Pa the vast majority of the slide would show a packing density very close to $\phi_{rcp}$, where the pressure equation has an asymptote, $\partial p_s/\partial \phi_g \to \infty$. In such a scenario, small variations in the packing density would lead to very high effective pressure spikes and eventually simulation failures. The total set of model parameters is shown in Table 1. Notably, there is substantial room for improvement and optimisation in terms of constitutive parameters. Note that the chosen parameters were not fitted by comparing the computational results with the observations. This would not be possible due to the high computational demand.

The geometry of the simulation domain, defined by the finite volume mesh, is based on a terrain model that neither contains the landslide deposition nor the initial landslide geometry (see supplementary materials). It can be seen as a combination of the terrain before the event in the deposition area and the terrain after the event in the release area (for more details on terrain data see Gylfadóttir et al.[6]). We applied cartesianMesh of the cfMesh toolbox[65] to generate a suitable

mesh defining the simulation boundary, bottom geometry and local refinements. The final mesh (see supplementary materials) consists of 30.6 M, mostly hexahedral and quadratic cells with cell sizes between 160 and 2.5 m (corresponding to six refinements levels). The mesh resolution is discussed in detail in the supplementary materials. The simulation duration was set to 300 s.

The initial landslide geometry (represented by $\phi_g$) was defined by the intersection between the terrain before the event and the mesh, which represents the terrain after the event in this region. The granular packing density $\phi_g$ matches the loose limit $\phi_{rlp}$ at the landslide surface and increases with depth, such that the particle pressure is in balance with the lithostatic pressure (i.e. in the critical state). The lake (represented by $\alpha_w$) was defined by the intersection of the mesh with the horizontal plane at $z = 1058.25$ m. The groundwater or general water content of the soil was not included in the simulation although the model is capable to do so.

The time step was limited to $t = 0.002$ s to avoid stability problems. The $CFL^{conv}$ numbers are well below the stable limits at this time step duration, except for short velocity peaks during which the time step is further reduced. The simulation was executed on 220 Intel Xeon E5-2690 v4 cores on the high-performance cluster LEO4 of the University of Innsbruck and took roughly 600 h for 300 s simulation time. The memory usage peaked at roughly 120 GB.

## Data availability

The geographic data is available upon request from Sigríður Sif Gylfadóttir (siggasif@vedur.is). The data generated in this study and the respective measurements are provided in the supplementary information. Source data are provided with this paper.

## Code availability

The code is available upon request from Matthias Rauter (matthias@rauter.it).

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

## Acknowledgements

This project has received funding from the European Union's Horizon 2020 research and innovation programme SLATE under the Marie Skłodowska-Curie grant agreement No. 721403 (M.R. and F.L.). The computational results presented have been achieved using the HPC infrastructure LEO of the University of Innsbruck. We thank Tómas Jóhannesson for interesting discussions and valuable comments, as well as the HPC team of the University of Innsbruck for their exceptional support.

## Author contributions

M.R. contributed conceptualisation, methodology, software, validation, visualisation, investigation, writing—original draft and writing—review and editing; S.V. contributed validation, visualisation, writing—review and editing, resources and data curation; S.S.G. contributed investigation, resources and data curation; W.F. contributed conceptualisation, writing—review and editing, resources and project administration; F.L. contributed conceptualisation, writing—original draft and project administration.

## Competing interests

The authors declare no competing interests.
