## [Peer Review File · Nature Communications]

Reviewers' Comments:

Reviewer #1:

Remarks to the Author:

The manuscript by Rauter and coauthors is about the application of porous granular flow models for better understanding of landslide-triggered tsunami. Tsunami caused by landslides are much more common than previously thought, a number of examples is correctly provided in the manuscript. In the year 2018 alone in Indonesia landslides caused the deadly Palu tsunami, a landslide from Krakatau volcano affected coastal regions of Sumatra and Java, and a collapse of a litoral dome on Kadovar Island lead to small tsunami at Papua New Guinea. The manuscript submitted to Nature Communication nicely describes the difficulty and lack of complexity in former tsunami models, where the tsunami source might be considered by granular medium.

I found the manuscript very interesting to read, the organization is mostly logic (see comments below), the figures are well prepared and of high quality, and the supplementary materials, especially video simulations, make the outcomes of the work enjoyable for a broad readership. A pleasure to review such a high quality piece of work, thank you! I therefore can recommend publication of the manuscript after considering the minor changes suggested as follows.

- 1) Reconsider the title. Is it about prediction of tsunami? Would mentioning of Askja be valuable here?
- 2) Abstract: Tone down the last sentence of the abstract. Otherwise the abstract is almost perfect, it summarizes the technical advances and Askja application. Could clarify that tsunami propagation physics is well understood, but the initiation and source region is not.
- 3) In parts the manuscript rather reads like a technical report instead of a nature / geologic work aiding process understanding. I would recommend to highlight the application more and move technical details to accompanying documents.
- 4) Of course tsunami are oceanic reach (line 28), i do not understand what authors want to tell me with this?
- 5) I would prefer the term landslide-triggered tsunami, instead of landslide tsunami.
- 6) L52: why a landslide should be granular only? Landslides elsewhere develop into blocks and granular flows. Also submarine studies show landslides partially move as blocks and partially in granular fashion. This would need a more careful consideration and conceptual explanation/justification.
- 7) L70 and following. Here i would suggest some changes, as the methods are appearing too prominent without even introducing the problem. I would suggest to move L71-82 to the methods section, and chapters 2.1, 2.2, 2.3, and 2.4 to the appendix. The reason for that is, that in the current version the Askja landslide is not even introduced before page 7, and the manuscript rather appears like a technical report than a nature-related study. Also fig 2 could be moved accordingly.
- 8) The result calibration to analog experiments is helpful, but could be moved to appendix/supplementary materials.
- 9) The Askja part starts after L 142 on p7. Please add here missing references, such as those that document the inundation or those that map terrain and bathymetry before and after the event.
- 10) L155. I might have overlooked it in this chapter, but the dense core assumption is based on what? Which earlier study?
- 11) fig. 5 caption. Which dense core? Please clarify the scale and time in the figure, it is hardly readable on my printout (same for fig. 6). Please specify the shaded relief/dem origin, sampling and year.

12) what is known about the initial geometry of the Askja rockslide? Looking at the figure 7 and the very simplified geometry in (a) i am wondering if there is much known about it. How relevant is the basal decollement plane, its listric curvature and complexity? Curved decollement planes desintegrate landslide into few large blocks, such as documented at the MSH rockslide (cf. <https://doi.org/10.1130/G32198.1>).

13) ch2.6. I recommend the authors to include the Viti lake as an important reference for two reasons: first, the lake was inundated and infilled by the tsunami, possibly giving an important height constraint. Second, the lake is a major touristic destination (for north icelandic standards), so that a certain hazard aspect might be demonstrated.

14) L191 - authors mention uncertainties in initialization of the landslide geometry. I recommend adding further detail and broaden the discussion adequately, considering block movements (also Toreva blocks), listric and planar decollement, and other effects.

15. L215, if already noted before authors can delete repetition.

16. L217. What exactly is "acceptable"?

17. Could results contribute to analyse and predict how deposits may look like on the lake floor? I am thinking also about the Krakatau example, where new result suggests the deposition of a number of large blocks (instead of granular rheology only)

18. The supplements and methods are described very clearly.

I hope my comments help to further refine this already excellent study.

Reviewer #2:

Remarks to the Author:

In this paper, the authors apply a CFD model with a new rheology for multi-phase flows, developed within the OpenFoam library to simulate one very small scale two-dimensional experimental benchmark and one large scale field case study of landslide tsunamis generated by a granular material subaerial slide. This is a very computationally expensive model and hence, they basically present 2 simulations (although they simulate a few cases for the less-costly experiments) with no sensitivity study of results to numerical or physical parameters.

The authors present their work as a large step forward, leading to remarkable results (such superlative statements are made many times throughout the paper) and often make strong but not always well-supported statements in the paper regarding previous work and their own work and quality of results. They imply directly in their abstract that the "lack of consistent models incorporating both the landslide and the wave mechanics represents a gap in providing unbiased predictions of real event", indicating in the text that apparently most other modelers tune their model parameters to match observations and hence other models can't be predictive ! This reviewer strongly disagrees with both statements and while the new presented model here features some new rheology and includes dilatancy effects that are missing in most models, judging by the results for wave generation and propagation, they do not get better results or get even worse results in the far-field than other modelers could achieve (based on this reviewer's experience) for similar experimental benchmarks or field case studies. Their model clearly has numerical dissipation (and they admit it), which has been well-known for decades when simulating wave propagation over a significant distance with this class of NS models. A fact that has led many modelers to implement model coupling. The latter, which has been done for over 2 decades as well, is presented here as the solution to their dissipation problem for future work.

Another important concern is over-stressing the improved accuracy of including new physics (rheology, air bubbles,...), but still considering a homogeneous porous medium with particles of the same size. This may be the case in experiments, but this is not the real world at all of flank

collapses and avalanches. Hunt et al.'s (2021) recent field work on the Anak Krakatau 2018 collapse showed deposits made of many enormous blocks (100s of meters in dimension) sitting together with small granular material and everything in between. The material used in the field case study here is 10 cm across and uniform ! So refining the modeling of a porous medium and its rheology does not necessary lead to better results for real events and, in fact, simpler models based on particles of many size and their shocks have done quite well in the past (e.g., many works from Ward and days, 2001 to Wang, 2019). Recent work in modeling the seminal and well observed AK 2018 event, using the latest generation of two-layer models (not reviewed or cited here; e.g., Zhang et al., 2021a,b) showed that non-hydrostatic Euler models that are quasi 3D in the water with a depth-integrated layer of viscous or granular material, which is non-hydrostatic as well, and a simpler rheology than here did very well in predicting the large near-field runups and by coupling to a Boussinesq model the far-field inundation and runup (Grilli et al., 2019a,b,2021; Zhang, 2020). The measured runup heights up to 85 m on AK's nearby island was closely predicted by such models (actually much closer than results presented here in the lake). Another example of successful modeling with other landslide tsunami models is Palu 2018 (Schambach et al., 2021).

One other concern about the paper is its lack of an exhaustive review to date of other well experimentally-validated landslide tsunami models also applied to real case studies, that may just use a simpler model of the granular medium than here, but have performed well for predicting waves. Perhaps those models will not simulate the slide runout and deposits as well as the proposed model (which still needs to be demonstrated in the present paper for their cases as they do not show precise slide deposits), but they model well enough the tsunamigenic part of the slide motion and wave generation to be predictive without needing parameter adjustments. The authors here either lack knowledge of that other work or prefer to cite a subset of literature related to their own group of collaborators they know better ? In any case an exhaustive and accurate review of the state of the art is missing here. And in tsunami hazard assessment, predicting the waves in the near- and far-field is more important than predicting the details of the slide deposits !

The authors also state: "For the first time, we show a model that scales consistently from small scale laboratory experiments to full scale catastrophic events. "Well, NHWAVE and its many applications to lab and field cases did just that (e.g., Ma et al., 2015; Grilli et al., 2017; Zhang et al., 2021a,b; Grilli et al., 2019a,b,2021; Schambach et al., 2021). Besides, the authors here only show a 2D lab case with reasonable results, where only the first wave is accurately predicted, and a single modeling of a field event where they do not capture inundations that well (the only telltale of the quality of their wave simulations since no other wave measurements were made), and infer that their model performs well across scales.

The authors state they have: "Unique and complete field data, along with the limited geographic extent of lake Askja, reminiscent of a large scale laboratory, are instrumental for the rigorous validation." This event although a good case study is far from unique. Other well-documented case studies involving subaerial and submarine granular slides have been recently successfully modeled. For AK 2018, see Hunt et al. (2021) and new successful modeling by Grilli et al. (2021) and Zhang (2020) based on it of both landslide and tsunami generation and impact (also presented at AGU; Grilli et al., 2019b; Tappin et al. 2019; Watt et al., 2019). See Palu 2018 (Schambach et al., 2021). See 2015 Taan Fiord Alaska (Higman et al., 2018).

The CVV case is discussed but the seminal paper on volcanic collapse landslide tsunami by Ward and Day (2001) should be mentioned. Besides Abadie et al. (2020) which revisits Abadie et al. (2012) with a modified geometry, Tehranirad et al. (2015) shows the CVV wave propagation ? 3D-NS modeling by Horrillo et al. (2013) and subsequent in Gulf of Mexico and Puerto Rico work is also missing. As well as the whole suite of work by Ward et al., starting with tsunami Ball model (Ward and Day, 2001) to tsunami square (e.g., Wang et al., 2019).

Laboratory work for landslide tsunamis, solid or granular, is also insufficiently reviewed and why not model a 3D laboratory case here, for granular subaerial slides as there are quite a few which are a better test of the present model, as others have done (e.g., Ma et al., 2015; Zhang et al., 2021a,b).

Regarding effects of rheology and material properties on tsunami generation, the citations are also incomplete and the authors do not mention studies with "simplified rheologies" that well match both laboratory experiments and full scale events (e.g., Ma et al., 2015; Grilli et al., 2017, 2019a,b, 2021; Schambach et al., 2021; Zhang et al., 2021a,b; and many others).

The main reference to indicate a "paradigm shift is needed" is a recent conference presentation. The authors mention without support: "key issue has been the inability to explain both the landslide dynamics and the tsunami with a unified model and parametrization." Not quite true in many recent work regarding accurately generating and propagating waves. See some references listed above. This statement is much too strong and while one can always improve models (rheology etc...), this reviewer does not believe the statement truly reflects published papers and the state of the work in the field. This would need more support in the paper.

The authors propose a "significant leap forward" but the type of rheology used here is not new and may not be needed to explain wave generation in many large full scale events (a sensitivity study would show this better).

As mentioned, Hunt et al (2021) and Watt et al (2019), for instance show that AK's collapse had many blocks as large as hundredth of meter across and hence is far from a continuous porous material. The granular and viscous rheologies used in successful models (e.g., Grilli et al. 2019a,b, 2021; Zang, 2020) well explain all near-field and far-field tsunami observations at AK, as well as landslide deposits. Models based on particle collisions such as tsunami-square also performed rather well.

The unknown level of saturation in actual collapses makes it doubtful a continuous saturated porous medium would work better than simpler models. More important is the actual geometry and vertical accelerations in the slide.

How can the authors conclude based on a single simulation of Lake Askja, how predictive their model would be for other events such as AK? Other successful work on this and other events, not cited by the authors, using a state of the art landslide tsunami model, did not require making ad-hoc assumptions or iterating on parameters as they suggest. Granted some of the other successful landslide tsunami models do not include dilatancy effects, which may be important to model the landslide in its underwater phase, but based on published results, may not be important in many cases to predict the main tsunami characteristics. Zengaffinen et al. (2020) modeled AK using advanced porous media rheologies and comparing to the simpler rheology used by Grilli et al. (2019) concluded that the latter was sufficient to reproduce the observed tsunami features.

For Lake Askja, authors do not detail the event in introduction? No background is provided and this reviewer has unsuccessfully looked for a single georeferenced map with bathy/topo contour in the paper; and also one indicating where this lake might be. In tsunami propagation and coastal impact modeling, the main source of information is the bathy-topo map.

L76: of constitutes -> of constituents?

The model used has many parameters (Table 1) that need to be measured or estimated/selected with a realistic value? How do they know those for a future event? Some of the many parameters in Table 1 seem arbitrarily selected without justification? No sensitivity study was made by invoking the great computational cost. This significantly weakens the present results.

In Figs. 3 and 4, results of the model are OK but not particularly striking. Essentially the first wave is well captured which is mostly due to inertial entry of the material into the water. Later phases underwater are much less tsunamigenic. Other models with simpler physics would do as well on glass bead experiments (see Grilli et al., 2017; Zhang et al., 2021a,b). Fig. 5 could have bathy/topo contours on it. Show field results for landslide first before modeling. Hard to understand Fig. 5 out of context of model.

Authors write: "To this end, we note that other recent modelling studies could match parts of the wave but not the entire event chain and not with a single set of parameters." Be specific please. Which other studies are you referring to? There are many recent modeling studies where both the landslide and the waves are well modeled for similar lab benchmarks. And also for more

demanding cases in 3D with a higher entry velocity of the slide (e.g., Zhang et al. 2021b).

As expected in most NS models, the authors observe numerical diffusion and, as waves propagate the accuracy is less. This would be a problem in using this model for actual tsunami propagation and hazard assessment where wave elevations are important.

The authors note: "Constitutive parameters for the rock slide are highly uncertain but can be estimated from observations and comparable materials". If this is not model parameter fitting ?

In the lake, the main tsunami generation has occurred for $t < 45s$, when the slide has just penetrated the water. Hence details of the slide underwater beyond that may not be tsunamigenic. However, one needs bathymetry to understand this wave generation. Fig. 7 should have labels on axes. Fig. 7c shows a simple wavemaker motion would work likely well here meaning a simpler rheology would likely work well.

Missing from Fig. 8, to have a truly complete multi-model comparison would be results of a two-layer non-hydrostatic model such as NHWAVE (open source on github and used by many groups). In discussion of slide acceleration, it would be interesting to compute and show the location vel and accel of the slide center of mass. Again having bathy-topo contours on the figure would be helpful.

Slide runout is shown on Fig. 8, but is there data on deposit thickness that could be compared to model results ? Fig. 8 should have different line color/type for lake outline and maximum observed inundation for better readability.

The authors state: "The documented inundation is matched well by the numerical model across the periphery of the lake, as shown in Fig. 8.", but in this reviewer's opinion, runup does not appear to be particularly good in the model results, for a number of areas along the coast ? Is the model adequate for runup (the authors indicate a 1.25 m accuracy due to discretization but this is not really a moving shoreline algorithm) ? Other recent works on nearfield runup around AK showed better agreement of model results and field data. Clearly coupling to a better nearshore model all around the lake could help here.

The authors state: "We conclude from this observation that the wave generation is represented well by the model but that the wave propagation involves some degree of artificial damping." This is true of many NS models that are too numerically dissipative. A sensitivity analysis to mesh and other model parameters might have helped elucidate this, but they claim it is too costly. Also using a two-layer non-hydro model such as NHWAVE to compare to would have been interesting.

Where is the comparison of the slide runout in model and field measurements ? What do you mean by "orographic right branch" ?

The authors claim: "Many reasons for uncertainty". All the more important to perform a sensitivity analysis to assumed model parameters such as packing and initial stresses ? This could have perhaps been done on an idealized 2D slice-slide.

The authors claim: "we anticipate that a landslide with a higher friction angle would have reached the same run-out" Why not trying it out ?

Because of the lack of a sensitivity analysis, many conclusions are based on conjectures about what the model could do.

The authors note: "These results render it unlikely that the deposit is a reliable indicator for the tsunamigenic potential." This reviewer agrees, which likely justifies that using a simpler model of the granular flow would be sufficient to predict the main tsunami features as confirmed in many studies (not cited here).

L190: manifold -> many ?

L213: the authors think the answer is in ever more complex models, but only a sensitivity analysis and complete comparison to other model classes will show which part of the physical processes matters most for tsunami generation. This reviewer believes this will likely not be the details of the flow within the porous medium.

Authors also do not discuss the exact nature of the debris flow in the field event and whether assuming it was made of uniformly distributed 10 cm particles was realistic? Likely not. Was there a sub-bottom survey of the deposits made? Instead the authors wonder about the role played by air expulsion, which is likely negligible for slide dynamics and wave generation. The authors are trying to find guidance for this from their extremely small scale lab experiments where many processes do not scale correctly (such as air bubbles, surface tension, viscosity).

The authors state: "Our model performs very well compared with the depth-integrated models. This is remarkable, because the three-dimensional model was not optimized to fit the observations" Nothing quite remarkable here, as this was an oversimplified depth-integrated model for a sliding block! It was expected that the BM would not perform well. The BM model performs better in the far-field which confirms there is likely too much dissipation in the 3D model. The authors agree with that on L 243. So, the model can't be trusted at some distance from the source which is not quite known! Not that useful a feature in tsunami hazard assessment studies...

The claim that the present resolution is better than in some other studies is not a justification in itself as it all depends on model type, physics, and numerical methods. What is missed in runup and inundation is much greater than the cell size. Is the shoreline algorithm good enough in the model?

L:229,235,249: "This is remarkable, because", "the inundation allows remarkable conclusions on the generation event": " and that the results are of remarkable quality": there are really too many, not quite appropriate, superlatives in this paper.

L 239: The excuse of high-computational cost for not doing any parameter or mesh sensitivity studies is really not a good excuse when one claims to have a new breakthrough in modeling!

L 254: "and the application of a sub grid turbulence model seems appropriate" Do they authors mean they claim to solve NS equations at that scale with a 2.5 m mesh and no turbulence closure modeling? It seems in Table 1, they have multiplied without real justification the standard water viscosity in both the scale model and field problems by about 15! The small scale lab modeling may have sufficient viscosity owing to its low Re value, but for the field scale event, unless a turbulence model is used, it is likely that simulations are not sufficiently dissipative around the wave generation area and the model solves a version of Euler equations. No discussion of this is made.

L257: "The lake Askja event is challenging for a three-dimensional landslide tsunami model because the wave amplitude is very shallow in relation to the lake size". I can't really understand the meaning of comparing the maximum surface elevation to lake lengths. typical parameters in wave models are nonlinearity and dispersions that combine surface elevation with depth and wavelength. Depth is never mentioned nor discussed at any point before? Three-dimensional models have been used in many similar contexts. I suspect that a 2.5 m cell might not be necessary for a correct tsunami modeling with a 3D non-hydrostatic model in this case but maybe 10-20 m with depth layers of various thickness. Of course those models are not even mentioned or considered here.

Model coupling is of course a way to deal with these issues and in addition to the Lovholt et al. (2008) paper, they could refer to a suite of earlier and later works that have perfected the use of model-coupling of 3D/2D-tsunami propagation for hypothetical or actual events. Ignoring a lot of the work done outside of their collaborator group is strange. Abadie et al., (2012) is cited as a "promising approach". Well this was 10 years ago work that has had many improvement, in particular regarding the landslide tsunami models (e.g., Tappin et al., 2014; Grilli et al., 2015,2017; Schambach et al., 2019, 2020, 2021)

L 265: "This work represents a large step forward in modelling of subaerial landslide tsunamis."
"demonstrates remarkable accuracy of wave kinematics and landslide runout"

In this first statement in the conclusion, the authors quite pleased with their work summarize the flavor of a paper that this reviewer believes is not sufficiently self-critical. This conclusion should be demonstrated by results and made by readers, but results presented here are not particularly impressive (considering a model that needs to run for 100s of hrs on 100s of processors) and do not demonstrate in this reviewer's opinion a step forward in subaerial tsunami modeling. There are a few other existing models with just some aspects of the physics simplified, as compared to the present model, that would very likely perform as well on similar lab experiments or even better on field case studies of landslide tsunamis as they would not damp waves in this manner in the far-field.

L267: "without the need for optimized parameters" Authors seem to imply all other modelers optimize their parameters to match observation ? It is not the case in references cited above, whereas the present model has quite a few parameters set to arbitrary values.
"However, the simulations are computationally expensive and wave damping can be observed in the far field due to the limited grid resolution." What is the point in using a model one cannot afford which, besides, overdamps the wave in an unknown grid-dependent manner ? There are other published and well-validated models with just slightly simpler physics that are very efficient and can be and have been used for meaningful hazard assessment cases.

L263: "Full three-dimensional methods are relatively rare in the tsunami community". These have been used whenever necessary, but not over-used when not necessary. This is where the art of the modelers comes in to decide about the best class of models/physics to apply to a given problem. In any cases these aspects have not been exhaustively reviewed and discussed in this paper.

L316: "sub grid turbulence that might play a role at high Reynolds numbers" The field case in the lake does have a large Reynolds number !

To conclude, this reviewer finds the presented work to have merit and interest in the context of landslide tsunami modeling. With substantial revisions, it could likely be published in a relevant engineering or geophysics journal. However, the reviewer believes that the work does not raise to the level and general audience interest of a Nature Communication paper and, considering the concerns expressed above, does not believe it can be revised in a suitable manner.

Regarding level of detail to reproduce the work, there are many missing information and data and no electronic repository is provided. The authors asked to be contacted for information.

References

Abadie, S., J.C. Harris, S.T. Grilli and R. Fabre 2012. Numerical modeling of tsunami waves generated by the flank collapse of the Cumbre Vieja Volcano (La Palma, Canary Islands) : tsunami source and near field effects. *J. Geophys. Res.*, 117, C05030, doi:10.1029/2011JC007646

Grilli S.T., O'Reilly C., Harris J.C., Tajalli-Bakhsh T., Tehranirad B., Banihashemi S., Kirby J.T., Baxter C.D.P., Eggeling T., Ma G. and F. Shi 2015. Modeling of SMF tsunami hazard along the upper US East Coast: Detailed impact around Ocean City, MD. *Natural Hazards*, 76(2), 705-746, doi: 10.1007/s11069-014-1522-8

Grilli, S.T., Shelby, M., Kimmoun, O., Dupont, G., Nicolsky, D., Ma, G., Kirby, J. and F. Shi 2017. Modeling coastal tsunami hazard from submarine mass failures: effect of slide rheology, experimental validation, and case studies off the US East coast. *Natural Hazards*, 86(1), 353-391.

Grilli S.T., D.R. Tappin, S. Carey, S.F.L. Watt, S.N. Ward, A.R. Grilli, S.L. Engwell, C. Zhang, J.T. Kirby, L. Schambach and M. Muin 2019a. Modelling of the tsunami from the December 22, 2018 lateral collapse of Anak Krakatau volcano in the Sunda Straits, Indonesia, *Scientific Reports*, 9, 11946 (open access) doi:10.1038/s41598-019-48327-6.

Grilli S.T., Schambach L.C., Zhang C., Kirby J.T., Grilli A.R., Tappin D., Carey S., Watts S., Day S., Engwell S., Ward S. and M. Muin 2019b. Modeling of the slide and tsunami generation from the 12/22/18 lateral collapse of Anak Krakatau volcano (Sunda Straits, Indonesia): comparison with recent field surveys of slide deposits and tsunami impact. In AGU Fall Meeting Abstract, NH32A-05.

Grilli, S.T., Zhang, C., Kirby, J.T., Grilli, A.R., Tappin, D.R., Watt, S.F.L., Hunt, J.E., Novellino, A., Engwell, S.L., Nurshal, M.E., Abdurrachman, M., Cassidy, M., Madden-Nadeau A.L. and S. Day 2021. Modeling of the Dec. 22nd 2018 Anak Krakatau volcano lateral collapse and tsunami based on recent field surveys: comparison with observed tsunami impact. *Marine Geology* (submitted).

Higman, B., Shugar, D.H., Stark, C.P., Ekström, G., Koppes, M.N., Lynett, P., Dufresne, A., Haeussler, P.J., Geertsema, M., Gulick, S. and Mattox, A., 2018. The 2015 landslide and tsunami in Taan Fiord, Alaska. *Scientific Reports*, 8(1), pp.1-12.

Horrillo, J., Wood, A., Kim, G.-B., Parambath, A., 2013. A simplified 3-d Navier-Stokes numerical model for landslide-tsunami: Application to the gulf of Mexico. *J. Geophys. Res.: Oceans*, 118, 6934–6950. <http://dx.doi.org/10.1002/2012JC008689>.

Hunt, J.E., Tappin, D.R., Watt, S.F.L., Susilohadi, S., Novellino, A., Ebmeier, S.K., Cassidy, M., Engwell, S.L., Grilli, S.T., Hanif, M., Priyanto, W.S., Clare, M.A., Abdurrachman, M., and U., Udrekh 2021. Submarine observations show half of the island of Anak Krakatau failed on December 22nd 2018. *Nature Communications* (in press).

Ma, G., Kirby, J.T., Shi, F., 2013. Numerical simulation of tsunami waves generated by deformable submarine landslides. *Ocean Model.*, 69, 146–165. <http://dx.doi.org/10.1016/j.ocemod.2013.07.001>.

Ma, G., Kirby, J.T., Hsu, T.-J., Shi, F., 2015. A two-layer granular landslide model for tsunami wave generation: Theory and computation. *Ocean Model.*, 93, 40–55. <http://dx.doi.org/10.1016/j.ocemod.2015.07.012>.

Schambach L., Grilli S.T., Kirby J.T. and F. Shi 2019. Landslide tsunami hazard along the upper US East Coast: effects of slide rheology, bottom friction, and frequency dispersion. *Pure and Applied Geophys.*, 176(7), 3,059-3,098,doi.org/10.1007/s00024-018-1978-7

Schambach L., Grilli S.T., Kirby J.T. and F. Shi 2019. Landslide tsunami hazard along the upper US East Coast: effects of slide rheology, bottom friction, and frequency dispersion. *Pure and Applied Geophys.*, 176(7), 3,059-3,098,doi.org/10.1007/s00024-018-1978-7

Schambach L., Grilli S.T., Tappin D.R., Gangemi M.D., and G. Barbaro 2020. New simulations and understanding of the 1908 Messina tsunami for a dual seismic and deep submarine mass failure source, *Marine Geology*, 421, 106093

Schambach L., Grilli S.T. and D.R. Tappin 2021. New high-resolution modeling of the 2018 Palu tsunami, based on supershear earthquake mechanisms and mapped coastal landslides, supports a dual source. *Frontiers in Earth Sciences*, 8, 627, doi:10.3389/feart.2020.598839

Tappin D.R., Grilli S.T., Harris J.C., Geller R.J., Masterlark T., Kirby J.T., F. Shi, G. Ma, K.K.S. Thingbaijamg, and P.M. Maig 2014. Did a submarine landslide contribute to the 2011 Tohoku tsunami ?, *Marine Geology*, 357, 344-361 doi: 10.1016/j.margeo.2014.09.043

Tappin D., Hunt J., Grilli S.T., Watts S., Day S.J., Grilli A.R., Engwell S. Carey S. and S. Susilohadi 2019. The 1883 and 2018 Krakatau tsunamis -- new marine evidence on their generation. In AGU Fall Meeting Abstract, NH32A-04.

Tehranirad B., Harris J.C., Grilli A.R., Grilli S.T., Abadie S., Kirby J.T. and F. Shi 2015. Far-field tsunami impact in the north Atlantic basin from large scale flank collapses of the Cumbre Vieja volcano, La Palma. *Pure and Applied Geophysics*, 172(12), 3,589-3,616 doi:10.1007/s00024-015-1135-5

Ward, S. N., and S. Day (2001), Cumbre Vieja Volcano—Potential collapse and tsunami at La Palma, Canary Islands, *Geophys. Res. Lett.*, 28, 3397–3400, doi:10.1029/2001GL013110.

Watt S., Cassidy M., Engwell S., Madden-Nadeau A., Abdurrachman M., Tappin D., Grilli S.T., Day S., Carey S., Hunt J., Hayer C., Burton M. and A. Novellino 2019. Evaluating the role of eruptive processes in the source of the 2018 Anak Krakatau tsunami. In AGU Fall Meeting Abstract, NH31A-02..

Wang, J., Ward, S.N. and Xiao, L., 2019. Tsunami Squares modeling of landslide generated impulsive waves and its application to the 1792 Unzen-Mayuyama mega-slide in Japan. *Engineering Geology*, 256, 121-137.

Zengaffinen, T., Løvholt, F., Pedersen, G. K., & Muhari, A. 2020. Modelling 2018 Anak Krakatoa Flank Collapse and Tsunami: Effect of Landslide Failure Mechanism and Dynamics on Tsunami Generation. *Pure and Applied Geophysics*, 177(6), 2493-2516.

Zhang C. 2020. A two-layer non-hydrostatic landslide model for tsunami generation on irregular bathymetry. PhD Dissertation, University of Delaware, 173 pps.

Zhang C., Kirby J., Shi F., Ma G. and S.T. Grilli 2021a. A two-layer non-hydrostatic landslide model for tsunami generation on irregular bathymetry. 1. Theoretical basis. *Ocean Modelling*, 101749, doi:10.1016/j.ocemod.2020.101749 .

Zhang C., Kirby J., Shi F., Ma G. and S.T. Grilli 2021b. A two-layer non-hydrostatic landslide model for tsunami generation on irregular bathymetry. 2. Numerical discretization and model validation. *Ocean Modelling*, 101769, doi:10.1016/j.ocemod.2021.101769 .

Reviewer #3:

Remarks to the Author:

Summary:

The Authors investigate subaerial landslide tsunamis with a (partially) newly implemented model based on the tri-phase Navier-Stokes equations in the OpenFOAM environment. They model successfully the small-scale granular cases of Viroulet et al. (2013) in 2D, followed by a full 3D-simulation of the 2014 Lake Askja case with good agreement. They further compare their run-up results with past results based on simplified slide and hydrodynamic models.

Evaluation:

I would like to congratulate the Authors on this excellent study. The topic and study match well into Nature Communications with similar contributions published in the past. The Authors combine state-of-the-art knowhow in granular dynamics and hydrodynamics and implement a new model in the open-source environment OpenFOAM. This model includes a low level of “modelling” parameters, models the full process (slide kinematics, hydrodynamics) and a very good agreement with the laboratory data and a good agreement with the field data is achieved as well, without parameter optimisations and one set of parameters for both the laboratory and nature event. This is indeed an excellent and useful result; landslide-tsunami models often achieve good agreement, but only after significant parameter fitting and optimisations based for example on laboratory experiments such that they may not be applied to more complex real or other laboratory cases with confidence. In past studies also often the slide model is prescribed or not fully realistic (e.g. modelled as a rigid block).

Given the substantial required computational resources (e.g. 450 h and 220 cores required to run the 2014 Lake Askja case, L438), no convergence tests have been presented (a basic requirement for numerical simulations on peer-review level) and some compromises were necessary for the wave propagation (too low resolution to avoid numerical diffusion (L243)). These limitations are clearly communicated in the article. As recognised by the Authors, simulation time could be saved by coupling the model with e.g. a non-linear non-hydrostatic shallow-water equation model

(considering frequency dispersion) which is fully appropriate to model wave propagation and inundation, as already demonstrated for the far field of the 2014 Lake Askja case, once only the water phase is left (e.g. SWASH, Zijlema et al. 2011, there are probable similar implementations in OpenFOAM available as well, which would still enable the use of one model environment only for the full process). In other words, the computational expensive full Navier-Stokes model is not required in the far field and at the same time the described limitations of the numerical diffusion due to a too small resolution would be overcome by using e.g. SWASH. It is also made clear in the manuscript that the laboratory experiments have been simulated in 2D. In my view, it needs more justification on whether boundary layer effects in the granular material would not affect the slide kinematics (a 3D simulation of at least one of the laboratory cases would be desirable to shed light on this point). The underlying laboratory experiments are small and will be affected by surface tension scale effects, which cannot be modelled (L132, L314) and in addition grain Reynolds number effects (e.g. affecting slide run-out) appear not to be captured as well.

In general, I would wish to see more quantitative measures to judge a goodness of fit rather than qualitative statements such as “a very good agreement” has been achieved. There is actually a method (combining Discrete Element Modelling DEM with Computational Fluid Dynamics CFD) implemented in an open-source environment (e.g. CFDEM), which appears to involve even less modelling and the new method should be put into the context of this method. I also understand that the Lake Askja case simulation relies on past event field data, such as the deposit (to define an appropriate friction parameter) or the initial slide shape based on the topography obtained after the event. In other words, more work will be required to also reliably predict future events (which should be the long-term goal), in addition to the modelling of a past event. However, the achievements of the Authors are still a step forward.

These weaknesses together with further less major points such as some unclear statements, typos and inconsistencies are highlighted under the following Detailed points on the suggested improvements.

Detailed points on the suggested improvements:

L13: Please replace the sentence “a gap in providing unbiased predictions of real events.” with a more specific statement. Again on L48, the word biased can have many meanings, which one do the Authors mean?

L15: “For the first time, we show a model that scales consistently from small scale laboratory experiments to full scale catastrophic events.” I find this statement opens up some questions. It is well-known that the physics of the described phenomena change with the scale due to scale effects (e.g. Kessler et al. 2020). E.g. Grain Reynolds number effect becomes more dominant at small scale laboratory experiments and e.g. affects the slide run-out, and as such tsunami generation. Also the water domain is affected by significant scale effects, given that the underlying experiments of Viroulet et al. (2013) have been conducted at a small water depth 0.15 m and the model does exclude surface tension (L132). Parts of this is later communicated in the article, but my immediate thought at this location in the manuscript would be that a consistent scaling is not necessarily something positive, as the physics changes at small scales due to scale effects.

L16: The statement that the model accurately captures the inundation is not fully reflected in Fig. 8b where locally deviations of up to 50% are shown, e.g. the measurements show 40 m and the simulation nearly 60 m. Please communicate this more carefully.

L31: What are “impulsive landslide tsunamis”? The term is not common and nearly every tsunami is generated by an impulse.

L36/L52/L281: Would a Discrete Element Modelling DEM coupled with Computational Fluid Dynamics not involve less modelling than in the proposed model? There are open-source codes available to do so, e.g. CFDEM coupling (CFDEM@coupling Documentation — CFDEMcoupling v3.X documentation) and there are already publications into subaerial-landslide tsunamis available based on CDEM-CFD coupling (e.g. Shan et al. 2014). The work should be put into the context of such methods.

L42: "and allows mixing of the granular material with fluids that represent the tsunami." I assume fluids stands for air and water, but the former is not representing the tsunami.

L117: Which recent modelling studies could not match part of the waves or are based on different parameter sets? These studies should really be mentioned for the Reader to be able to verify this statement.

L136: "The simulated wave (Fig. 4) matches the measurements from the experiment very well." This should be backed up with a quantitative analysis, e.g. a normalised Root Mean Square Error with respect of the maximum wave amplitude or the full wave profile, or similar.

L152: "...we can avoid complex boundary conditions and eventual artificial wave reflections." Please clarify what artificial wave reflections is.

Fig. 5: Please make the time indication better visible.

Caption Fig. 8: "The lake outline at rest" remains unclear, should it read "The lake is outline at rest"? Also in (b) I would write "Measurements" in plural form.

Fig. 8 and L218: Also here, a quantitative, not only qualitative ("modelled well"), comparison is necessary. Further, please make clear that the Boussinesq and Shallow-water results are taken from (4) and that it is based on a different slide model.

L200: "Scenarios with similar runout but e.g. different impact velocities will differ strongly in their tsunamigenic potential, which highlights once more the value of consistent models that provide the whole history." There are studies which looked into the effect of the underwater slide kinematics on the wave in depth already, this was often modelled/considered with the "characteristic time of submerged landslide motion" (see Walder et al. 2013; Heller and Spinneken, 2013).

L201: "A definitive conclusion" on what, there are several aspects mentioned?

L222: Where would this "artificial damping" originate from? I can see that this is discussed on L242, but the Reader will have this question here.

L228: "very well" needs a qualitative measure of the goodness of fit.

L235: The statement "...indicating that the inundation allows remarkable conclusions on the generation event." remains unclear, please try to be more specific.

L271: A reference to the section where the computational resources are addressed would help the Reader.

L315: It is not only surface tension which affects small-scale models (and results in scale effects), but also the grain Reynold number (see e.g. Kessler et al. 2020). As far as I understand, these effects are also not modelled appropriately by modelling the granular phase with a rheology (in contrast to a DEM coupled with CFD).

L316: High Reynolds number of what, of the slide or water?

L329: Please specify whether these are vectors or parameters, e.g. for g.

Eqs. (6) and (7), are there not some parameters required a priori, such as the granular viscosity and the friction coefficients? Also, the grain diameter $d = 0.01$ m in the real case (L398) is a choice which may affect the results?

L401: It is stated "The quasi-static friction coefficient ... was chosen to match the overall slope gradient from release to deposition". This also appears to be a parameter which is not known a priori of an event? This seems to contradict the statement on L412 "Note that the chosen parameters were not fitted by comparing the computational results with the observations." and to

some degree the overall conclusion that there is no need for "optimized parameters" on L267.

L416: "terrain before the event in the deposition area and the terrain after the event" also here information not available prior to the event are required.

L519: The access data 20.01.2020 is rather a long time ago. Please check and update this date if possible.

Typos and minor issues: L17: Please write "Lake Askja" rather than "lake Askja" (in general, the writing style is not consistent in the manuscript (lake versus Lake)). Caption Figure 1: Write "from" rather than "form". Fig. 2: Write "saturated" rather than "sat.". There is also an unnecessary abbreviation used in Table 1. Fig. 4: Add free spaces between the parameter and the units on all axes. Caption Fig. 6: Write "in Figure 5" rather than "in the figure before". Caption Fig. 7: Add "Lake" in front of "Askja" and write "black" rather than "back". L451: There are issues. L467: Write "Italian" rather than "italian". L102: Remove the full stop. L106: Write "signals" rather than "signal". L113: Add "the" ahead of "landslide". L202: Add a comma after "however". L308: Add "a" in front of "part" and "the" in front of "home". L462: Add a comma after the first author. L486: Write OpenFOAM rather than openfoam. L487: Write the name Lovholt correctly; L510: The capital letters in the article title are not consistent with other titles. L565: Please use capital letters for the Journal name.

General comment: The Authors use American (e.g. L96 "visualized") rather than British English (visualised). Nature Communications may prefer the latter.

Movie supplement: Please write "Lake Askja" rather than "Askja" only (5 times).

Supplementary material:

Caption Fig. 2: Please drop one of the "cases". Also replace rho with the symbol.

Captions Figs. 3 and 7: Please write "Lake Askja" rather than "lake Askja" or "Askja".

Caption Figs. 8 to 11: Please add free spaces between the parameters and the units on all axes. The text in the figures is too larger compared to the main text (e.g. degrease the figure size).

2nd paragraph under section 3: Please write listing rather than lst. (several times). There is not reason to abbreviate this word.

References:

- Heller, V., Spinneken, J. (2013). Improved landslide-tsunami predictions: effects of block model parameters and slide model. *Journal of Geophysical Research-Ocean* 118(3):1489-1507
- Kessler, M., Heller, V., Turnbull, B. (2020). Grain Reynolds number scale effects in dry granular slides. *Journal of Geophysical Research-Earth Surface* 125(1):1-19.
- Shan, T., Zhao, J. (2014) A coupled CFD-DEM analysis of granular flow impacting on a water reservoir. *Acta Mechanica* 225:2449-2470.
- Walder, J.S., Watts, P., Sorensen, O.E., Janssen, K., (2003). Tsunamis generated by subaerial mass flows. *Journal of Geophysical Research* 108(B5):2236(2).
- Zijlema, M., Stelling, G., Smit, P. (2011). SWASH: an operational public domain code for simulating wave fields and rapidly varied flows in coastal waters. *Coastal Engineering* 58 (10):992-1012.

Answers to reviewer 1 on "Porous granular flow models can predict full-scale landslide tsunamis"

The original authors

October 10, 2021

The review is shown in blue, answers in black.

The manuscript by Rauter and coauthors is about the application of porous granular flow models for better understanding of landslide-triggered tsunami. Tsunami caused by landslides are much more common than previously thought, a number of examples is correctly provided in the manuscript. In the year 2018 alone in Indonesia landslides caused the deadly Palu tsunami, a landslide from Krakatau volcano affected coastal regions of Sumatra and Java, and a collapse of a littoral dome on Kadovar Island led to small tsunami at Papua New Guinea. The manuscript submitted to Nature Communication nicely describes the difficulty and lack of complexity in former tsunami models, where the tsunami source might be considered by granular medium.

I found the manuscript very interesting to read, the organization is mostly logic (see comments below), the figures are well prepared and of high quality, and the supplementary materials, especially video simulations, make the outcomes of the work enjoyable for a broad readership. A pleasure to review such a high quality piece of work, thank you! I therefore can recommend publication of the manuscript after considering the minor changes suggested as follows.

Thank you very much for your kind comments and the review. We will give detailed answers in the following.

1) Reconsider the title. Is it about prediction of tsunami? Would mentioning of Askja be valuable here?

Thank you very much. We agree that the title is a bit cumbersome. We changed the title to "Granular porous landslide tsunami modelling - the 2014 Lake Askja flank collapse".

2) Abstract: Tone down the last sentence of the abstract. Otherwise the abstract is almost perfect, it summarizes the technical advances and Askja application. Could clarify that tsunami propagation physics is well understood, but the initiation and source region is not.

We changed and toned down the last sentence of the abstract.

3) In parts the manuscript rather reads like a technical report instead of a nature / geologic work aiding process understanding. I would recommend to highlight the application more and move technical details to accompanying documents.

Thank you very much for this comment. We moved many technical details, especially of simulation setups to the supplementary materials or removed them completely.

4) Of course tsunami are oceanic reach (line 28), i do not understand what authors want to tell me with this?

Not all tsunamis have oceanic reach, especially landslide tsunamis are often diffusing rapidly with distance and thus have only limited/local reach. In fact, landslide tsunamis are often seen as having only local effects. Their reach and importance has thus been underestimated for a long time. Therefore, we want to emphasize that these events can have oceanic reach in the manuscript.

5) I would prefer the term landslide-triggered tsunami, instead of landslide tsunami.

The term landslide tsunami is the common term in the literature. We prefer this term for brevity, as it is repeated very often.

6) L52: why a landslide should be granular only? Landslides elsewhere develop into blocks and granular flows. Also submarine studies show landslides partially move as blocks and partially in granular fashion. This would need a more careful consideration and conceptual explanation/justification.

We are aware that slides may move as blocks but this behaviour is nevertheless granular. The granules are in such a case hold together by cohesion or apparent cohesion. True cohesion is usually small and neglected in our work. Apparent cohesion arises from the combination of (negative) excess pore pressure and inter-granular friction (think of a sealed pack of coffee that appears hard). Negative pore pressure decays quickly in permeable soil and slides (e.g. sandy, rocky slides as the permeability scales with particle diameter, see manuscript) and we get a frictional behaviour, where the shear strength depends on the overburden pressure. This is the classic granular slide.

In fine grained slides (e.g. clay slides) the permeability is very low and the excess pore

Figure 1: Interactions in the landslide model.

pressure cannot decay. This uncouples the effective pressure from overburden pressure and one observes a nearly constant shear strength, i.e. apparent cohesion and slides can move in blocks. Although this behaviour is not termed granular in some communities and works, the vast majority of the geotechnics and granular flow community would also consider this a granular slide.

Excess pore pressure and the grain size dependent permeability is represented in our model, see Fig. 1. The packing density ϕ_g determines the effective pressure p_s . In turn, the effective pressure is pushing grains apart, which leads to a balance of effective pressure and overburden pressure over time. The drag \mathbf{m}_{gc} (the inverse of the permeability) is able to delay this process. The shear strength $\mathbf{T}_g^{\text{dev}}$ is determined by the effective pressure.

Frictional, see Fig. 2 left, (the classic granular slide) and plastic behaviour (block slides), see Fig. 2 right, are limit cases of vanishing permeability and vanishing drag and these cases are represented in the model as well. This is covered in detail in our previous publication Rauter (2021), which is cited in our manuscript.

7) L70 and following. Here I would suggest some changes, as the methods are appearing too prominent without even introducing the problem. I would suggest to move L71-82 to the methods section, and chapters 2.1, 2.2, 2.3, and 2.4 to the appendix. The reason for that is, that in the current version the Askja landslide is not even introduced before page 7, and the manuscript rather appears like a technical report than a nature-related study. Also fig 2 could be moved accordingly.

Regarding L71-82: We shortened this part as much as possible and added it to the

Figure 2: Limit states of the model leading to plastic and frictional behaviour.

last paragraph of the introduction. We still introduce the model (therefore we think it fits well into the introduction) as far as it is required to understand the presented results. We agree that the introduction of parameters and OpenFOAM is not useful at this point and moved/removed it. Figure 2 shows the reader how to read/interpret the result figures and we think it is required upfront. Removing it at this point lead to questions in the later parts of the paper.

Regarding chapters 2.1 to 2.4: The laboratory simulation takes a very important role in this work. The combination of the very accurate laboratory case and the still accurate real case makes this work in our eyes special. This is also highlighted in the abstract and the introduction. Therefore we want to keep chapters 2.1 to 2.4 in the result section.

8) The result calibration to analog experiments is helpful, but could be moved to appendix/supplementary materials.

This is less of a calibration but more of a validation of the newly introduced method and very important for the paper from a modellers perspective, and as said above, an important step forward in its own right. This example shows that we can predict the wave of small scale events based on the granular material parameters which can be measured independently of the event. This capability is rather uncommon in the tsunami community and one of the major distinguishing points of our manuscript/model. Many works focus on the small case laboratory example only (e.g. Si et al., 2018; Clous and Abadie, 2019), others on the real case only (e.g. Abadie et al., 2020; Franco et al., 2019). We think that the paper is especially interesting as we can do both scales, lab and real case with the same model. Therefore we would like to keep it prominently in the paper.

9) The Askja part starts after L 142 on p7. Please add here missing references, such as those that document the inundation or those that map terrain and bathymetry before and after the event.

All the data is from Gylfadóttir et al. (2017), which is referenced.

10) L155. I might have overlooked it in this chapter, but the dense core assumption is based on what? Which earlier study?

Dense granular flows usually have a packing density between 0.6 and 0.4, so 0.5 ± 0.1 (Pouliquen et al., 2006). To have a value for the iso-surface of dense flow, we took half of this average value, i.e. 0.25, similar as in Rauter (2021). The iso-surface should thus be located between the most upper layer of sand ($\phi_g \approx 0.5$) and the grain free domain ($\phi_g = 0$). This does not influence the results and is solely for visualization purposes and therefore we do not discuss it in depth.

11) fig. 5 caption. Which dense core? Please clarify the scale and time in the figure, it is hardly readable on my printout (same for fig. 6). Please specify the shaded relief/dem origin, sampling and year.

The "dense core" is the dense core of the landslide, we corrected this.

We put a white rectangle behind the scale and time step.

The shaded relief/dem is the bottom surface of the simulation domain. This is, as described in the manuscript, "[...] based on a terrain model that neither contains the landslide deposition nor the initial landslide geometry (see supplementary material). It can be seen as a combination of the terrain before the event in the deposition area and the terrain after the event in the release area." Therefore we cannot assign a year and also the sampling was different between the various parts that were merged to form this terrain model. Details can be found in Gylfadóttir et al. (2017).

12) what is known about the initial geometry of the Askja rockslide? Looking at the figure 7 and the very simplified geometry in (a) i am wondering if there is much known about it. How relevant is the basal decollement plane, its listric curvature and complexity? Curved decollement planes desintegrate landslide into few large blocks, such as documented at the MSH rockslide (cf. <https://doi.org/10.1130/G32198.1>).

We know the initial geometry well, due to laser scans of the terrain before and after the event. The geometry shown in 7a is a cut through the three dimensional initial slide geometry, therefore it looks rather simple. What we do not know is the composition

of the failed slope and the geometry of any planes, weak layers, etc. within the failed mass. We ignore any such inner geometrically inhomogeneities and initialize the initial slide geometry as a homogenous mass, except for the packing density, which has to increase with depth to be consistent with the model. Disintegration into blocks, or similarly the formation of shearbands, can be observed in similar simulations (Rauter, 2021) but not in the presented simulations. We found such a failure mechanism unlikely, considering the relatively loose material and the shallow failure surface. Anyway, there are better methods (i.e. Finite Element Method with complex constitute models for soil) to investigate the initial failure. Including inhomogeneities (although possible) would increase the complexity to a very high level. We want to handle this problem as if it was a real event about to happen and designed the work flow accordingly. This will allow us to predict such events in the future, which is the overall goal of this work.

Anyway, we think that there must have been inhomogeneities and weak zones/layers that triggered the event and we put some further thoughts into it. As it looks right now, the uniformly initialized slide flows overly strong towards the orographic (in flow direction) right side, i.e. the north side. This results in an overestimation of the inundation on the northern shore and an underestimation of the inundation on the southern shore. We ran a second simulation with different material parameters and although the intensity of the inundation changed, this pattern stays the same. Therefore, we think that the slope failure was triggered by a weak zone on the southern edge of the failure area that propagated towards north. The flow in reality was respectively stronger on the southern side as the simulation with homogenous failure area would suggest. We added a discussion on this topic to the manuscript.

13) ch2.6. I recommend the authors to include the Viti lake as an important reference for two reasons: first, the lake was inundated and infilled by the tsunami, possibly giving an important height constraint. Second, the lake is a major touristic destination (for north icelandic standards), so that a certain hazard aspect might be demonstrated.

Thank you for this comment. Both, Lake Askja and Lake Viti are popular tourist attractions as mentioned already by (Gylfadóttir et al., 2017). Lake Viti was only marginally affected and mentioning it in the paper would not strongly increase the informative value of our paper. We hope you can understand this decision.

14) L191 - authors mention uncertainties in initialization of the landslide geometry. I recommend adding further detail and broaden the discussion adequately, considering block movements (also Toreva blocks), listric and planar decollement, and other effects.

Thank you very much. We extended the respective discussion. We do not expect decollement in the rather loose and near-surface material. However, there are hints in our investigation that the initial failure occurred on the southern side of the failure area.

We discuss it in the paper now.

15. L215, if already noted before authors can delete repetition.

Done, thank you.

16. L217. What exactly is "acceptable"?

This is a very good question that is equally hard to answer. We meant acceptable from a hazard/mitigation engineering point of view. That means that the general behaviour is correctly described and that points that require the engineers attention are highlighted in the simulation. Further we looked at various error quantifications and describe this issue in detail in the new supplementary materials. The quantified errors (e.g. L2 norm or average error) are remarkably small and within an uncertainty that we are used to deal with. Finally, we also think of acceptable in comparisons to other methods, which do not provide more accurate results. Usually we get such levels of accuracy only after the event happened and the simulation could be optimized (see Gylfadóttir et al., 2017). However, we see that this is weakly communicated (and hardly communicable) and thus changed the respective part.

17. Could results contribute to analyse and predict how deposits may look like on the lake floor? I am thinking also about the Krakatau example, where new result suggests the deposition of a number of large blocks (instead of granular rheology only)

Yes, this is indeed possible with the presented model, at least better than with depth-integrated models. However, this would require a much longer simulation duration, that has to simulate all the flow and all the settlements that occur.

As mentioned before, we can simulate the flow of blocks with the model. This is better visible in the overconsolidated experiment found in the previous publication Rauter (2021). The permeability in the presented cases is small and the flows look more "granular". We could also provoke a more "blocky" flow by adding a cohesion to the material, as done by e.g. Li et al. (2020) for snow avalanches. However, this is out of the scope of the presented paper.

18. The supplements and methods are described very clearly.

Thank you. We further extended the supplementary material following the other referees suggestions.

I hope my comments help to further refine this already excellent study.

Thank you very much for your comments. It is very helpful to get comments from such a different point of view and we appreciated your constructive comments.

References

- S. Abadie, A. Paris, R. Ata, S. Le Roy, G. Arnaud, A. Poupardin, L. Clous, P. Heinrich, J. Harris, R. Pedreros, et al. La Palma landslide tsunami: calibrated wave source and assessment of impact on French territories. *Natural Hazards and Earth System Sciences*, 20(11):3019–3038, 2020. doi: 10.5194/nhess-20-3019-2020.
- L. Clous and S. Abadie. Simulation of energy transfers in waves generated by granular slides. *Landslides*, 16(9):1663–1679, 2019. doi: 10.1007/s10346-019-01180-0.
- A. Franco, J. Moernaut, B. Schneider-Muntau, M. Aufleger, M. Strasser, and B. Gems. Lituya bay 1958 tsunami “ detailed pre-event bathymetry reconstruction and 3d-numerical modelling utilizing the cfd software flow-3d. *Natural Hazards and Earth System Sciences Discussions*, 2019:1–34, 2019. doi: 10.5194/nhess-2019-285. URL <https://www.nat-hazards-earth-syst-sci-discuss.net/nhess-2019-285/>.
- S. S. Gylfadóttir, J. Kim, J. K. Helgason, S. Brynjólfsson, Á. Höskuldsson, T. Jóhannesson, C. B. Harbitz, and F. Løvholt. The 2014 Lake Askja rockslide-induced tsunami: Optimization of numerical tsunami model using observed data. *Journal of Geophysical Research: Oceans*, 122(5):4110–4122, 2017. doi: 10.1002/2016JC012496.
- X. Li, B. Sovilla, C. Jiang, and J. Gaume. The mechanical origin of snow avalanche dynamics and flow regime transitions. *The Cryosphere*, 14(10):3381–3398, 2020.
- O. Pouliquen, C. Cassar, P. Jop, Y. Forterre, and M. Nicolas. Flow of dense granular material: towards simple constitutive laws. *Journal of Statistical Mechanics: Theory and Experiment*, 2006(07):P07020, 2006. doi: 10.1088/1742-5468/2006/07/P07020.
- M. Rauter. The compressible granular collapse in a fluid as a continuum: validity of a Navier-Stokes model with $\mu(J), \phi(J)$ -rheology. *Journal of Fluid Mechanics*, 915, 2021. doi: 10.1017/jfm.2021.107.
- P. Si, H. Shi, and X. Yu. A general numerical model for surface waves generated by granular material intruding into a water body. *Coastal Engineering*, 142:42–51, 2018. doi: 10.1016/j.coastaleng.2018.09.001.

Answers to reviewer 2 on "Porous granular flow models can predict full-scale landslide tsunamis"

The original authors

October 10, 2021

The review is shown in blue, answers in black.

Thank you very much for the detailed review of our manuscript. We would like to summarise the referees criticism and give answers to the main issues and a statement on why we think the paper is worth publishing. The detailed point-by-point answers can be found after the summary (page 11) Unfortunately, it has to be quite lengthy because basic principles are questioned in the review.

Claim 1) The model is neither novel nor better than simplified models.

To make this discussion easier, we would like to categorize models for landslide tsunamis in three groups (see Table 1): (1) purely empirical relations (e.g. Fritz, 2002), (2) highly simplified, optimized (i.e. depth-integrated or layered) models (e.g. Kim et al., 2019b) and (3) models with minimal simplifications (e.g. the presented model). Models become increasingly expensive (bad) but also increasingly detailed (good) and predictions are less dependent on an extensive parametrisation and optimisation (also good). Models from group 1 can be solved with a hand calculator, models from group 2 can be solved on a laptop. As far as we know, group 3 requires a high performance cluster or at least a high performance workstation. On the other side, empirical scaling relations (1) require many physical (or numerical) experiments to determine the necessary parameters (see e.g. Rauter et al., 2021). Simplified models (2) can be fitted with a few experiments and will then deliver reasonable results for many similar cases. Models from group 3 promise to require no flow experiments as they are based on fundamental physics (conservation equations) and intrinsic material parameters such as the friction angle (can be derived in a geotechnical laboratory with simple shear or triaxial test or

with the slope of the deposition of a sample) and the grain diameter (can be similarly determined). The parameters have an actual physical meaning that is connected to the material. This is at least the idea that we describe, execute and test with remarkable success in our paper.

These models coexisted for a long time and they will continue to do so for a long time. Which model is better is a question of the perspective and the task at hand. Sometimes a scaling relation, giving us the peak wave amplitude within seconds might be good enough, sometimes we might want to go a step further and run a simple flow simulation. Other times, and especially for advanced scientific investigations and case studies, it might be interesting to run a simulation that considers as much physical effects as possible.

Table 1: **Classes of landslide tsunami models**

Model	Computational cost	Modelled details	Parameters
(1) Empirical scaling relations, machine learning, artificial intelligence	minimal, with hand calculator	No details, only peak wave height or inundation for a fixed geometry	purely empirical from experiments, not adaptable to other geometries or flow regimes
(2) Simplified, optimized, layered and depth-integrated models	medium (1-10 ⁵ CPU hours)	Details of simplified wave kinematics as far as described by wave theory (but no breaking, etc.), sometimes approximate landslide runout and velocity, sometimes inundation (requires additional algorithm)	some physical, intrinsic material parameters, almost always extended with empirical parameters that are hardly adaptable to other geometries or flow regimes
(3) Full three-dimensional models with minimal simplification, Navier-Stokes, Discrete Elements, Lattice-Boltzmann	high (100-10 ⁷ and even more CPU hours)	Full details on wave, landslide, and all interactions (velocity, density, mixture, etc.)	mostly physical, intrinsic material parameters, ideally no empirical parameters, adaptable to other geometries and flow regimes

Our model is clearly positioned in group 3 and we point out the upsides (details, accuracy, predictive power of intrinsic material parameters) as well as the downsides (numerical diffusion, computation cost) in comparison to simplified/depth-integrated models. The referee points out the downsides of our model, e.g. "Their model clearly has numerical dissipation (and they admit it)", weakens the upsides "while the new presented model here features some new rheology and includes dilatancy effects that are missing in most models, [...] they do not get better results", does not recognize other upsides (e.g. direct simulation of breaking waves and inundation) and criticizes us for the few aspects that are still simplified, e.g. "[...] still considering a homogeneous porous medium with particles of the same size". In contrast, we strongly argue that our model (1)

includes previously unconsidered physics, (2) does not require a (backward) parameter optimisation, and (3) provides exciting new insight and detailed flow structures. The downsides of the method are (1) a high computational cost and (2) numerical diffusion (further reduced in the new version) in the far field which should be acceptable for many applications (source study, near field study, coupling with depth-integrated model for the far field). Many researches agree with us which can be seen with the increasing amount of studies focusing on category 3 models (e.g. Liu et al., 2005; Gisler et al., 2006; Pastor et al., 2008; Abadie et al., 2010, 2012; Horrillo et al., 2013; Shan and Zhao, 2014; Gabl et al., 2015; Wang et al., 2015; Shi et al., 2016; Viroulet et al., 2016; Kim et al., 2019a; Clous and Abadie, 2019; Yu and Lee, 2019; Franco et al., 2019; Romano et al., 2020; Xu et al., 2020; Abadie et al., 2020; Hu et al., 2020; Lee and Huang, 2020; Rauter et al., 2021; Franco et al., 2021). Note the rapid increase of such studies in recent years, Abadie et al. (2010) could only name four studies ("Few numerical models have been proposed to date, which can describe the full coupling between slide and water, together with the surrounding viscous/turbulent water flow.") with three-dimensional models while many attempts have been made since. This is clearly laid out in the following as well as in the original and updated manuscript. Note that we cannot give this full review in the manuscript, as that would exceed the allowed number of references. We especially try to give S. Abadie credit in our manuscript, as he pushed this topic very early with multiple publications.

We do not think that any group of model will replace any other group. They all take a very unique and important role. In fact, we highly value depth-integrated flow models and applied them in many works (e.g. Løvholt et al., 2008, 2015; Rauter and Tuković, 2018; Rauter et al., 2018; Kim et al., 2019b; Zengaffinen et al., 2020; Gylfadóttir et al., 2017, just to name those already included at some point in this review). However, class three models can do something very important and that is a reliable prediction of landslide generated tsunamis based on intrinsic material parameters and we show this with our work. The porous model is absolutely imperative for this task as it allows the prediction of pore pressure, where simpler models struggle (e.g. Ma et al., 2015; Clous and Abadie, 2019). The referee thinks that we overstress this but without a realistic pore pressure model, every rheology for a granular material will be quite questionable, since the concept of effective stresses of Terzaghi (1925) must be considered. Further, it resolves the full water surface and thus naturally predicts the breaking of waves, which turned out to be one of the most important mechanisms to determine the wave amplitude of subaerial landslide tsunamis (e.g. Bullard et al., 2019; Rauter et al., 2021). This is what we mean with complex wave dynamics.

Further, these advanced models give an exciting amount of flow details that are worth studying. We agree, that some of the results of group 3 models are also achievable with group 2 models, however, in the exact same way, a group 1 model is able to deliver some of the results of group 2 models. The referee does not question the value of group 2 models in the presence of group 1 models, so we do not understand why he question the value of group 3 models.

Further we would like to note that group 3 models are usually simpler than group 2 models when it comes to the mathematical equations: No depth-averaging, no corrections

for curved coordinate systems (this is usually dropped anyway, which is mechanically and mathematically not correct) and a smaller loss of physical effects. Group 3 models simply simulate all those effects, which requires all the computing power. This makes them very attractive for scientific investigations.

One last point we want to raise is the fact that the formal validity of simplified/optimized depth-integrated and layered models (group 2) is established with scaling analyses and the removal of terms that contain high powers of small scales. For the landslide this analysis was first conducted by Savage and Hutter (1989, 1991) and the respected depth-integrated flow model is still known as Savage-Hutter model in many communities. This analysis or formal proof of validity breaks down for subaerial landslide tsunamis as the dominance of lateral scales over vertical scales is not sufficiently prominent during the impact and tsunami generation. High waves (in relation to water depth or wave length) lead to the same effect and cannot be calculated with depth-integrated flow models. The same is the case for complex topography that is very hard to handle in depth-integrated models (I had to experience that myself in Rauter and Tuković, 2018) and nearly impossible if the surface curvature is large (Bouchut and Westdickenberg, 2004). The vast amount of the current generation of landslide tsunami models neglect these issues entirely and it is barely discussed in the tsunami community. The same issues are present in layered models. These are very strong upsides of models from group 3. In fact, we have to use full three-dimensional models for subaerial landslide tsunamis, if we trust the mathematical proofs that established depth-integrated flow models in the first place Savage and Hutter (1989, 1991). We do not see an advantage in spending a large part of the paper on discussing the applicability and validity of a model that we do not even use. We believe that users of such models are usually aware of these issues and readers that are not too familiar with these models would just be confused if such a discussion was substantially extended. Therefore we think that the short section where we mention this problem should be sufficient.

When it comes to novelty, we have to admit that there are full 3D models applied to real case landslide tsunamis (e.g. Abadie et al., 2012, more references above) and that there are porous granular flow models that deliver exceptional results in laboratory experiments (e.g. Si et al., 2018b, more references above). What makes our work novel is the application of the porous, granular $\mu(I), \phi(I)$ -rheology that simplifies the porous granular flow model sufficiently to attempt full scale 3D simulations with it. We think that our simulations and the reproduced figures speak for themselves, as no such detailed flow simulations can be found elsewhere. The referee argues that these flow details are not important. We disagree because with better landslide physics, we can provide a platform for progressively gaining new insights and can improve our understanding of the processes substantially. This is not possible without a porous granular flow model for the landslide dynamics. This is why we think that this work fits well into Nature Communications.

The referee did not find any strong weaknesses in our model, except for the drawbacks that are reported in the manuscript and repeated here. Notably, no model is perfect and we had to neglect various phenomena to keep complexity to a level that can be handled in a full scale application: Sub-grid turbulence, surface tension, capillary stresses,

polydispersity, particle lift, virtual mass, etc. Further, we put emphasize on a model that is simple enough to communicate its mechanics and to give it a chance of being applied in practice. However, to our knowledge, there is presently no other landslide tsunami model that takes these phenomena into account (possible exception: sub-grid turbulence, see e.g. Rauter et al., 2021, and supplementary materials) and they can be added to the model at a later point in time, once we understand the basic model of leading order processes. In fact, we think that the more simplified tsunami models suggested by the referee can be derived from our model and various assumptions/simplifications. This shows once more the value of the model - we can use it to investigate and validate simplified models.

Weaknesses in the presentation (sensitivity study, mesh refinement study, a wrong value for the viscosity of water) have been corrected in the updated revision.

Claim 2) The model will not be applied and does not fit into Nature Communications.

Regarding model complexity, the referee writes "this is where the art of the modelers comes in to decide about the best class of models/physics to apply to a given problem." and we fully agree with this. Every modeller has to decide which effects are worth modelling and we had to make this decisions too. The focus of this paper is to carry out research high resolution modelling with limited number of assumptions. We spend a lot of time during the last three years to find out which level of complexity is appropriate. Notably, a lot of time was spend on models that were too simple, as we were often too optimistic. A very interesting example: A constant friction coefficient μ (which we would have preferred for its simplicity and two less model equations) did not work and we had to switch to the $\mu(I)$ -rheology to get reasonable results (Rauter, 2021).

The referee underwent the same process and took a different path, which we fully accept. We are convinced that we need a large varieties of studies that consider different processes so that we can identify the really important ones. Our model focuses on a few phenomena of which a large group of scientists think that they important (a large part of the granular flow community, especially the one working with the $\mu(I)$ -rheology, see references in the manuscript), and apply it for the first time to a real scale landslide tsunami.

We think that our model deserves a place in the toolbox of modellers and that it will play an important role in the future. This is especially the case with the decreasing cost for computational capacities. The real case simulation required approximately 100 000 CPU hours, which approximately corresponds to a cost of 10 000 EURO at e.g. Amazon EC2 cloud service. This cost is reasonable in comparison to e.g. personal cost or material cost of experimental studies and will decrease rapidly in the future (decrease by half every two years following Moore's law https://en.wikipedia.org/wiki/Moore%27s_law). Currently, the presented model is at the edge of what is reasonably possible and we had to make compromises with resolution, number of simulations, etc. However, this cutting edge nature is also the reason why we see fit in a journal like Nature Communications.

It is up to the reader to decide whether to choose a model from class 1, 2 or 3. However, especially from a researchers point of view, group 3 models are very interesting and thus we see good fit of our model in Nature Communications. Not publishing a class 3 model because it is more expensive than class 2 models and because it might deliver similar results in some cases hinders the progress in modelling. With the same arguments depth-integrated models would probably have never been published, as they are computationally more expensive than empirical scaling relations.

Claim 3) Literature on existing models, laboratory experiments and real cases is not reviewed extensively.

We follow the author guidelines of Nature Communications regarding the literature that we include in our manuscript. As a consequence, we had to limit the references before submission to a rather brief selection. By definition our paper can not include an extensive review of three different topics, which is requested by the referee (depth-integrated landslide tsunami models, real events and laboratory studies). Our paper is not intended to be a review paper. Therefore we have to limit the references strictly and cannot include references that are only distantly related to the topic of our work (that is the simulation of real scale subaerial landslide tsunamis with a full 3D porous granular flow models). This excludes many of the suggested case studies with depth-integrated models or for subaquatic landslide tsunamis.

Regarding **existing models** we focus on recent (all from the last 2-3 years) developments in category 3 models for waves (Si et al., 2018b; Clous and Abadie, 2019; Mulligan et al., 2020; Chen et al., 2020; Abadie et al., 2020; Rauter et al., 2021) or slides (Rauter, 2021) and include review papers on category 2 models (Løvholt et al., 2015; Yavari-Ramshe and Ataie-Ashtiani, 2016). We further include category 2 models if they share a property with our model, e.g. porous granular rheology (Pailha and Pouliquen, 2009; Pudasaini, 2012; George et al., 2017; Kafle et al., 2019) or coupling to a full 3D model (Løvholt et al., 2008; Abadie et al., 2012). Empirical relations (group 1 models) can be found in some of our references but they are too different from category 3 models to be discussed in the manuscript. Notably, we added some of the referees suggestions regarding models, i.e. Grilli et al. (2019), Zhang et al. (2021b) and Ma et al. (2015).

Regarding **laboratory experiments** we include the one that we simulate in our paper (Viroulet et al., 2013, 2016) and others which give some specific insights, e.g. on particle size effects (Lindstrøm, 2016; Bougouin et al., 2020) or scale effects (Heller et al., 2008). We are well aware about the large amount of laboratory experiments but if they are not strongly related to our work we cannot include these references, as this would exceed the allowed number of references.

Regarding **real events** we cite geological studies for events with very high impact and importance (Walter et al., 2019; Wang et al., 2015; Paris et al., 2019; Sepúlveda and Serey, 2009; Sassa et al., 2016; Panizzo et al., 2005; Harbitz et al., 2014; Ramalho et al., 2015; Paris et al., 2017; Weiss et al., 2009) as well as the study that is simulated here (Gylfadóttir et al., 2017).

The remaining references are **the basis of our method**, e.g. regarding OpenFOAM and CFD (Weller et al., 1998; OpenCFD Ltd., 2004; Rusche, 2002; Passalacqua and Fox, 2011; Roenby et al., 2017; Courant et al., 1928; Juretić, 2015; Rauter et al., 2018; Ergun, 1952) or granular porous rheology (Roscoe et al., 1958; Pouliquen et al., 2006; Lambe, 1973; Schaeffer, 1987; MiDi, 2004; Jop et al., 2006; Lagrée et al., 2011; Johnson and Jackson, 1987; Boyer et al., 2011; Trulsson et al., 2012).

Finally there are some **extended reading suggestions** regarding polydisperse granular flows (Barker et al., 2021; Festa et al., 2015), non-local rheology and granular temperature (Campbell, 2006) and turbidity currents Heerema et al. (2020).

The only reference that did not fit into this scheme is Kim et al. (2019b), which we removed from the revised manuscript to make space for other references. It was previously included because it has become a standard reference for depth-integrated landslide tsunami modelling in our group. We agree that this is not enough to justify the additional reference. We replaced this reference with some references that the referee suggested.

We screened all of the reviewers references, which were available to us. Notably, five references are not available and only eight references contained modelling of subaerial landslide tsunamis, only one of which used a full three-dimensional (category 3) model. This reference is already included in the manuscript (Abadie et al., 2012). Given the restrictions in number of references, we could only add a few of the suggested references. We selected the additional references following this reasoning:

- Abadie et al. (2012): **already included**
- Grilli et al. (2015): **not included**: Only subaquatic, no group 3 model
- Grilli et al. (2017): **not included**: Only subaquatic, no group 3 model
- Grilli et al. (2019): **newly included** as additional reference for AK: important event, subaerial landslide tsunami.
- Grilli et al. (2019b): **not available**: AGU presentation
- Grilli et al. (submitted): **not available**, submitted (?) manuscript, not found on arxiv
- Higman et al. (2018): **not included**: Case study with (although interesting!) limited impact which we do not model in this work.
- Horrillo et al. (2013): **not included**: Only subaquatic landslides, Newtonian rheology only
- Hunt et al. (2021): **newly included** as an additional references to AK: important event, subaerial landslide tsunami.
- Ma et al. (2013): **not included**: Only subaquatic landslides, Newtonian rheology

- Ma et al. (2015) **newly included** to the manuscript because it contains granular rheology and subaerial landslide tsunamis. It provides an interesting viewpoint on the pore pressure and the related difficulties. Thank you for this suggestion.
- Schambach et al. (2019): **not included**: Only subaquatic landslides, prescribed landslide motion (no slide modelling)
- Schambach et al. (2020): **not included**: Only subaquatic landslides, prescribed landslide motion (no slide modelling)
- Schambach et al. (2021): **not included**: Only subaquatic landslides, simplified (category 2) landslide modelling
- Tappin et al. (2014): **not included**: Only subaquatic landslides, prescribed landslide motion (no slide modelling)
- Tappin et al. (2019): **not available**: AGU presentation
- Tehranirad et al. (2015): **not included**: The source model/simulation is the same as in Abadie et al. (2012). Regarding coupling to a far-field model this is neither the first nor the most updated reference.
- Ward and Day (2001): **not included**: Class 2 model. Prescribed landslide motion (?), La Palma is covered by (Abadie et al., 2012, 2020).
- Watt et al. (2019): **not available**: AGU presentation
- Wang et al. (2019): **not included**: Class 2 model. Mount Unzen is already covered.
- Zengaffinen et al. (2020): **not included**: Group 2 model, Anak Krakatoa is already covered by multiple references.
- Zhang (2020): **not available**: PhD Thesis, related to the following papers (?)
- Zhang et al. (2021a): **not included**: No relevant simulations, only numerics validation.
- Zhang et al. (2021b): **newly included**: We added this reference as an example of depth-integrated models.

Claim 4) The lab case is a bad choice and the 2014 Lake Askja event is not an exceptional case study.

We choose the benchmark from the NTHMP Landslide Tsunami Benchmark Workshop (<https://www1.udel.edu/kirby/landslide/reporting.html>) to make sure that we use a benchmark that is accepted by the community.

We do not see any use in modelling tsunamis generated by solids (Benchmarks 1-3) as the granular porous rheology is the novelty of this work. However, such simulations with OpenFOAM (the wave treatment is the same as in our work) have been conducted and validated by Romano et al. (2020).

Further, we do not see any use in modelling subaquatic landslide tsunami experiments (Benchmark 4) because they do not cover all effects seen in subaerial landslide tsunami events. However, we are certain that the model will perform well in this case and consider this benchmark for a future work that is focusing on subaquatic landslide tsunamis.

Benchmark 6 includes too many components that are difficult to model and which introduce more uncertainty than they bring clarity. This is, for example, the piston accelerated landslide and the smooth basal surface. Further, the three-dimensional nature of this experiments makes model runs much more expensive (even for simplified models, see Zhang et al., 2021b) and the comparison between simulation and experiment much harder (we have no nice side view or cut). Notably, the Navier-Stokes Equations are space-invariant (in contrast to depth-integrated models) and therefore it is not expected that the accuracy differs between 2D and 3D cases. Again, this benchmark is interesting but in our opinion less appropriate for our needs than Benchmark 5.

Benchmark 7 is a real event in Alaska from 1964. We think that the Lake Askja event is much better documented and constrained (not surprising, considering modern data collection methods) and thus used this event for the 3D real scale study.

For these reasons we decided to use benchmark 5 from the NTHMP Landslide Tsunami Benchmark Workshop for the model validation and further Lake Askja to give an insight into a real event.

Further we want to note that our study should not be seen in isolation. OpenFOAM was validated with a large variety of 2D and 3D experiments in the last couple of years (e.g. Chen et al., 2020; Rauter et al., 2021; Romano et al., 2020) and the wave treatment is the same in all of these investigations (MULES algorithm and Counter Gradient Transport). Solely the tsunami source was considered differently in these works, either as solid (Chen et al., 2020; Romano et al., 2020) or as a Newtonian fluid (Rauter et al., 2021). Similarly the landslide model was validated with various experiments by Rauter (2021); Barker et al. (2021); Rauter et al. (2020). From these studies we also get a very good idea on the required mesh size, sensitivity to parameters, etc.

Lake Askja event shows some very particular features that other events do not provide. This is all mentioned in the manuscript but we are happy to repeat it here in more detail and more directly:

- An event in an isolated lake allows us to simulate the whole event with the full 3D model. We do not have to worry about wave absorbing boundaries and their influence on the event and we do not necessarily need coupling with a depth-integrated model in the far field. This would complicate the development further and probably necessitate a separate study. Further, the landslide volume is well constrained by the increased water level in the lake.
- Good documentation with bathymetry before and after the event and an exact delineation of the inundation (due to the snow covered shore). This eliminates

other events such as Loen or Vajont (The wave height above the dam is not really well constrained) and most other events before the year 1960. Also Anak Krakatau is not so well constrained in this regard as it occurred during volcanic activity. This is for example shown by the fact that most attempts to simulate this event include multiple scenarios.

- Further, Lake Askja includes wave propagation over a horizontal distance of approximately 3000 m, which still allows us to investigate near field tsunami propagation and inundation.

Finally we would like to note that we provide a high number of simulations considering that this paper introduces a new model of high complexity. Unquestionably, there are more case studies for other, long established models, such as NHWAVE or Bingclaw/Geoclaw, but we do not understand why that should be a reason against publishing our manuscript. In fact, we think it is the exact opposite, presenting a fresh model and a fresh new case to the community.

We answer all raised questions and issues in detail in the following.

In this paper, the authors apply a CFD model with a new rheology for multi-phase flows, developed within the OpenFoam library to simulate one very small scale two-dimensional experimental benchmark and one large scale field case study of landslide tsunamis generated by a granular material subaerial slide. This is a very computationally expensive model and hence, they basically present 2 simulations (although they simulate a few cases for the less-costly experiments) with no sensitivity study of results to numerical or physical parameters.

We added the requested sensitivity study and mesh refinement study to supplementary materials.

The authors present their work as a large step forward, leading to remarkable results (such superlative statements are made many times throughout the paper) and often make strong but not always well-supported statements in the paper regarding previous work and their own work and quality of results. They imply directly in their abstract that the “lack of consistent models incorporating both the landslide and the wave mechanics represents a gap in providing unbiased predictions of real event”, indicating in the text that apparently most other modelers tune their model parameters to match observations and hence other models can’t be predictive ! This reviewer strongly disagrees with both statements and while the new presented model here features some new rheology and includes dilatancy effects that are missing in most models, judging by the results for wave generation and propagation, they do not get better results or get even worse results in the far-field than other modelers could achieve (based on this reviewer’s experience) for similar experimental benchmarks or field case studies.

We admit to use superlative statements because we were excited by our good results. We tried to tone down the statements.

The described issue ("lack of consistent models incorporating both the landslide and the wave mechanics") is supported by referee 3 and by all of the referees references (see below). The references mentioned here that the landslide motion is either predefined by a two-parameter curve (6 articles), simulated as Newtonian or inviscid fluid in 3D (3 articles), or simplified depth-integrated models with fitted parameters (7 articles). The remaining articles contain no model or are unavailable.

Further, many of the articles deal exclusively with subaquatic landslides that represent much simpler physical phenomena (no impact/splashing, no dry to wet transition, no mixing, no breaking waves, etc.) and that fit well to the assumptions of depth-integration (see, e.g. Savage and Hutter, 1989, 1991). This is in strong contrast to subaerial landslide tsunamis where all these effects make a more complex model necessary or at least highly appropriate.

The referee disagrees with the statement that "modelers tune their model parameters

to match observations" yet all his references support this statement. We screened all of his references and the result shows clearly that the determination of model parameters (if even present) is always based to reproduce final results (either wave amplitude or inundation). Most of them prescribe the landslide motion with a simple two-parameter function $s = f(t, s_0, t_0)$ (based on transition time t_0 and transition distance s_0 , which are fitted to produce the wave).

In detail:

- Abadie et al. (2012) uses an inviscid fluid (no rheology, no rheology parameters) but highlight the value a realistic rock avalanche model (as in our work) would have: **"Building a model dedicated to simulating a rock avalanche entering the water would be a challenging task and so far there has not been, to our knowledge, any attempt to do so in the literature.** In the present study [...], we used the simpler standard approach, in which the CVV slide is considered as an inviscid fluid."
- Grilli et al. (2015): Prescribed landslide motion: "In the absence of more detailed information on SMF kinematics, we will use this simple law of motion [... an equation $s(t) =$ follows ...]"
- Grilli et al. (2017) use a lot of fitting but the following sentence is most revealing: "[...] a Manning coefficient $n = 0.04$ was calibrated so that the modeled slide reached the bottom of the slope at the time measured in experiments [...]."
- Grilli et al. (2019), supplementary material: "[...] some preliminary simulations and sensitivity analyses to the AK collapse parameters were performed [...]". Unfortunately this is the only information that is available. However, the application of a "viscous model" makes clear that parameter fitting was involved, as soil/rocks do not have an intrinsic viscosity.
- Higman et al. (2018): Not a modelling paper, parameters not mentioned.
- Horrillo et al. (2013): "The friction term in the momentum equation can be adjusted to mimic the internal friction within the fluid body, i.e., the viscosity coefficient. This coefficient has been chosen to give the best possible agreement with the reference data. [...] a typical value [...] ranges between 0.001 m²/s and 10 m²/s"
- Ma et al. (2013): Prescribed landslide motion: "The displacement of the rigid landslide is described as [... an equation $s(t) =$ follows ...]"
- Ma et al. (2015) uses a granular rheology but struggles to describe the pore pressure: "the parameter λ determines the magnitude of Coulomb friction imposed on the landslide [...] Here, we simply assume λ is a constant which will be calibrated using laboratory measurements." Notably porosity in our models allows us to simply calculate the parameter λ at every point in time and space.

- Schambach et al. (2019): Prescribed landslide motion: "Here, this simple law of motion for rigid slumps is used, which reads [... an equation $s(t) =$ follows ...]"
- Schambach et al. (2020): Prescribed landslide motion: "rigid slumps, with their kinematics described by that of their center of mass motion"
- Schambach et al. (2021): "we used the same granular density and internal friction values for the slide material as in Nakata et al. (2020) [...]. Hence we have, [...] internal friction angle $\phi_i = 30^\circ$, and basal friction angle $\phi_b = 2^\circ$." We strongly assume that this low value has been fitted, as such low values are hardly ever used.
- Tappin et al. (2014): Prescribed landslide motion: "[...] a rigid slump with constant basal friction and negligible hydrodynamic drag, we find (Grilli and Watts, 2005), [... an equation $s(t) =$ follows ...]"
- Tehranirad et al. (2015): "the slide material was modeled as a heavy Newtonian fluid."
- Ward and Day (2001): Prescribed landslide motion (?). We are actually not sure.
- Wang et al. (2019): "At initiation, we set a larger initial basal friction coefficient $\mu_b = 0.5$ for the solid-like sliding and then decrease the value $\mu_b = 0.01$ to mimic the fluid-like movement. The dynamic friction coefficient on dry land is $\mu_d = 0.002$. Underwater dynamic friction coefficient increases to $\mu_d = 0.02$. We obtain these parameters by trial and experiment [...]."
- Zengaffinen et al. (2020): A work from our group. Material parameters are optimized to the wave signal and a sensitivity study investigates the wide range of possible parameters.
- Zhang et al. (2021a): No relevant simulations.
- Zhang et al. (2021b): "The estimated range of the effective viscosity is [0.5630, 4.8419] kg/(m s) and we choose $\mu_e = 1.408$ kg/(m s) to best fit the results to laboratory data on the wave generation"

We commented on the requested literature review above.

Their model clearly has numerical dissipation (and they admit it), which has been well-known for decades when simulating wave propagation over a significant distance with this class of NS models. A fact that has led many modelers to implement model coupling. The latter, which has been done for over 2 decades as well, is presented here as the solution to their dissipation problem for future work.

Numerical diffusion is present in every numerical model and there is nothing to "admit", rather we report it in an objective and clear manner as it should be. Notably,

depth-integrated models suffer also from numerical diffusion and it can lead to an over-estimation of the slide spread and runout (e.g. Rauter et al., 2018). This is also clearly visible in the referees references, e.g. Zhang et al. (2021a). Numerical diffusion is controlled well in our model and is a smaller issue than thought prior to this study. This is shown, among others, by the decrease of the first wave amplitude between the first and fourth gauge, which is only 0.2 mm or 1% of the wave amplitude (see supplementary materials). This is approximately the error due to numerical diffusion (it will obviously grow with distance from the source) and it is basically neglectable considering the high geological and physical uncertainty of the problem under investigation.

Notably, we used a wrong value for the water viscosity which introduced further unrealistic (but still small) diffusion. We corrected this in the new version and the accuracy of the model is improved.

The coupling to a depth-integrated model in the far field is not a replacement for the presented model/study. The presented model could simply be one of the models that are involved in the coupling. The second model would be the depth-integrated model, e.g. Geoclaw or NHWAVE. For subaerial landslide tsunamis, the assumptions of depth-integrated models do not hold in the generation phase (see above) and one will always need a full three-dimensional model for consistent and valid simulations. In contrast to the referees statement, we did not find an example of a 3D porous landslide model that was coupled to a 2D model in the far field and the referee does not provide a reference either. To our knowledge, this has only been done with simple Newtonian models (which we aim to replace with the porous granular flow model) and we cite the respective literature.

Another important concern is over-stressing the improved accuracy of including new physics (rheology, air bubbles, . . .), but still considering a homogeneous porous medium with particles of the same size. This may be the case in experiments, but this is not the real world at all of flank collapses and avalanches. Hunt et al.'s (2021) recent field work on the Anak Krakatau 2018 collapse showed deposits made of many enormous blocks (100s of meters in dimension) sitting together with small granular material and everything in between. The material used in the field case study here is 10 cm across and uniform ! So refining the modeling of a porous medium and its rheology does not necessary lead to better results for real events and, in fact, simpler models based on particles of many size and their shocks have done quite well in the past (e.g., many works from Ward and days, 2001 to Wang, 2019). Recent work in modeling the seminal and well observed AK 2018 event, using the latest generation of two-layer models (not reviewed or cited here; e.g., Zhang et al., 2021a,b) showed that non-hydrostatic Euler models that are quasi 3D in the water with a depth-integrated layer of viscous or granular material, which is non-hydrostatic as well, and a simpler rheology than here did very well in predicting the large near-field runups and by coupling to a Boussinesq model the far-field inundation and runup (Grilli et al., 2019a,b,2021; Zhang, 2020). The measured runup heights up to 85 m on AK's nearby island was closely predicted by such models (actually much closer

than results presented here in the lake). Another example of successful modeling with other landslide tsunami models is Palu 2018 (Schambach et al., 2021).

Porosity allows us to determine the effective pressure which is imperative for the landslide friction. This fact cannot be over-stressed.

Previous works which did not consider porosity had to guess the effective pressure, e.g. one of the referees references (Ma et al., 2015): "the parameter λ determines the magnitude of Coulomb friction imposed on the landslide [...] Here, we simply assume λ is a constant which will be calibrated using laboratory measurements.". The role of the factor λ in Ma et al. (2015) is to make the splitting $p_{\text{total}} = \lambda p_s + (1 - \lambda) p$ and our model can base this on the physically known relation with the porosity ($p_s = f(\phi_s)$). In fact, we think that this is a very good example for the issue we aim to solve and we decided to include this reference in our manuscript. If one ignores pore pressure completely, one will get the issues shown by Clous and Abadie (2019), namely that the pore pressure artificially increases the slide friction and thus retards the slide too quickly. If one tries to rely on pressure-independent rheologies (Newtonian, Bingham, Herschel-Bulkley), one will get the issues shown by Viroulet et al. (2016). The only reliable solution, as far as we know, is a porous flow model, as pore pressure and porosity are highly correlated. The importance of pore pressure was first described by Terzaghi (1925), the correlation with the packing density by Roscoe et al. (1958). It is in geotechnical standard literature for a long time (e.g. Schofield and Wroth, 1968) and fundamentally established in that community. Many researchers agree with us and this concept is getting increasingly common for geophysical phenomena (e.g. Savage et al., 2014; Wang et al., 2017b,a; Cheng et al., 2017; Chauchat et al., 2017; Heyman et al., 2017; Si et al., 2018b,a; Schaeffer et al., 2019; Baumgarten and Kamrin, 2019; Rauter, 2021).

Further, even depth-integrated models try to incorporate pore pressure and dilatancy effects (e.g. Pailha and Pouliquen, 2009; Pudasaini, 2012; Bouchut et al., 2017), although it is very difficult in such a setup and probably not possible with the added difficulty of the free water surface and the tsunami. Note that most of these works (except Si et al., 2018b) focus only on the landslide and not the tsunami and that none of them approaches a real event, which highlights the value of our work once more. Please note that a complete review is not possible in the article due to the limitations in number of references and length. However, we cite the most important and recent articles and we are sure that the interested reader will be able to navigate through the literature from there.

We agree with the referee on the importance of polydisperse (scientific name for "particles of many size") granular flows (see introduction of Barker et al., 2021) and hope that we will be able to include them in the future. The referee surely agrees with us that there are many steps necessary to make that possible and that the simulation of a monodisperse granular landslide tsunami is a good and imperative starting point. We work intensively on polydisperse granular flow models and are in fact able to model them with OpenFOAM (Barker et al., 2021). As far as we know, we have the only continuum mechanical model which is able to do such simulations, however, not yet in combination with the tsunami. Therefore we are rather surprised that the referee criticizes this as-

pect and calls for polydisperse flow models. Anyway, our model is the first, where this question even arises, which shows again its innovative character.

Regarding the decision to neglect the polydisperse nature of landslides: There is evidence from experiments and DEM simulations, that the average grain diameter can be used in the $\mu(I)$ -rheology to model polydisperse granular flows (see Barker et al., 2021, and references therein). In the absence of segregation, the average grain diameter will be constant for the whole slide and thus a material constant. We do not expect segregation to occur in the short flow until the tsunami impact and it is safe to assume that the average grain diameter can be used in the $\mu(I)$ -rheology with good accuracy. Unfortunately, there is less known about the permeability of a polydisperse grain mixtures but the filter rules of Terzaghi (Fannin, 2008) suggest that the smallest 15% of the particles are responsible for the permeability. Therefore the 15%-percentile of the grain diameters, d_{15} , is usually used in estimating the permeability. For the sake of simplicity we used only one diameter in our model and that is the estimated value for d_{15} . This parameter aims to fulfil the permeability and the filter rule of Terzaghi because the $\mu(I)$ -rheology is not very sensitive to the particle diameter (at least not in this range, see sensitivity analysis), where it corresponds to a change in I_0 . This is discussed, although shorter, in our manuscript.

Contrary to the reviewers claim, we do not find that Ward and Day (2001) or Wang et al. (2019) consider the particle size and polydispersed granular flows.

We reviewed all of the referees references and they support our claim that parametrisations are empirical, fitted to wave signals and inconsistent, see the listing above. This need for optimisation to the wave results is exactly what we try to solve.

Zhang et al. (2021b) neglect many three-dimensional flow patterns, including mixing of grains and water, impact craters, braking waves, eddy in the impact region, the terrain curvature is completely neglected. We are aware that depth-integrated/layered models can be fitted to reproduce observed wave patterns and we even show such results for comparison in our article.

One other concern about the paper is its lack of an exhaustive review to date of other well experimentally-validated landslide tsunami models also applied to real case studies, that may just use a simpler model of the granular medium than here, but have performed well for predicting waves. Perhaps those models will not simulate the slide runout and deposits as well as the proposed model (which still needs to be demonstrated in the present paper for their cases as they do not show precise slide deposits), but they model well enough the tsunamigenic part of the slide motion and wave generation to be predictive without needing parameter adjustments. The authors here either lack knowledge of that other work or prefer to cite a subset of literature related to their own group of collaborators they know better? In any case an exhaustive and accurate review of the state of the art is missing here.

This article does not contain an exhaustive review of depth-averaged models because it is not a review paper and depth-averaged models are not the focus of the paper. We

are also close to the maximum limit of references and have to select relevant references carefully. We cite review papers of depth-averaged models for this sake (Løvholt et al., 2008; Yavari-Ramshe and Ataie-Ashtiani, 2016) and added some of the referees suggestions in the new version. Most three-dimensional models use non-granular non-porous rheologies and all exceptions we know of are cited in the manuscript. If the referee knows any other study with such models we would be very happy to include it in our manuscript.

We checked the suggested references as stated above.

And in tsunami hazard assessment, predicting the waves in the near- and far-field is more important than predicting the details of the slide deposits !

We show that we can do both.

We believe that it is not very consistent to simulate the tsunami wave correctly based on the wrongly simulated tsunami source, i.e. the landslide. This is exactly the problem we try to tackle. How should we predict future events if we (have to?) use wrong tsunami sources to get the inundation right?

The authors also state: "For the first time, we show a model that scales consistently from small scale laboratory experiments to full scale catastrophic events. "Well, NHWAVE and its many applications to lab and field cases did just that (e.g., Ma et al., 2015; Grilli et al., 2017; Zhang et al., 2021a,b; Grilli et al., 2019a,b,2021; Schambach et al., 2021).

All references (except Ma et al., 2015, see below) are applying non-intrinsic material parameters that are situational and thus not constant for a specific material but highly depending (multiple orders of magnitude) on e.g. landslide size.

This is, for example, the landslide viscosity, that is described as a constant in many landslide tsunami models. Other models go further and apply a plasticity model with a constant yield strength, but also this property is situational and changing with scale. We know that granular flows, grain suspensions, etc. are not described well by a constant viscosity or yield strength (e.g. Jop et al., 2006) but as a particle pressure depended and shear thinning viscosity or yield strength.

The strongest scaling is related to the pressure level and thus the size of the landslide. A large landslide (e.g. real scale) will have a high effective viscosity or yield strength, a small landslide (e.g. laboratory experiment) will have a low effective viscosity or yield strength. This can also be seen in the suggested references: Grilli et al. (2017) use "a fairly high equivalent slide viscosity $\mu_s = 500 \text{ kg}/(\text{m s})$ " for a large real scale landslide. Zhang et al. (2021b) on the other hand write "The value of $\nu_e = \mu_e/\rho = 0.00001 \text{ m}^2\text{s}$ provided the best fit to the slide motion and shape." (corresponds to $\mu_e = 0.02 \text{ kg}/(\text{m s})$) about the application to a small scale laboratory experiment. Another example from our group: Gauer et al. (2006) use a plasticity model and a yield strength of 25 Pa for the

laboratory experiment but Gauer et al. (2005) use a yield strength between 25 kPa and 195 kPa for the real scale simulation. This is a factor of $10^3 - 10^4$ that corresponds quite well to the scaling of the pressure. The material is both time fine grained sediment/clay. This is, in our opinion, not consistent scaling and requires improvement.

As mentioned above, Ma et al. (2015) differ from this general scheme of non-intrinsic material parameters and uses a granular rheology with friction coefficients that can be understood as intrinsic material parameters. This work is the closest to our work. However, as mentioned before, Ma et al. (2015) are not able to determine the pore pressure and have to assume it, where our model can make physically based predictions.

Besides, the authors here only show a 2D lab case with reasonable results, where only the first wave is accurately predicted, and a single modeling of a field event where they do not capture inundations that well (the only telltale of the quality of their wave simulations since no other wave measurements were made), and infer that their model performs well across scales.

We appreciate that the referee wants to see more simulations. We will provide more case studies in the future but we do not think that it is necessary for the first study. We perform about as good as an automatically and fully optimized depth-integrated model in the real case. We added a quantitative error description. The claim that we perform "not that well" is not supported by this error quantification.

The authors state they have: "Unique and complete field data, along with the limited geographic extent of lake Askja, reminiscent of a large scale laboratory, are instrumental for the rigorous validation." This event although a good case study is far from unique. Other well-documented case studies involving subaerial and submarine granular slides have been recently successfully modeled. For AK 2018, see Hunt et al. (2021) and new successful modeling by Grilli et al. (2021) and Zhang (2020) based on it of both landslide and tsunami generation and impact (also presented at AGU; Grilli et al., 2019b; Tappin et al. 2019; Watt et al., 2019). See Palu 2018 (Schambach et al., 2021). See 2015 Taan Fiord Alaska (Higman et al., 2018).

None of the mentioned cases are lake tsunamis. Events in lakes have some substantial upsides for model testing, see above.

The CVV case is discussed but the seminal paper on volcanic collapse landslide tsunami by Ward and Day (2001) should be mentioned. Besides Abadie et al. (2020) which revisits Abadie et al. (2012) with a modified geometry, Tehranirad et al. (2015) shows the CVV wave propagation ? 3D-NS modeling by Horrillo et al. (2013) and subsequent in Gulf of Mexico and Puerto Rico work is also missing. As well as the whole suite of work by Ward et al., starting with tsunami Ball model (Ward and Day, 2001)

to tsunami square (e.g., Wang et al., 2019).

We discussed the issues related to including more references above.

Laboratory work for landslide tsunamis, solid or granular, is also insufficiently reviewed and why not model a 3D laboratory case here, for granular subaerial slides as there are quite a few which are a better test of the present model, as others have done (e.g., Ma et al., 2015; Zhang et al., 2021a,b).

We appreciate that the referee would like to see more simulations of our model. We intent to publish more case studies in the future.

We are aware about the high number of laboratory work on landslide tsunamis, however, a research article is not the format to review this extensive literature. Experimental work that is relevant for the study is cited and an extended review on laboratory experiments can be found therein. We discussed the cited literature in detail above.

Regarding effects of rheology and material properties on tsunami generation, the citations are also incomplete and the authors do not mention studies with “simplified rheologies” that well match both laboratory experiments and full scale events (e.g., Ma et al., 2015; Grilli et al., 2017, 2019a,b, 2021; Schambach et al., 2021; Zhang et al., 2021a,b; and many others).

We discussed the cited literature in detail above.

The main reference to indicate a “paradigm shift is needed” is a recent conference presentation. The authors mention without support: “key issue has been the inability to explain both the landslide dynamics and the tsunami with a unified model and parametrization.” Not quite true in many recent work regarding accurately generating and propagating waves. See some references listed above. This statement is much too strong and while one can always improve models (rheology etc. . .), this reviewer does not believe the statement truly reflects published papers and the state of the work in the field. This would need more support in the paper.

We cite references of high quality and although we prefer peer-reviewed journal articles. The recent conference presentation describes the problem very well, better than we could phrase it. Further the reference comes from a scientist that is well established in the community.

The authors propose a “significant leap forward” but the type of rheology used here is not new and may not be needed to explain wave generation in many large full scale events (a sensitivity study would show this better). As mentioned, Hunt et al (2021)

and Watt et al (2019), for instance show that AK's collapse had many blocks as large as hundredth of meter across and hence is far from a continuous porous material. The granular and viscous rheologies used in successful models (e.g., Grilli et al. 2019a,b, 2021; Zang, 2020) well explain all near-field and far-field tsunami observations at AK, as well as landslide deposits. Models based on particle collisions such as tsunami-square also performed rather well.

The rheology was published in Rauter (2021) and is thus new. The application of simpler versions of the $\mu(I)$ -rheology to landslide tsunamis is also new (Clous and Abadie, 2019). In fact, we think that we apply the compressible $\mu(I),\phi(I)$ -rheology for the first time to a real full scale event. As such we even exceed the state of the art in landslide modelling, yet alone landslide tsunami modelling.

The comment that the rheology "may not be needed to explain wave generation" is exactly what our paper aims to answer. Our findings suggest that we need realistic rheologies to make realistic predictions.

Further, all of the referees references that use 3D models suggest to include a granular rheology in the outlook, e.g. "First of all, real slides are complex phenomena whose behavior depends on soil type as well as environmental and geometrical parameters (Varnes, 1978). This complexity must be taken into account in the model, in particular, by using a relevant rheological law." (Abadie et al., 2010), "Through means of a sensitivity test and by applying it to a real tsunami event (PNG), it was concluded that the generated wave depends strongly on the constitutive law of the landslide rheology." (Horrillo et al., 2013), "In the present paper, the computations carried out in Abadie et al. (2012) are redone, improving their accuracy by calibrating the slide fluid viscosity in order to approach a granular slide (Sects. 2.1 and 3.1) with a Newtonian model." (Abadie et al., 2020)

Large blocks are usually not single particles or boulders and rocks but conglomerates of smaller particles (Festa et al., 2015). As shown by Rauter (2021), the rheology is able to model the cohesive plug-flow (block-like, conglomerate-like) movement of overconsolidated granular flows. This example shows very well the value of realistic rheologies.

We acknowledge that simplified models are able to explain some observations, however, more complex models are able to explain more observations with less assumptions. Explaining more observations with less assumptions is the definition of higher predictive power and thus better in this regard.

We would like to note once more that very simple empirical scaling relations are similarly able to explain many landslide (e.g. α - β -model) and tsunami observations (e.g. the scaling relation of Fritz (2002)) after a very extensive fitting of parameters. However, simple (depth-integrated) models add knowledge and insights on top of the information which scaling relations are able to provide. Our model goes a step forward and provides even more knowledge and insights.

The unknown level of saturation in actual collapses makes it doubtful a continuous saturated porous medium would work better than simpler models.

Simpler models have to assume a constant saturation during the whole event. The saturation is not able to change in non-porous flow models and the slide density will remain constant during the whole event (except by changing it manually). Our model is much more general in this regard and can simulate the saturation and its change during the event. Obviously, we have to know the saturation (i.e. ground water level) at the start of the simulation (initial condition).

Anyway, our end member assumptions (completely dry or completely saturated in the beginning and evolving from there on) is still much better than the best assumption in non-porous models (completely dry or completely saturated during the whole event). We do not understand how the referee can see an upside of non-porous models in this regard.

More important is the actual geometry and vertical accelerations in the slide.

Vertical accelerations in the slide require a three-dimensional model with dilatancy. This is clearly visible in the experiments of Viroulet et al. (2013) and well reflected in our simulations. Similar experiments show similar flow behaviour with a large frontal vortex that cannot be described by layered or depth-integrated models. This speaks again for our model. The correct consideration of the terrain geometry is also very difficult with depth-integrated flow models (e.g. Bouchut and Westdickenberg, 2004; Rauter and Tuković, 2018). To our knowledge, no depth-integrated landslide tsunami model incorporates the curvature terms that are at minimum required for an accurate landslide simulation in complex terrain (except maybe SHALTOP, if coupled with a tsunami model; Kuo et al., 2009). This is actually a big concern of many scientists and will require substantial attention in the upcoming years. The correct geometrical representation of impact craters and breaking waves requires three-dimensional models as well. All of these arguments underline the upsides of our model.

How can the authors conclude based on a single simulation of Lake Askja, how predictive their model would be for other events such as AK ? Other successful work on this and other events, not cited by the authors, using a state of the art landslide tsunami model, did not require making ad-hoc assumptions or iterating on parameters as they suggest. Granted some of the other successful landslide tsunami models do not include dilatancy effects, which may be important to model the landslide in its underwater phase, but based on published results, may not be important in many cases to predict the main tsunami characteristics. Zengaffinen et al. (2020) modeled AK using advanced porous media rheologies and comparing to the simpler rheology used by Grilli et al. (2019) concluded that the latter was sufficient to reproduce the observed tsunami features.

We conclude that our model will predict other events too, because we rely on very basic physical phenomena and a direct simulation of many effects to predict complex behaviour. The Navier-Stokes Equations (and the OpenFOAM implementation) has

been evaluated for many cases. The same stands for the $\mu(I)$ -rheology and the drag function of Ergun (1952), which are very basic equations which should be accurate in a wide range of situations and regimes. Regardless, we agree that more work would be needed in the future to gain more experience, but this should be left to future studies.

We test this with two very different cases in this work (as you mention yourself, a very small one and a real scale event). Further, parts of the applied model have been tested for many cases. The relations that we assemble to our unified model, that are the Navier-Stokes Equations, the particle pressure equation of Johnson and Jackson (1987) and the extension for dynamic dilatancy following the $\phi(I)$ -curve Pouliquen et al. (2006), the drag law of Ergun (1952), the viscosity model of the $\mu(I)$ -rheology (Jop et al., 2006), are well established and validated for many different cases. There is no reason to think that these basic relations work well for Lake Askja and all previous studies, but not for Anak Krakatau. Further, and as mentioned before, components of the model have been tested independently before for landslide tsunamis (Romano et al., 2020; Chen et al., 2020; Rauter, 2021; Rauter et al., 2021).

Zengaffinen et al. (2020) did not use porous media rheologies and the parameters that they used were optimized to fit the wave signal at multiple gauges. "Porous" is not mentioned a single time in this publication.

Unfortunately, we cannot assess this issue for Grilli et al. (2019) because the material parameters are not mentioned in the paper. However, the application of a (among others) "high density fluid" indicates strongly that a parameter fitting was done, as the viscosity differs strongly with size as shown above.

Finally we would like to suggest to compare the video material of Grilli et al. (2019) with our video material. This shows very well the rich details that our model is able to provide.

For Lake Askja, authors do not detail the event in introduction ? No background is provided and this reviewer has unsuccessfully looked for a single georeferenced map with bathy/topo contour in the paper; and also one indicating where this lake might be. In tsunami propagation and coastal impact modeling, the main source of information is the bathy-topo map.

We refer to our previous publication, Gylfadóttir et al. (2017) where all the mentioned information can be found. Anyway, we added a georeferenced map and a map with bathy/topo contours to the supplementary materials.

L76: of constitutes -> of constituents ?

Thank you. Fixed.

The model used has many parameters (Table 1) that need to be measured or esti-

mated/selected with a realistic value ? How do they know those for a future event ? Some of the many parameters in Table 1 seem arbitrarily selected without justification ? No sensitivity study was made by invoking the great computational cost. This significantly weakens the present results.

The model parameters are mostly material parameters. We discuss in depth the selection of every single parameter in the method section. As the material parameters are widely intrinsic material parameters, they can be far easier estimated for future events than with previous methods. If there are any non-intrinsic material parameters we discuss their choice in detail and we provide estimations for such parameters based on e.g. the length scale. This is for example the case for ν_{\max} , see Rauter (2021).

We further added a rigorous sensitivity study to supplementary materials.

In Figs. 3 and 4, results of the model are OK but not particularly striking. Essentially the first wave is well captured which is mostly due to inertial entry of the material into the water. Later phases underwater are much less tsunamigenic. Other models with simpler physics would do as well on glass bead experiments (see Grilli et al., 2017; Zhang et al., 2021a,b).

We cannot follow this comment, as Figure 4 shows a very good fit, especially in comparison with previous three-dimensional methods Viroulet et al. (2016); Hoße et al. (2019); Si et al. (2018b). The same is the case for Figure 3. Notably, we achieved this without fitting of any material parameters.

We do not see such good comparisons in the suggested references for subaerial landslide tsunamis.

Fig. 5 could have bathy/topo contours on it. Show field results for landslide first before modeling. Hard to understand Fig. 5 out of context of model.

Figure 5 becomes very cluttered with contour lines. Field results are shown as a black line in all figures. We cannot add an additional figure just for the field results alone due to the limitation to nine figures.

Authors write: “To this end, we note that other recent modelling studies could match parts of the wave but not the entire event chain and not with a single set of parameters.” Be specific please. Which other studies are your referring to ? There are many recent modeling studies where both the landslide and the waves are well modeled for similar lab benchmarks. And also for more demanding cases in 3D with a higher entry velocity of the slide (e.g., Zhang et al. 2021b).

We think this is true for the relevant studies for our benchmark (Clous and Abadie, 2019; Si et al., 2018b; Viroulet et al., 2016). However, Si et al. (2018b) might be an exception and they use a very similar model to ours. Therefore we removed this sentence.

We want to note that the slide velocity in Zhang et al. (2021b) was determined by a boundary condition and that no figure of the deposition is shown.

As expected in most NS models, the authors observe numerical diffusion and, as waves propagate the accuracy is less. This would be a problem in using this model for actual tsunami propagation and hazard assessment where wave elevations are important.

This is a repetition of criticism, which we answered above (page 13-14).

The authors note: “Constitutive parameters for the rock slide are highly uncertain but can be estimated from observations and comparable materials”. If this is not model parameter fitting ?

Our parameters are material (aka. constitutive) parameters and are meant to be fitted to a material, not to an event.

An "effective viscosity", just to name one example of the parameters in the referees references, is not constant for a material but will depend on the situation and the event and has to be fitted to such. This is explained and underlined with references extensively above. Our parameters are meant to describe the material, e.g. friction coefficient or particle diameter. They are not only constant for various situations but also inferable from very simple laboratory tests.

This is conceptualised and very well described by Lambe (1973).

Notably, there is still some influence of the length scale for slides with a volume above $500\,000\text{ m}^3$ (Scheidegger, 1973; Lucas et al., 2014). However, this dependence is much smaller than any previous dependence in the references of the referee. This is well known in the community and discussed in the manuscript.

In the lake, the main tsunami generation has occurred for $t < 45\text{s}$, when the slide has just penetrated the water. Hence details of the slide underwater beyond that may not be tsunamigenic. However, one needs bathymetry to understand this wave generation. Fig. 7 should have labels on axes.

We also assume that details of the side underwater may not be important for the tsunamigenic potential but more investigations will be required to evidence this assumption.

The bathymetry is shown as a relief in the manuscript and as contour lines and in 3D in supplementary materials.

Labels provide no important information in the case of Fig. 7 and were thus removed before submission. This will allow to show the figures bigger. The diagram axes represent solely physical space and the respective scaling is indicated by a small scale within the diagrams to save space. The relation (offset from origin) to the geographic reference system plays no role and is thus not shown.

Fig. 7c shows a simple wavemaker motion would work likely well here meaning a simpler rheology would likely work well.

A simple wave source worked to generate the observed wave, as shown by Gylfadóttir et al. (2017). However, the motion that produces the wave is unknown and had to be found with an optimization study as clearly laid out in the discussed work. The same would be the case for simplified rheologies or even simple scaling relations (e.g. the relation of Fritz (2002) $\eta/H = a Fr^b S^c$). Further, this would give us no insights into the wave generation and the slide event. In contrast, we simulate the wave source and the optimization of the wave source to the inundation is not required. The starting points are completely different, and so is the problem that we aim to solve, although they might come to a similar result.

Missing from Fig. 8, to have a truly complete multi-model comparison would be results of a two-layer non-hydrostatic model such as NHWAVE (open source on github and used by many groups). In discussion of slide acceleration, it would be interesting to compute and show the location vel and accel of the slide center of mass. Again having bathy-topo contours on the figure would be helpful.

We choose GeoClaw, which is also a two-layer, non-hydrostatic model, to have comparisons to the best possible (due to wave source optimisation) depth-integrated flow model. We are certain that we could not do better with NHWAVE. The included physics is basically the same but we are better skilled with GeoClaw and we conducted an optimization to find the optimal wave source. We do not find the location, velocity and acceleration of the centre of mass interesting and do not include it into the manuscript. Anyway, we added this data to the supplementary materials.

Slide runout is shown on Fig. 8, but is there data on deposit thickness that could be compared to model results ? Fig. 8 should have different line color/type for lake outline and maximum observed inundation for better readability.

Data on the deposition is available but the material in the simulation did not settle yet. Therefore such a comparison is not straight forward and we decided to use the outline as a comparison. This approach is common in landslide research.

We updated colour/line type in Fig. 8.

The authors state: “The documented inundation is matched well by the numerical model across the periphery of the lake, as shown in Fig. 8.”, but in this reviewer’s opinion, runup does not appear to be particularly good in the model results, for a number of areas along the coast ? Is the model adequate for runup (the authors indicate a 1.25 m accuracy due to discretization but this is not really a moving shoreline algorithm) ? Other recent works on nearfield runup around AK showed better agreement of model

results and field data. Clearly coupling to a better nearshore model all around the lake could help here.

Moving shoreline algorithms come from depth-integrated flow models and this is not required for Navier-Stokes type models.

We added a quantified comparison in the version of the manuscript and show that the error is comparable with depth-integrated fully optimized results.

As the inundation of the depth-integrated simulations is algorithm-optimized with hundreds of iterations, we do not think that any depth-integrated simulation could do better. Higher grid resolution might have helped, but this is beyond the scope of this paper.

The authors state: “We conclude from this observation that the wave generation is represented well by the model but that the wave propagation involves some degree of artificial damping.” This is true of many NS models that are too numerically dissipative. A sensitivity analysis to mesh and other model parameters might have helped elucidate this, but they claim it is too costly. Also using a two-layer non-hydro model such as NHWAVE to compare to would have been interesting.

We corrected an error that led to overly strong numerical diffusion and further investigated this issue with a grid coarsening study. The numerical diffusion could be further reduced and the grid coarsening study revealed no substantial signs of numerical diffusion. Under these circumstances we come to the conclusion that artificial wave damping, although present in every model, is not a particularly strong problem in the presented model and simulations. We changed the respective part of the manuscript.

Where is the comparison of the slide runout in model and field measurements ? What do you mean by “orographic right branch” ?

The comparison is in Fig. 7. Orographic right is defined as right if looking in flow direction.

The authors claim: “Many reasons for uncertainty”. All the more important to perform a sensitivity analysis to assumed model parameters such as packing and initial stresses ? This could have perhaps been done on an idealized 2D slice-slide.

We performed a sensitivity analysis of the 2D simulation and added it to supplementary materials. We do not think that a sensitivity study of a 2D slice will tell us much about branching of the slide. We conducted some studies on 2D slices during the review of this manuscript but found that their significance for the real case is very low. The lateral spread of both the slide and the wave are too important in the presented case. Therefore and because the work is already very long, we decided against including a sensitivity study on 2D slices.

The full 3D model is currently too expensive for a rigorous sensitivity analysis at the full resolution. We consider such a study on reduced resolutions (a mesh coarsening study revealed that these results are still quite useful) for the future but this is out of the scope for the current work.

The authors claim: “we anticipate that a landslide with a higher friction angle would have reached the same run-out” Why not trying it out ?

We tried out a reduced friction angle during this review as we found this case more interesting, especially considering the underestimation of some inundation on the southern shore. As we expected, the runout is very similar.

As mentioned above, a sensitivity study on the 3D case is out of the scope of this work.

Because of the lack of a sensitivity analysis, many conclusions are based on conjectures about what the model could do.

We added a sensitivity study to the supplementary materials. These back up our observations related to the importance of various landslide parameters and on their tsunami-genic strength.

The authors note: “These results render it unlikely that the deposit is a reliable indicator for the tsunamigenic potential.” This reviewer agrees, which likely justifies that using a simpler model of the granular flow would be sufficient to predict the main tsunami features as confirmed in many studies (not cited here).

We appreciate that the reviewer agrees with us on this point. Simpler models can provide some features as we showed in the previous study of Gylfadóttir et al. (2017). Our model provides much more features and details and does so with a very unique and better parametrisation. We also agree that there are many studies (too many?) focusing on simpler models. This only highlights the uniqueness and value of our study.

L190: manifold -> many ?

Not changed.

L213: the authors think the answer is in ever more complex models, but only a sensitivity analysis and complete comparison to other model classes will show which part of the physical processes matters most for tsunami generation. This reviewer believes this will likely not be the details of the flow within the porous medium.

To make such a comparison possible, it is important to have multiple models of various complexities.

Notably, the structure of our model is very simple. In fact, it is even simpler than depth-integrated models. However, it is computationally expensive which we make very clear.

Computationally expensive and complex are two different things. We think that many so called simplifications (e.g. depth-integration) are in fact optimisations that make the model more efficient but also more complex in their mathematical derivation and resulting formulation. Some physics may be lost, which is for example shown by the fact that Ma et al. (2015) need to assume the pore pressure in the slide, which we directly simulate. Other models, e.g. Clous and Abadie (2019) struggle with the same problem. Our approach to directly simulate as many aspects as possible adds substantial insights and reliability to the model and justifies the added computational cost. It is the porous medium which enables us to do so. We stress this fact strongly in the article.

As mentioned before, we added a sensitivity analysis.

Authors also do not discuss the exact nature of the debris flow in the field event and whether assuming it was made of uniformly distributed 10 cm particles was realistic ? Likely not. Was there a sub-bottom survey of the deposits made ? Instead the authors wonder about the role played by air expulsion, which is likely negligible for slide dynamics and wave generation. The authors are trying to find guidance for this from their extremely small scale lab experiments where many processes do not scale correctly (such as air bubbles, surface tension, viscosity).

We agree that the assumption of homogenous (uniform) slide material is somehow unrealistic, however all other models we are aware of are affected by this issue in a similar or even worse manner. Our model allows for a non-homogenous (i.e. non uniformly distributed or polydisperse mixture) and would have a strong upside in such cases need to be modelled.

We discuss the polydispersity of the real event in the method section and more in depth in the references cited there (Barker et al., 2021; Festa et al., 2015). Further this fact was discussed above.

The authors state: “Our model performs very well compared with the depth-integrated models. This is remarkable, because the three-dimensional model was not optimized to fit the observations” Nothing quite remarkable here, as this was an oversimplified depth-integrated model for a sliding block ! It was expected that the BM would not perform well. The BM model performs better in the far-field which confirms there is likely too much dissipation in the 3D model. The authors agree with that on L 243. So, the model can’t be trusted at some distance from the source which is not quite known ! Not that useful a feature in tsunami hazard assessment studies. . .

We agree that the depth-integrated model is oversimplified which is why we apply a three-dimensional model. Anyway, the wave source of the depth-integrated model was optimized to fit the inundation and should thus represent the absolute best result achievable with depth-integrated flow models.

The claim that the present resolution is better than in some other studies is not a justification in itself as it all depends on model type, physics, and numerical methods. What is missed in runup and inundation is much greater than the cell size. Is the shoreline algorithm good enough in the model ?

We agree with the referee and added a grid coarsening study to supplementary materials. It revealed that the resolution is not a substantial problem and that we can even get useful results on coarser grids.

Our resolution is better than other studies with the same model, method and numerics (OpenFOAM, multiphase-Navier-Stokes, Counter Gradient Transport and MULES). This shows that our simulations are state of the art. That is an important fact and we are happy to repeat it here.

The model directly simulates inundation and there is no such thing as a "shoreline algorithm" required. This is another upside of our model. Thank you for mentioning this.

L:229,235,249: “This is remarkable, because”, “the inundation allows remarkable conclusions on the generation event”: “ and that the results are of remarkable quality”: there are really too many, not quite appropriate, superlatives in this paper.

We agree with the reviewer provided softer formulation here.

L 239: The excuse of high-computational cost for not doing any parameter or mesh sensitivity studies is really not a good excuse when one claims to have a new breakthrough in modeling !

We agree with the referee and added both, a sensitivity study and a mesh refinement study to supplementary materials for the 2D case and a mesh refinement study for the 3D case.

L 254: “and the application of a sub grid turbulence model seems appropriate” Do they authors mean they claim to solve NS equations at that scale with a 2.5 m mesh and no turbulence closure modeling ? It seems in Table 1, they have multiplied without real justification the standard water viscosity in both the scale model and field problems by about 15 ! The small scale lab modeling may have sufficient viscosity owing to its low Re value, but for the field scale event, unless a turbulence model is used, it is likely

that simulations are not sufficiently dissipative around the wave generation area and the model solves a version of Euler equations. No discussion of this is made.

We mixed up the viscosity of air ($1.48 \cdot 10^{-5}$) and water ($1 \cdot 10^{-6}$) and we are very sorry about this error. Thank you very much for finding this.

Therefore we repeated all simulations with the corrected viscosity but all other parameters unchanged. This had a small influence on the permeability of the slide and therefore its velocity. The slide front velocity fits now better to the experiment. Further the overestimation of the wave diffusion is reduced. In fact, the numerical diffusion could be reduced even further to a point where the first wave crest is equally well matched at the first wave gauge and the last wave gauge. We also want to note the downsides of this change: The final deposition of the small scale case fits the experiment worse now. The sub grid turbulence model (see below) would improve this aspect and restored the previous accuracy. We discuss this in the new version of the manuscript.

We evaluated the application of a sub grid turbulence model and we added the respective simulation to the supplementary materials. We further had some experience with sub grid turbulence models in landslide tsunami simulations from our previous work (Rauter et al., 2021). We concluded that it is not really important for the results of interest while being remarkably complex. Further, the structure/form of sub grid turbulence model in multiphase/multicomponent/porous granular flows is still highly debated and we are not experts in this topic. It seem absolutely not clear how these concepts should interact and there are many researchers looking into this detail. Therefore we decided to keep the sub grid turbulence out of this work for now.

However, we added a very simple version of a sub grid turbulence model to a simulation that we show in supplementary materials. We wanted to avoid such a rudimentary solution (no consideration of multiphase, free water surface, porosity, etc.) in the main article and hope that we will find a better solution in future investigations.

L257: “The lake Askja event is challenging for a three-dimensional landslide tsunami model because the wave amplitude is very shallow in relation to the lake size”. I can’t really understand the meaning of comparing the maximum surface elevation to lake lengths. typical parameters in wave models are nonlinearity and dispersions that combine surface elevation with depth and wavelength. Depth is never mentioned nor discussed at any point before ? Three-dimensional models have been used in many similar contexts. I suspect that a 2.5 m cell might not be necessary for a correct tsunami modeling with a 3D non-hydrostatic model in this case but maybe 10-20 m with depth layers of various thickness. Of course those models are not even mentioned or considered here.

The required mesh size is related to the wave amplitude and the size of the mesh is related to the lake extension. Therefore the relation of wave amplitude to lake extension is a good indicator for the required number of cells and thus the numerical cost.

Depth is, in contrast to depth-integrated and layered models, included in the geometry and there is not much to discuss about. Dispersion and non-linearities, appear naturally in the simulation without including them explicitly in the model equations. The three-

dimensional model includes also further effects like wave breaking that require explicit modelling in depth-integrated or layered models. This is one of the many positives of complete three-dimensional models.

Layered and non-hydrostatic models are mentioned in our manuscript, especially in form of the respective review papers. They have of course been considered for this project but we decided to go full 3D for the reasons mentioned above and in the manuscript.

Model coupling is of course a way to deal with these issues and in addition to the Lovholt et al. (2008) paper, they could refer to a suite of earlier and later works that have perfected the use of model-coupling of 3D/2D-tsunami propagation for hypothetical or actual events. Ignoring a lot of the work done outside of their collaborator group is strange. Abadie et al., (2012) is cited as a “promising approach”. Well this was 10 years ago work that has had many improvement, in particular regarding the landslide tsunami models (e.g., Tappin et al., 2014; Grilli et al., 2015,2017; Schambach et al., 2019, 2020, 2021)

As mentioned above, none of the suggested publications contain 3D (Navier-Stokes type) to 2D (Shallow water type) coupling. Further, none of the suggested publications contains a Navier-Stokes type landslide model. Citing other references than Løvholt et al. (2008) and Abadie et al. (2012) in this context, seems therefore unsubstantiated. More details are discussed in the introduction.

L 265: “This work represents a large step forward in modelling of subaerial landslide tsunamis.” “demonstrates remarkable accuracy of wave kinematics and landslide runout” In this first statement in the conclusion, the authors quite pleased with their work summarize the flavor of a paper that this reviewer believes is not sufficiently self-critical. This conclusion should be demonstrated by results and made by readers, but results presented here are not particularly impressive (considering a model that needs to run for 100s of hrs on 100s of processors) and do not demonstrate in this reviewer’s opinion a step forward in subaerial tsunami modeling. There are a few other existing models with just some aspects of the physics simplified, as compared to the present model, that would very likely perform as well on similar lab experiments or even better on field case studies of landslide tsunamis a they would not damp waves in this manner in the far-field.

L267: “without the need for optimized parameters” Authors seem to imply all other modelers optimize their parameters to match observation ? It is not the case in references cited above, whereas the present model has quite a few parameters set to arbitrary values.

This is somehow a repetition of criticism, which we answered above.

“However, the simulations are computationally expensive and wave damping can be

observed in the far field due to the limited grid resolution.”. What is the point in using a model one cannot afford which, besides, overdamps the wave in an unknown grid-dependent manner ? There are other published and well-validated models with just slightly simpler physics that are very efficient and can be and have been used for meaningful hazard assessment cases.

We can afford the model as shown in the manuscript. The damping is limited and well under control, see Fig. 3. The numerical cost is a drawback of the model compared to depth-integrated/layered models, that is correct. However, in comparison to those models (which we also used extensively in many studies), we provide substantially more insights, simpler parametrisations and better results. More details are discussed in the introduction.

L263: “Full three-dimensional methods are relatively rare in the tsunami community”. These have been used whenever necessary, but not over-used when not necessary. This is where the art of the modelers comes in to decide about the best class of models/physics to apply to a given problem. In any cases these aspects have not been exhaustively reviewed and discussed in this paper.

This is somehow a repetition of criticism, which we answered above.

L316: “sub grid turbulence that might play a role at high Reynolds numbers” The field case in the lake does have a large Reynolds number !

Yes, and sub grid turbulence models might play a role as mentioned in the manuscript. We evaluated the role of a subgrid turbulence model for the small scale experiment in the supplementary materials. The additional subgrid turbulence model leads to a difference in local small scale flow patterns but the macroscopic kinematics and wave signals are not affected. Subgrid turbulence models are still a topic of research, especially in combination with porous materials and transient behaviour. There is no readily applicable and generally agreed on model. Therefore we do not want to put this aspect in the centre of our investigations, there is in fact a large community of researchers who focus on this topic entirely.

To conclude, this reviewer finds the presented work to have merit and interest in the context of landslide tsunami modeling. With substantial revisions, it could likely be published in a relevant engineering or geophysics journal. However, the reviewer believes that the work does not raise to the level and general audience interest of a Nature Communication paper and, considering the concerns expressed above, does not believe it can be revised in a suitable manner.

Thank you for your review. We have to be especially thankful for spotting the error in the water viscosity. This error was addressed and corrected without substantial influence

on the results.

We addressed all of your concerns in the revised manuscript or in the answers given here.

Regarding level of detail to reproduce the work, there are many missing information and data and no electronic repository is provided. The authors asked to be contacted for information.

This is correct and we checked with the guidelines of Nature Communications before submission and it is an acceptable procedure. Anyway, we would like to give some reasons to do it this way.

Regarding the data: The data is related to a real event in a highly frequented tourist area and as such there could have been sever loss of life if it happened at a different time. Therefore the data, coming from an official investigation of IMO is delicate and sensitive and we could not outright publish it. However, IMO was very cooperative and we are sure they will cooperate with any serious researcher who requests access to the data.

Regarding the code: We would like to publish the code and we are working on it, however, this will take time and we cannot postpone the publication for so long. We intent to publish the model as part of the official OpenFOAM version. We did this before with the avalanche module (<https://develop.openfoam.com/Community/avalanche>) and it showed to be the best way to provide a reliable and updated code basis to users.

References

- S. Abadie, D. Morichon, S. Grilli, and S. Glockner. Numerical simulation of waves generated by landslides using a multiple-fluid navier–stokes model. *Coastal Engineering*, 57(9):779–794, 2010. doi: 10.1016/j.coastaleng.2010.03.003.
- S. Abadie, A. Paris, R. Ata, S. Le Roy, G. Arnaud, A. Poupardin, L. Clous, P. Heinrich, J. Harris, R. Pedreros, et al. La Palma landslide tsunami: calibrated wave source and assessment of impact on French territories. *Natural Hazards and Earth System Sciences*, 20(11):3019–3038, 2020. doi: 10.5194/nhess-20-3019-2020.
- S. M. Abadie, J. C. Harris, S. T. Grilli, and R. Fabre. Numerical modeling of tsunami waves generated by the flank collapse of the Cumbre Vieja volcano (La Palma, Canary Islands): Tsunami source and near field effects. *Journal of Geophysical Research*, 117 (C05030), 2012. doi: 10.1029/2011JC007646.
- T. Barker, M. Rauter, E. Maguire, C. Johnson, and J. Gray. Coupling rheology and segregation in granular flows. *Journal of Fluid Mechanics*, 909, 2021. doi: 10.1017/jfm.2020.973.

- A. S. Baumgarten and K. Kamrin. A general fluid–sediment mixture model and constitutive theory validated in many flow regimes. *Journal of Fluid Mechanics*, 861: 721–764, 2019. doi: doi:10.1017/jfm.2018.914.
- F. Bouchut and M. Westdickenberg. Gravity driven shallow water models for arbitrary topography. *Commun. Math. Sci.*, 2(3):359–389, 09 2004. doi: 10.4310/CMS.2004.v2.n3.a2.
- F. Bouchut, E. D. Fernández-Nieto, E. Koné, A. Mangeney, and G. Narbona-Reina. A two-phase solid-fluid model for dense granular flows including dilatancy effects: comparison with submarine granular collapse experiments. In *EPJ Web of Conferences*, volume 140, page 09039. EDP Sciences, 2017.
- A. Bougouin, R. Paris, and O. Roche. Impact of Fluidized Granular Flows into Water: Implications for Tsunamis Generated by Pyroclastic Flows. *Journal of Geophysical Research: Solid Earth*, page e2019JB018954, 2020. doi: 10.1029/2019JB018954.
- F. Boyer, É. Guazzelli, and O. Pouliquen. Unifying suspension and granular rheology. *Physical Review Letters*, 107(18):188301, 2011. doi: 10.1103/PhysRevLett.107.188301.
- G. K. Bullard, R. P. Mulligan, A. Carreira, and W. A. Take. Experimental analysis of tsunamis generated by the impact of landslides with high mobility. *Coastal Engineering*, 152:103538, 2019. doi: 10.1016/j.coastaleng.2019.103538.
- C. S. Campbell. Granular material flows—An overview. *Powder Technology*, 162(3): 208–229, 2006. doi: 10.1016/j.powtec.2005.12.008.
- J. Chauchat, Z. Cheng, T. Nagel, C. Bonamy, and T.-J. Hsu. Sedfoam-2.0: a 3-d two-phase flow numerical model for sediment transport. *Geoscientific Model Development*, 10(12):4367, 2017. doi: 0.5194/gmd-10-4367-2017.
- F. Chen, V. Heller, and R. Briganti. Numerical modelling of tsunamis generated by iceberg calving validated with large-scale laboratory experiments. *Advances in Water Resources*, 142:103647, 2020. doi: 10.1016/j.advwatres.2020.103647.
- Z. Cheng, T.-J. Hsu, and J. Calantoni. Sedfoam: A multi-dimensional eulerian two-phase model for sediment transport and its application to momentary bed failure. *Coastal Engineering*, 119:32–50, 2017. doi: 10.1016/j.coastaleng.2016.08.007.
- L. Clous and S. Abadie. Simulation of energy transfers in waves generated by granular slides. *Landslides*, 16(9):1663–1679, 2019. doi: 10.1007/s10346-019-01180-0.
- R. Courant, K. Friedrichs, and H. Lewy. Über die partiellen Differenzgleichungen der mathematischen Physik. *Mathematische Annalen*, 100(1):32–74, 1928.
- S. Ergun. Fluid flow through packed columns. *Chemical Engineering Progress*, 48:89–94, 1952.

- J. Fannin. Karl terzaghi: from theory to practice in geotechnical filter design. *Journal of geotechnical and geoenvironmental engineering*, 134(3):267–276, 2008.
- A. Festa, K. Ogata, G. A. Pini, Y. Dilek, and G. Codegone. Late Oligocene–early Miocene olistostromes (sedimentary mélanges) as tectono-stratigraphic constraints to the geodynamic evolution of the exhumed Ligurian accretionary complex (Northern Apennines, NW Italy). *International Geology Review*, 57(5-8):540–562, 2015.
- A. Franco, J. Moernaut, B. Schneider-Muntau, M. Aufleger, M. Strasser, and B. Gems. Lituya bay 1958 tsunami “ detailed pre-event bathymetry reconstruction and 3d-numerical modelling utilizing the cfd software flow-3d. *Natural Hazards and Earth System Sciences Discussions*, 2019:1–34, 2019. doi: 10.5194/nhess-2019-285. URL <https://www.nat-hazards-earth-syst-sci-discuss.net/nhess-2019-285/>.
- A. Franco, J. Moernaut, B. Schneider-Muntau, M. Strasser, and B. Gems. Triggers and consequences of landslide-induced impulse waves–3d dynamic reconstruction of the taan fiord 2015 tsunami event. *Engineering Geology*, page 106384, 2021.
- H. M. Fritz. *Initial phase of landslide generated impulse waves*. PhD thesis, ETH Zurich, 2002.
- R. Gabl, J. Seibl, B. Gems, and M. Aufleger. 3-D numerical approach to simulate the overtopping volume caused by an impulse wave comparable to avalanche impact in a reservoir. *Natural Hazards and Earth System Sciences*, 15(12):2617–2630, 2015. doi: 10.5194/nhess-15-2617-2015.
- P. Gauer, T. J. Kvalstad, C. F. Forsberg, P. Bryn, and K. Berg. The last phase of the storegga slide: simulation of retrogressive slide dynamics and comparison with slide-scar morphology. In *Ormen Lange—an Integrated Study for Safe Field Development in the Storegga Submarine Area*, pages 171–178. Elsevier, 2005. doi: 10.1016/B978-0-08-044694-3.50018-4.
- P. Gauer, A. Elverhoi, D. Issler, and F. V. De Blasio. On numerical simulations of subaqueous slides: back-calculations of laboratory experiments of clay-rich slides. *NORSK GEOLOGISK TIDSSKRIFT*, 86(3):295–300, 2006.
- D. L. George, R. M. Iverson, and C. M. Cannon. New methodology for computing tsunami generation by subaerial landslides: Application to the 2015 Tyndall Glacier landslide, Alaska. *Geophysical Research Letters*, 44(14):7276–7284, 2017. doi: 10.1002/2017GL074341.
- G. Gisler, R. Weaver, and M. L. Gittings. SAGE calculations of the tsunami threat from La Palma. *Science of Tsunami Hazards*, 24(4):288–312, 2006.
- S. T. Grilli, C. O’reilly, J. C. Harris, T. T. Bakhsh, B. Tehranirad, S. Banihashemi, J. T. Kirby, C. D. Baxter, T. Eggeling, G. Ma, et al. Modeling of smf tsunami hazard along the upper us east coast: detailed impact around ocean city, md. *Natural Hazards*, 76(2):705–746, 2015.

- S. T. Grilli, M. Shelby, O. Kimmoun, G. Dupont, D. Nicolsky, G. Ma, J. T. Kirby, and F. Shi. Modeling coastal tsunami hazard from submarine mass failures: effect of slide rheology, experimental validation, and case studies off the us east coast. *Natural hazards*, 86(1):353–391, 2017.
- S. T. Grilli, D. R. Tappin, S. Carey, S. F. Watt, S. N. Ward, A. R. Grilli, S. L. Engwell, C. Zhang, J. T. Kirby, L. Schambach, et al. Modelling of the tsunami from the december 22, 2018 lateral collapse of anak Krakatau volcano in the Sunda Straits, Indonesia. *Scientific reports*, 9(1):1–13, 2019.
- S. S. Gylfadóttir, J. Kim, J. K. Helgason, S. Brynjólfsson, Á. Höskuldsson, T. Jóhannesson, C. B. Harbitz, and F. Løvholt. The 2014 Lake Askja rockslide-induced tsunami: Optimization of numerical tsunami model using observed data. *Journal of Geophysical Research: Oceans*, 122(5):4110–4122, 2017. doi: 10.1002/2016JC012496.
- C. B. Harbitz, S. Glimsdal, F. Løvholt, V. Kveldsvik, G. K. Pedersen, and A. Jensen. Rockslide tsunamis in complex fjords: from an unstable rock slope at Åkerneset to tsunami risk in western Norway. *Coastal Engineering*, 88:101–122, 2014. doi: 10.1016/j.coastaleng.2014.02.003.
- C. J. Heerema, P. J. Talling, M. J. Cartigny, C. K. Paull, L. Bailey, S. M. Simmons, D. R. Parsons, M. A. Clare, R. Gwiazda, E. Lundsten, et al. What determines the downstream evolution of turbidity currents? *Earth and Planetary Science Letters*, 532:116023, 2020. doi: 10.1016/j.epsl.2019.116023.
- V. Heller, W. H. Hager, and H.-E. Minor. Scale effects in subaerial landslide generated impulse waves. *Experiments in Fluids*, 44(5):691–703, 2008. doi: 10.1007/s00348-007-0427-7.
- J. Heyman, R. Delannay, H. Tabuteau, and A. Valance. Compressibility regularizes the $\mu(I)$ -rheology for dense granular flows. *Journal of Fluid Mechanics*, 830:553–568, 2017. doi: 10.1017/jfm.2017.612.
- B. Higman, D. H. Shugar, C. P. Stark, G. Ekström, M. N. Koppes, P. Lynett, A. Dufresne, P. J. Haeussler, M. Geertsema, S. Gulick, et al. The 2015 landslide and tsunami in Taan Fiord, Alaska. *Scientific Reports*, 8(1):1–12, 2018.
- J. Horrillo, A. Wood, G.-B. Kim, and A. Parambath. A simplified 3-d Navier-Stokes numerical model for landslide-tsunami: Application to the Gulf of Mexico. *Journal of Geophysical Research: Oceans*, 118(12):6934–6950, 2013.
- L. Hoße, M. Rauter, and F. Løvholt. Numerical simulation of impulse waves using primitive flow models. In *Geophysical Research Abstracts*, volume 21, 2019. URL https://www.researchgate.net/publication/332423112_Numerical_Simulation_of_Impulse_Waves_using_Primitive_Flow_Models_Preliminary_Results.

- Y.-x. Hu, Z.-y. Yu, and J.-w. Zhou. Numerical simulation of landslide-generated waves during the 11 october 2018 baige landslide at the jinsha river. *Landslides*, 17(10): 2317–2328, 2020.
- J. Hunt, D. Tappin, S. Watt, S. Susilohadi, A. Novellino, S. Ebmeier, M. Cassidy, S. Engwell, S. Grilli, M. Hanif, et al. Submarine landslide megablocks show half of anak Krakatau island failed on december 22nd, 2018. *Nature Communications*, 12 (2827):1–15, 2021. doi: 10.1038/s41467-021-22610-5.
- P. C. Johnson and R. Jackson. Frictional–collisional constitutive relations for granular materials, with application to plane shearing. *Journal of Fluid Mechanics*, 176:67–93, 1987. doi: 10.1017/S0022112087000570.
- P. Jop, Y. Forterre, and O. Pouliquen. A constitutive law for dense granular flows. *Nature*, 441(7094):727, 2006. doi: 10.1038/nature04801.
- F. Juretić. cfMesh user guide. Technical report, Creative Fields, Zagreb, 2015.
- J. Kafle, P. Kattel, M. Mergili, J.-T. Fischer, and S. P. Pudasaini. Dynamic response of submarine obstacles to two-phase landslide and tsunami impact on reservoirs. *Acta Mechanica*, 230(9):3143–3169, 2019. doi: 10.1007/s00707-019-02457-0.
- G.-B. Kim, W. Cheng, R. C. Sunny, J. J. Horrillo, B. C. McFall, F. Mohammed, H. M. Fritz, J. Beget, and Z. Kowalik. Three dimensional landslide generated tsunamis: Numerical and physical model comparisons. *Landslides*, pages 1–17, 2019a. doi: 10.1007/s10346-019-01308-2.
- J. Kim, F. Løvholt, D. Issler, and C. F. Forsberg. Landslide material control on tsunami genesis–The Storegga slide and tsunami (8,100 years BP). *Journal of Geophysical Research: Oceans*, 124(6):3607–3627, 2019b. doi: 10.1029/2018JC014893.
- C.-Y. Kuo, Y. Tai, F. Bouchut, A. Mangeney, M. Pelanti, R. Chen, and K. Chang. Simulation of tsaoling landslide, taiwan, based on saint venant equations over general topography. *Engineering Geology*, 104(3-4):181–189, 2009.
- P.-Y. Lagrée, L. Staron, and S. Popinet. The granular column collapse as a continuum: validity of a two-dimensional Navier–Stokes model with a $\mu(I)$ -rheology. *Journal of Fluid Mechanics*, 686:378–408, 2011. doi: 10.1017/jfm.2011.335.
- T. Lambe. Predictions in soil engineering. *Géotechnique*, 23(2):151–202, 1973. doi: 10.1680/geot.1973.23.2.151.
- C.-H. Lee and Z. Huang. Multi-phase flow simulation of impulsive waves generated by a sub-aerial granular landslide on an erodible slope. *Landslides*, pages 1–15, 2020. doi: 10.1007/s10346-020-01527-y.
- E. K. Lindstrøm. Waves generated by subaerial slides with various porosities. *Coastal Engineering*, 116:170–179, 2016. doi: 10.1016/j.coastaleng.2016.07.001.

- P. L.-F. Liu, T.-R. Wu, F. Raichlen, C. E. Synolakis, and J. C. Borrero. Runup and rundown generated by three-dimensional sliding masses. *Journal of Fluid Mechanics*, 536:107–144, 2005. doi: 10.1017/S0022112005004799.
- F. Løvholt, G. Pedersen, and G. Gislér. Oceanic propagation of a potential tsunami from the La Palma Island. *Journal of Geophysical Research: Oceans*, 113(C9), 2008. doi: 10.1029/2007JC004603.
- F. Løvholt, G. Pedersen, C. B. Harbitz, S. Glimsdal, and J. Kim. On the characteristics of landslide tsunamis. *Philosophical Transactions of the Royal Society A*, 373(2053): 20140376, 2015. doi: 10.1098/rsta.2014.0376.
- A. Lucas, A. Mangeney, and J. P. Ampuero. Frictional velocity-weakening in landslides on earth and on other planetary bodies. *Nature communications*, 5(1):1–9, 2014. doi: 10.1038/ncomms4417.
- G. Ma, J. T. Kirby, and F. Shi. Numerical simulation of tsunami waves generated by deformable submarine landslides. *Ocean Modelling*, 69:146–165, 2013.
- G. Ma, J. T. Kirby, T.-J. Hsu, and F. Shi. A two-layer granular landslide model for tsunami wave generation: Theory and computation. *Ocean Modelling*, 93:40–55, 2015.
- MiDi. On dense granular flows. *The European Physical Journal E*, 14(4):341–365, 2004. doi: 10.1140/epje/i2003-10153-0.
- R. P. Mulligan, A. Franci, M. A. Celigueta, and W. A. Take. Simulations of landslide wave generation and propagation using the Particle Finite Element Method. *Journal of Geophysical Research: Oceans*, 125:e2019JC015873, 2020. doi: 10.1029/2019JC015873.
- OpenCFD Ltd. *OpenFOAM - The Open Source CFD Toolbox - User Guide*, 2004. URL <https://www.openfoam.com/documentation/user-guide/>. last checked: 20.01.2020.
- M. Pailha and O. Pouliquen. A two-phase flow description of the initiation of underwater granular avalanches. *Journal of Fluid Mechanics*, 633:115–135, 2009. doi: 10.1017/S0022112009007460.
- A. Panizzo, P. De Girolamo, M. Di Risio, A. Maistri, and A. Petaccia. Great landslide events in Italian artificial reservoirs. *Natural Hazards and Earth System Sciences*, 5: 733–740, 2005. doi: 10.5194/nhess-5-733-2005.
- A. Paris, E. A. Okal, C. Guérin, P. Heinrich, F. Schindel e, and H. H ebert. Numerical modeling of the June 17, 2017 landslide and tsunami events in Karrat Fjord, West Greenland. *Pure and Applied Geophysics*, 176(7):3035–3057, 2019. doi: 10.1007/s00024-019-02123-5.

- R. Paris, J. J. C. Bravo, M. E. M. González, K. Kelfoun, and F. Nauret. Explosive eruption, flank collapse and megatsunami at Tenerife ca. 170 ka. *Nature Communications*, 8:15246, 2017. doi: 10.1038/ncomms15246.
- A. Passalacqua and R. O. Fox. Implementation of an iterative solution procedure for multi-fluid gas-particle flow models on unstructured grids. *Powder Technology*, 213(1):174–187, 2011. doi: 10.1016/j.powtec.2011.07.030.
- M. Pastor, I. Herreros, J. F. Merodo, P. Mira, B. Haddad, M. Quecedo, E. González, C. Alvarez-Cedrón, and V. Drempevic. Modelling of fast catastrophic landslides and impulse waves induced by them in fjords, lakes and reservoirs. *Engineering Geology*, 109(1-2):124–134, 2008. doi: 10.1016/j.enggeo.2008.10.006.
- O. Pouliquen, C. Cassar, P. Jop, Y. Forterre, and M. Nicolas. Flow of dense granular material: towards simple constitutive laws. *Journal of Statistical Mechanics: Theory and Experiment*, 2006(07):P07020, 2006. doi: 10.1088/1742-5468/2006/07/P07020.
- S. P. Pudasaini. A general two-phase debris flow model. *Journal of Geophysical Research: Earth Surface*, 117(F3), 2012. doi: 10.1029/2011JF002186.
- R. S. Ramalho, G. Winckler, J. Madeira, G. R. Helffrich, A. Hipólito, R. Quartau, K. Adena, and J. M. Schaefer. Hazard potential of volcanic flank collapses raised by new megatsunami evidence. *Science advances*, 1(9):e1500456, 2015. doi: 10.1126/sciadv.1500456.
- M. Rauter. The compressible granular collapse in a fluid as a continuum: validity of a Navier-Stokes model with $\mu(J), \phi(J)$ -rheology. *Journal of Fluid Mechanics*, 915, 2021. doi: 10.1017/jfm.2021.107.
- M. Rauter and Ž. Tuković. A finite area scheme for shallow granular flows on three-dimensional surfaces. pages 184–199, 2018. doi: 10.1016/j.compfluid.2018.02.017.
- M. Rauter, A. Kofler, A. Huber, and W. Fellin. faSavageHutterFOAM 1.0: depth-integrated simulation of dense snow avalanches on natural terrain with OpenFOAM. *Geoscientific Model Development*, 11(7):2923–2939, 2018. doi: 10.5194/gmd-11-2923-2018.
- M. Rauter, T. Barker, and W. Fellin. Granular viscosity from plastic yield surfaces: The role of the deformation type in granular flows. *Computers and Geotechnics*, 122:103492, 2020. doi: 10.1016/j.compgeo.2020.103492.
- M. Rauter, L. Hoße, R. Mulligan, A. Take, and F. Løvholt. Numerical simulation of impulse wave generation by idealized landslides with OpenFOAM. *Coastal Engineering*, 165:103815, 2021. doi: 10.1016/j.coastaleng.2020.103815.
- J. Roenby, B. E. Larsen, H. Bredmose, and H. Jasak. A new volume-of-fluid method in OpenFOAM. In *VII International Conference on Computational Methods in Marine Engineering*. Nantes: International Center for Numerical Methods in Engineering, 2017.

- A. Romano, J. L. Lara, G. Barajas, B. Di Paolo, G. Bellotti, M. Di Risio, I. J. Losada, and P. De Girolamo. Tsunamis generated by submerged landslides: numerical analysis of the near-field wave characteristics. *Journal of Geophysical Research: Oceans*, 125:e2020JC016157, 2020. doi: 10.1029/2020JC016157.
- K. H. Roscoe, A. N. Schofield, and C. P. Wroth. On the yielding of soils. *Géotechnique*, 8(1):22–53, 1958. doi: 10.1680/geot.1958.8.1.22.
- H. Rusche. *Computational fluid dynamics of dispersed two-phase flows at high phase fractions*. PhD thesis, Imperial College London (University of London), 2002.
- K. Sassa, K. Dang, H. Yanagisawa, and B. He. A new landslide-induced tsunami simulation model and its application to the 1792 Unzen-Mayuyama landslide-and-tsunami disaster. *Landslides*, 13(6):1405–1419, 2016. doi: 10.1007/s10346-016-0691-9.
- S. B. Savage and K. Hutter. The motion of a finite mass of granular material down a rough incline. 199:177–215, 2 1989. doi: 10.1017/S0022112089000340.
- S. B. Savage and K. Hutter. The dynamics of avalanches of granular materials from initiation to runout. part i: Analysis. 86(1-4):201–223, 1991. doi: 10.1007/BF01175958.
- S. B. Savage, M. H. Babaei, and T. Dabros. Modeling gravitational collapse of rectangular granular piles in air and water. *Mechanics Research Communications*, 56:1–10, 2014. doi: 10.1016/j.mechrescom.2013.11.001.
- D. Schaeffer, T. Barker, D. Tsuji, P. Gremaud, M. Shearer, and J. Gray. Constitutive relations for compressible granular flow in the inertial regime. *Journal of Fluid Mechanics*, 874:926–951, 2019. doi: 10.1017/jfm.2019.476.
- D. G. Schaeffer. Instability in the evolution equations describing incompressible granular flow. *Journal of differential equations*, 66(1):19–50, 1987. doi: 10.1016/0022-0396(87)90038-6.
- L. Schambach, S. T. Grilli, J. T. Kirby, and F. Shi. Landslide tsunami hazard along the upper us east coast: effects of slide deformation, bottom friction, and frequency dispersion. *Pure and Applied Geophysics*, 176(7):3059–3098, 2019.
- L. Schambach, S. Grilli, D. Tappin, M. Gangemi, and G. Barbaro. New simulations and understanding of the 1908 messina tsunami for a dual seismic and deep submarine mass failure source. *Marine Geology*, 421:106093, 2020.
- L. Schambach, S. T. Grilli, and D. R. Tappin. New high-resolution modeling of the 2018 palu tsunami, based on supershear earthquake mechanisms and mapped coastal landslides, supports a dual source. *Frontiers in Earth Science*, 8, 2021.
- A. E. Scheidegger. On the prediction of the reach and velocity of catastrophic landslides. *Rock Mechanics and Rock Engineering*, 5(4):231–236, 1973. doi: 10.1007/BF01301796.
- A. Schofield and P. Wroth. *Critical state soil mechanics*. McGraw-Hill London, 1968.

- S. A. Sepúlveda and A. Serey. Tsunamigenic, earthquake-triggered rock slope failures during the April 21, 2007 Aisén earthquake, southern Chile (45.5 S). *Andean Geology*, 36(1):131–136, 2009.
- T. Shan and J. Zhao. A coupled CFD-DEM analysis of granular flow impacting on a water reservoir. *Acta Mechanica*, 225(8):2449–2470, 2014. doi: 10.1007/s00707-014-1119-z.
- C. Shi, Y. An, Q. Wu, Q. Liu, and Z. Cao. Numerical simulation of landslide-generated waves using a soil–water coupling smoothed particle hydrodynamics model. *Advances in water resources*, 92:130–141, 2016. doi: 10.1016/j.advwatres.2016.04.002.
- P. Si, H. Shi, and X. Yu. Development of a mathematical model for submarine granular flows. *Physics of Fluids*, 30(8):083302, 2018a. doi: 10.1063/1.5030349.
- P. Si, H. Shi, and X. Yu. A general numerical model for surface waves generated by granular material intruding into a water body. *Coastal Engineering*, 142:42–51, 2018b. doi: 10.1016/j.coastaleng.2018.09.001.
- D. R. Tappin, S. T. Grilli, J. C. Harris, R. J. Geller, T. Masterlark, J. T. Kirby, F. Shi, G. Ma, K. Thingbaijam, and P. M. Mai. Did a submarine landslide contribute to the 2011 tohoku tsunami? *Marine Geology*, 357:344–361, 2014.
- B. Tehranirad, J. C. Harris, A. R. Grilli, S. T. Grilli, S. Abadie, J. T. Kirby, and F. Shi. Far-field tsunami impact in the north atlantic basin from large scale flank collapses of the cumbre vieja volcano, la palma. *Pure and Applied Geophysics*, 172(12):3589–3616, 2015.
- K. Terzaghi. *Erdbaumechanik auf bodenphysikalischer Grundlage*. 1925.
- M. Trulsson, B. Andreotti, and P. Claudin. Transition from the viscous to inertial regime in dense suspensions. *Physical review letters*, 109(11):118305, 2012. doi: 10.1103/PhysRevLett.109.118305.
- S. Viroulet, A. Sauret, O. Kimmoun, and C. Kharif. Granular collapse into water: toward tsunami landslides. *Journal of Visualization*, 16(3):189–191, 2013. doi: 10.1007/s12650-013-0171-4.
- S. Viroulet, A. Sauret, O. Kimmoun, and C. Kharif. Tsunami waves generated by cliff collapse: comparison between experiments and triphasic simulations. In *Extreme Ocean Waves*, pages 173–190. Springer, 2016. doi: 10.1007/978-3-319-21575-4_10.
- T. R. Walter, M. H. Haghghi, F. M. Schneider, D. Coppola, M. Motagh, J. Saul, A. Babeyko, T. Dahm, V. R. Troll, F. Tilmann, et al. Complex hazard cascade culminating in the Anak Krakatau sector collapse. *Nature Communications*, 10(1):1–11, 2019. doi: 10.1038/s41467-019-12284-5.

- C. Wang, Y. Wang, C. Peng, and X. Meng. Dilatancy and compaction effects on the submerged granular column collapse. *Physics of Fluids*, 29(10):103307, 2017a. doi: 10.1063/1.4986502.
- C. Wang, Y. Wang, C. Peng, and X. Meng. Two-fluid smoothed particle hydrodynamics simulation of submerged granular column collapse. *Mechanics Research Communications*, 79:15–23, 2017b. doi: 10.1016/j.mechrescom.2016.12.001.
- J. Wang, S. N. Ward, and L. Xiao. Numerical simulation of the December 4, 2007 landslide-generated tsunami in Chehalis Lake, Canada. *Geophysical Journal International*, 201(1):372–376, 2015. doi: 10.1093/gji/ggv026.
- J. Wang, S. N. Ward, and L. Xiao. Tsunami squares modeling of landslide generated impulsive waves and its application to the 1792 unzen-mayuyama mega-slide in japan. *Engineering Geology*, 256:121–137, 2019.
- S. N. Ward and S. Day. Cumbre vieja volcano—potential collapse and tsunami at la palma, canary islands. *Geophysical Research Letters*, 28(17):3397–3400, 2001.
- R. Weiss, H. M. Fritz, and K. Wünnemann. Hybrid modeling of the mega-tsunami runup in Lituya Bay after half a century. *Geophysical Research Letters*, 36(9), 2009. doi: 10.1029/2009GL037814.
- H. G. Weller, G. Tabor, H. Jasak, and C. Fureby. A tensorial approach to computational continuum mechanics using object-oriented techniques. *Computers in Physics*, 12(6): 620–631, 1998. doi: 10.1063/1.168744.
- W.-J. Xu, Z.-G. Yao, Y.-T. Luo, and X.-Y. Dong. Study on landslide-induced wave disasters using a 3d coupled sph-dem method. *Bulletin of Engineering Geology and the Environment*, 79(1):467–483, 2020. doi: 10.1007/s10064-019-01558-3.
- S. Yavari-Ramshe and B. Ataie-Ashtiani. Numerical modeling of subaerial and submarine landslide-generated tsunami waves—recent advances and future challenges. *Landslides*, 13(6):1325–1368, 2016. doi: 10.1007/s10346-016-0734-2.
- M.-L. Yu and C.-H. Lee. Multi-phase-flow modeling of underwater landslides on an inclined plane and consequently generated waves. *Advances in Water Resources*, 133: 103421, 2019. doi: 10.1016/j.advwatres.2019.103421.
- T. Zengaffinen, F. Løvholt, G. K. Pedersen, and A. Muhari. Modelling 2018 anak krakatoa flank collapse and tsunami: Effect of landslide failure mechanism and dynamics on tsunami generation. *Pure and Applied Geophysics*, 177(6):2493–2516, 2020.
- C. Zhang, J. T. Kirby, F. Shi, G. Ma, and S. T. Grilli. A two-layer non-hydrostatic landslide model for tsunami generation on irregular bathymetry. 1. theoretical basis. *Ocean Modelling*, 159:101749, 2021a.

C. Zhang, J. T. Kirby, F. Shi, G. Ma, and S. T. Grilli. A two-layer non-hydrostatic landslide model for tsunami generation on irregular bathymetry. 2. Numerical discretization and model validation. *Ocean Modelling*, 160:101769, 2021b. doi: 10.1016/j.ocemod.2021.101769.

Answers to reviewer 3 on "Porous granular flow models can predict full-scale landslide tsunamis"

The original authors

October 10, 2021

The review is shown in blue, answers in black.

Summary: The Authors investigate subaerial landslide tsunamis with a (partially) newly implemented model based on the tri-phase Navier-Stokes equations in the OpenFOAM environment. They model successfully the small-scale granular cases of Viroulet et al. (2013) in 2D, followed by a full 3D-simulation of the 2014 Lake Askja case with good agreement. They further compare their run-up results with past results based on simplified slide and hydrodynamic models.

Evaluation: I would like to congratulate the Authors on this excellent study. The topic and study match well into Nature Communications with similar contributions published in the past. The Authors combine state-of-the-art knowhow in granular dynamics and hydrodynamics and implement a new model in the open-source environment OpenFOAM. This model includes a low level of "modelling" parameters, models the full process (slide kinematics, hydrodynamics) and a very good agreement with the laboratory data and a good agreement with the field data is achieved as well, without parameter optimisations and one set of parameters for both the laboratory and nature event. This is indeed an excellent and useful result; landslide-tsunami models often achieve good agreement, but only after significant parameter fitting and optimisations based for example on laboratory experiments such that they may not be applied to more complex real or other laboratory cases with confidence. In past studies also often the slide model is prescribed or not fully realistic (e.g. modelled as a rigid block).

Thank you very much for your kind words and the rigorous review. We conducted many more simulations that you requested. We agree that these simulations and studies are useful. You will see that we considerably extended the supplementary materials with mesh refinement (or better said, coarsening) studies, sensitivity analysis, quantitative descriptions of modelling errors and further investigations of 3D effects and the influence of turbulence.

Given the substantial required computational resources (e.g. 450 h and 220 cores required to run the 2014 Lake Askja case, L438), no convergence tests have been presented (a basic requirement for numerical simulations on peer-review level) and some compromises were necessary for the wave propagation (too low resolution to avoid numerical diffusion (L243)). These limitations are clearly communicated in the article. As recognised by the Authors, simulation time could be saved by coupling the model with e.g. a non-linear non-hydrostatic shallow-water equation model (considering frequency dispersion) which is fully appropriate to model wave propagation and inundation, as already demonstrated for the far field of the 2014 Lake Askja case, once only the water phase is left (e.g. SWASH, Zijlema et al. 2011, there are probable similar implementations in OpenFOAM available as well, which would still enable the use of one model environment only for the full process). In other words, the computational expensive full Navier-Stokes model is not required in the far field and at the same time the described limitations of the numerical diffusion due to a too small resolution would be overcome by using e.g. SWASH. It is also made clear in the manuscript that the laboratory experiments have been simulated in 2D. In my view, it needs more justification on whether boundary layer effects in the granular material would not affect the slide kinematics (a 3D simulation of at least one of the laboratory cases would be desirable to shed light on this point). The underlying laboratory experiments are small and will be affected by surface tension scale effects, which cannot be modelled (L132, L314) and in addition grain Reynolds number effects (e.g. affecting slide run-out) appear not to be captured as well.

The required computational resources are high but standard or even sub standard in many industries (automotive, aviation) and can be handled with know-how from these disciplines. The price for the computational resources for the complete study sums up to approximately 30 000 EUR (300 000 core hours á 0.1 EUR/core hour on e.g. Amazon EC cloud) which is acceptable. The presented model is on the edge of what we can currently afford and as such well suited for an innovative journal as Nature Communications. Notably, performance optimisations and increasing computational power will make full three-dimensional simulations much cheaper in the next decade and we are certain that three-dimensional simulation will play an important role in tsunami predictions.

Extensive convergence tests have been conducted in multiple studies leading up to this work (Rauter et al., 2021; Rauter, 2021; Barker et al., 2021) and we were able to derive criteria for grid and time step resolutions. Nevertheless we now added convergence tests for all cases in the supplementary materials. We changed the manuscript at the respective passages to refer to the supplementary materials.

We avoided coupling to a depth-integrated model in the far-field for the following reasons: (1) This coupling is complex and we do not want to introduce two new complex mechanisms in one work. There are examples and code for such couplings in OpenFOAM (Mintgen and Manhart, 2018; Di Paolo et al., 2021a,b) and outside OpenFOAM (Løvholt et al., 2008; Abadie et al., 2012) but it is not clear how to adopt this approach to tri-phase simulations and the complex lake geometry with inundations along all shores. (2) The simulation of the whole process (slide, wave generation, propagation, inundation) with one model convinces through clarity and simplicity. (3) It is scientifically interesting to

see the full lake simulated in 3D. (4) Lake Askja is relatively small and at least half of the lake should have been simulated in full 3D, as wave breaking and other 3D effects are visible in this area. The saving would probably be less than 50% and although relevant, it would not be a game-changer.

We thought that the successful 2D simulation shows that side wall effects are small in the laboratory case. Nevertheless we conducted a width-resolved simulation of one lab experiment and added it to supplementary material, alongside a short discussion of the issue.

The grain Reynolds number effect is a very interesting topic. Kessler et al. (2020) write: "These scale effects are strongly correlated with Re , suggesting that interactions between grains and air are primarily responsible for the observed scale effects." Our model considers part of this effect/interaction in terms of the Stokes number (Rauter, 2021) but more investigations are required to fully investigate this topic. We discuss scale effects like this in the new manuscript.

In general, I would wish to see more quantitative measures to judge a goodness of fit rather than qualitative statements such as "a very good agreement" has been achieved. There is actually a method (combining Discrete Element Modelling DEM with Computational Fluid Dynamics CFD) implemented in an open-source environment (e.g. CF-DEM), which appears to involve even less modelling and the new method should be put into the context of this method. I also understand that the Lake Askja case simulation relies on past event field data, such as the deposit (to define an appropriate friction parameter) or the initial slide shape based on the topography obtained after the event. In other words, more work will be required to also reliably predict future events (which should be the long-term goal), in addition of the modelling of a past event. However, the achievements of the Authors are still a step forward.

We added many quantitative measures of the model error in the supplementary material. We felt that it did not fit into the manuscript as we considered it too technical. We refer to it at the respective positions in the manuscript and pull out a number from time to time.

We think that discrete and continuum mechanical models have both their up- and downsides but we see continuum mechanical models excelling when it comes to real scale landslides. The discrete element method (DEM) scales poorly with the length scale because the number of particles (discrete elements) grows with the volume of the slide and the same is the case for the number of collisions that need to be calculated. Considering an average particle size of 0.1 m, the Lake Askja slide would consist of 10^{10} particles.

There is also an extreme diversity of DEM methods, ranging from "most effects are directly simulated" to "most effects are considered with empirical models" (e.g. rolling resistance as a replacement for non-spherical particles; unrealistic low (empirical) particle stiffness to accelerate simulation times; non-resolved particles in CFD against resolved particles in CFD). Depending on the exact choice of models, DEM can include the same

or even more modelling than the presented method.

As such, we (personally) see DEM as a very promising tool to get new insights into the physical processes but of little relevance to real cases, which is the focus of this work. Nevertheless, we added a respective mentioning of DEM as an alternative to the discussion part of the manuscript.

These weaknesses together with further less major points such as some unclear statements, typos and inconsistencies are highlighted under the following.

Detailed points on the suggested improvements: L13: Please replace the sentence “a gap in providing unbiased predictions of real events.” with a more specific statement. Again on L48, the word biased can have many meanings, which one do the Authors mean?

Thank you very much for this suggestion. We changed the sentence to "The lack of models incorporating both the landslide and the wave mechanics represents a gap in providing consistent predictions of real events."

L15: “For the first time, we show a model that scales consistently from small scale laboratory experiments to full scale catastrophic events.” I find this statement opens up some questions. It is well-known that the physics of the described phenomena change with the scale due to scale effects (e.g. Kessler et al. 2020). E.g. Grain Reynolds number effect become more dominant at small scale laboratory experiments and e.g. affect the slide run-out, and as such tsunami generation. Also the water domain is affected by significant scale effects, given that the underlying experiments of Viroulet et al. (2013) have been conducted a small water depth 0.15 m and the model does exclude surface tension (L132). Parts of this is later communicated in the article, but my immediate though at this location in the manuscript would be that a consistent scaling is not necessarily something positive, as the physics changes at small scales due to scale effects.

Thank you very much for this comment. We use "scaling" in various contexts and we see how this can be confusing.

We think that many tsunami models, especially Navier-Stokes type models, are based on the wrong physics (e.g. viscous or inviscid flow). Therefore, we cannot close the gap between successful small case experiment simulations and real case simulations without considerably and unsystematically changing "material parameters" by multiple orders of magnitude. This is what we meant with scalability but we understand your confusion and changed the respective sentence to: "For the first time, we show a model that consistently predicts small scale laboratory experiments as well as full scale catastrophic events."

L16: The statement that the model accurately captures the inundation is not fully reflected in Fig. 8b where locally deviations of up to 50% are shown, e.g. the measurements show 40 m and the simulation nearly 60 m. Please communicate this more carefully.

We described the deviations in detail now.

L31: What are “impulsive landslide tsunamis”? The term is not common and nearly every tsunami is generated by an impulse.

You are correct, we mixed together impulse wave and tsunami. We removed the respective term.

L36/L52/L281: Would a Discrete Element Modelling DEM coupled with Computational Fluid Dynamics not involve less modelling than in the proposed model? There are open-source codes available to do so, e.g. CFDEM coupling (CFDEM coupling Documentation — CFDEMcoupling v3.X documentation) and there are already publications into subaerial-landslide tsunamis available based on CDEM-CFD coupling (e.g. Shan et al. 2014). The work should be put into the context of such methods.

We added the suggested reference to the manuscript. We answer on questions regarding DEM above.

L42: “and allows mixing of the granular material with fluids that represent the tsunami.” I assume fluids stands for air and water, but the former is not representing the tsunami.

Yes, fluids stand for air and water that are combined in the continuous phase ϕ_c . The tsunami model, as previously used in Rauter et al. (2021) is operating within this phase such that air is playing a role too.

L117: Which recent modelling studies could not match part of the waves or are based on different parameter sets? These studies should really be mentioned for the Reader to be able to verify this statement.

We thought about the previous studies of that model the experiment of Viroulet et al. (2013), e.g. Viroulet et al. (2016); Clous and Abadie (2019) and our own approach to do this with a simpler model (Hoße et al., 2019). However, we see that Si et al. (2018) was actually quite successful in simulating both the slide and the wave so we removed this sentence. Notably, Si et al. (2018) use a similar approach as we do but do not attempt 3D cases, complex terrain or real scale cases.

L136: “The simulated wave (Fig. 4) matches the measurements from the experiment very well.” This should be backed up with a quantitative analysis, e.g. a normalised Root Mean Square Error with respect of the maximum wave amplitude or the full wave profile, or similar.

Thank you very much for this comment. We added a very detailed quantitative analysis to the supplementary materials and pull numbers into the manuscript where we see a fit.

L152: “...we can avoid complex boundary conditions and eventual artificial wave reflections.” Please clarify what artificial wave reflections is.

A boundary in the CFD model will reflect the wave in some form and magnitude. The simplest boundary condition ($\mathbf{u}_s = \mathbf{u}_c = \mathbf{0}$, $\mathbf{n} \cdot \nabla p = 0$) will reflect a wave as if there was a wall. If the boundary represents a real physical object, this reflection is accurate. If the boundary is just there to limit the simulation space, it will generate reflections that do not exist in nature. There are complex boundary conditions that should avoid this but they do not work perfectly or suppress reflections that are actually realistic. Further they can be problematic and unstable. By simulating the whole lake we can avoid these artificial reflections and problems.

Fig. 5: Please make the time indication better visible.

Fixed.

Caption Fig. 8: “The lake outline at rest” remains unclear, should it read “The lake is outline at rest”? Also in (b) I would write “Measurements” in plural form.

Fixed.

Fig. 8 and L218: Also here, a quantitative, not only qualitative (“modelled well”), comparison is necessary. Further, please make clear that the Boussinesq and Shallow-water results are taken from Gylfadóttir et al. (2017) and that it is based on a different slide model.

We added a qualitative analysis to the supplementary materials and refer to it in the manuscript. We note that the results are taken from Gylfadóttir et al. (2017) and that they use a different slide model.

L200: “Scenarios with similar runout but e.g. different impact velocities will differ strongly in their tsunamigenic potential, which highlights once more the value of con-

sistent models that provide the whole history.” There are studies which looked into the effect of the underwater slide kinematics on the wave in depth already, this was often modelled/considered with the “characteristic time of submerged landslide motion” (see Walder et al. 2013; Heller and Spinneken, 2013).

Thank you for these two suggested references. However, we are not sure that these references fit here and that we speak about the same effect. In the simulations we get a relatively violent impact that decelerates the majority of the slide. From here on the wave forms and the height is proportional to the impact velocity (we tried it with a second simulation of Lake Askja). Further, the front is slowly mixed with water and this mixture keeps flowing until the bump in the lake. The runout is widely independent of the initial velocity as the momentum is widely lost during impact. Therefore we get a large variation in the tsunamigenic potential while the runout distance is basically the same. This was observed before and we refer to the respective studies (Moretti et al., 2012; Løvholt et al., 2020). Notably, not all of the momentum lost is transferred to the water body, but the high pressure peak at impact generates strong friction in the granular material that further slows down the slide. This leads to a very complex system and a wide range of possible situations and outcomes.

L201: "A definitive conclusion" on what, there are several aspects mentioned?

Thank you very much. We meant the last statement ("These results render it unlikely that the deposit is a reliable indicator for the tsunamigenic potential."). We were interested by this question and ran another simulation with reduced friction coefficients $\mu_s = 0.14$, $\mu_d = 0.34$. The results show that the runout is similar as the slide is stopped by the elevation in the lake. The inundation on the other hand differs drastically with derivations of up to 100%. We added this simulation to the supplementary materials and refer to it in the manuscript. We think that this shows that the deposit is unlikely a reliable indicator for the tsunamigenic potential. We changed the respective part in the manuscript.

L222: Where would this “artificial damping” originate from? I can see that this is discussed on L242, but the Reader will have this question here.

We fixed some problems in the simulations (An error in the water viscosity, which was by a factor of 15 too big) and reviewed the numerical diffusion effects, that are now smaller due to this correction. We came to the conclusion that this is much less problematic than we initially thought. We changed respectively large parts of the manuscript including this line.

L228: “very well” needs a qualitative measure of the goodness of fit.

We added quantitative measures to all simulation results in the supplementary mate-

rials.

L235: The statement "... indicating that the inundation allows remarkable conclusions on the generation event." remains unclear, please try to be more specific.

Changed to: "This shows that traces from the wave, such as the inundation, allow remarkable conclusions on the generation event, i.e. the landslide. In fact, these indirect traces seem to be more reliable than direct traces such as the deposition."

L271: A reference to the section where the computational resources are addressed would help the Reader.

Added.

L315: It is not only surface tension which affects small-scale models (and results in scale effects), but also the grain Reynold number (see e.g. Kessler et al. 2020). As far as I understand, these effects are also not modelled appropriately by modelling the granular phase with a rheology (in contrast to a DEM coupled with CFD).

The grain Reynold number, comparable to the Stokes number ($Re = |u|d/\nu_c$ and $St = 2d^2|\mathbf{S}|\rho_g/(\nu_c\rho_c)$ and with $|\mathbf{S}| = 0.5|\mathbf{u}|/d$ we get $St = |\mathbf{u}|d/\nu_c\rho_g/\rho_c = Re\rho_g/\rho_c$), is included indirectly in many terms. This is either by choosing $\mu(I), \phi(I)$ - (high Stokes numbers), $\mu(J), \phi(J)$ - (low Stokes numbers) or $\mu(K), \phi(K)$ -scaling (low, intermitten and high Stokes numbers) (see, e.g. Rauter, 2021) and the drag terms are scaling with the grain Reynolds number. Therefore we think that we consider some of the scaling with grain Reynolds number but this requires more investigations.

There are further effects that lead to lower friction at larger scales (slide volume $> 500\,000\text{m}^3$) and this is generally known in the community (Scheidegger, 1973; Lucas et al., 2014). We do not know how this is linked to the grain Reynolds number. We tried to present this in more detail in the new version of the manuscript.

L316: High Reynolds number of what, of the slide or water?

We thought of Reynolds numbers in a general way, taking length and velocity scale from the slide and viscosity scale from the water.

L329: Please specify whether these are vectors or parameters, e.g. for g.

Added.

Eqs. (6) and (7), are there not some parameters required a priori, such as the granular viscosity and the friction coefficients? Also, the grain diameter $d = 0.01$ m in the real case (L398) is a choice which may affect the results?

The granular viscosity is not a parameter but calculated following Eq. (6). The friction coefficients μ_s and μ_d are parameters as stated in Tab. 1 and the text. These parameters are usually known relatively well and also some corrections that need to be applied following the overall size of the event Scheidegger (1973); Lucas et al. (2014).

The grain diameter is indeed a choice, which may affect results, although the sensitivity on this parameter is not very strong (see supplementary material). The grain diameter appears in two equations, in the $\mu(I)$ -rheology and in the equation for the permeability. We aimed to set this parameter in such a way that we get the permeability correct.

The diameter d has only a minor role in the $\mu(I)$ -rheology that can be further adjusted with the choice of I_0 . However, literature (Barker et al., 2021, and references therein) indicate that the average diameter can be chosen in polydisperse flows with good accuracy.

Terzaghi (see Fannin, 2008) suggest that the smallest 15% of the particles are responsible for the permeability. Therefore the 15%-percentile of the grain diameters, d_{15} , is usually used in estimating the permeability. For the sake of simplicity we used only one diameter in our model (both in the $\mu(I)$ -rheology and the permeability equation) and that is d_{15} . Generally, we know from the sensitivity analysis that the grain diameter has a limited effect, even when changed by an order of magnitude (see supplementary materials).

L401: It is stated “The quasi-static friction coefficient ... was chosen to match the overall slope gradient from release to deposition”. This also appears to be a parameter which is not known a priori of an event? This seems to contradict the statement on L412 “Note that the chosen parameters were not fitted by comparing the computational results with the observations.” and to some degree the overall conclusion that there is no need for “optimized parameters” on L267.

We used multiple sources to estimate the friction angle of the slide. We were careful with this value because we estimated that it would have a strong influence on the results and we did not have computational resources multiple simulation (we did it during the review and indeed, the friction angle has strong influence). We know from literature that the friction angle reduces for slides larger than $500\,000\text{ m}^3$ (e.g. Scheidegger, 1973; Lucas et al., 2014) and a static test of the material at a low pressure level would not have been useful to estimate this value.

The overall slope gradient was the simplest way to derive this quantity and in order to keep the paper streamlined we only mentioned this approach. Notably, we could only get a rough estimation from the deposition and there is sever uncertainty, depending on how we measure the runout. If we use the tip of the runout, the overall slope angle would be about 7° . If we use the peak of the subaquatic deposition it would be 10° and if we use the peak of the subaerial part of the deposition it would be 15° (measured from

centre of the initial mass to centre of the final mass).

Following the empirical relation of Scheidegger (1973); Lucas et al. (2014) and using the slide volume of $2 \cdot 10^7 \text{ m}^3$, we would expect an overall slope angle of 16° . Other sources, such as the Swiss and Austrian avalanche guidelines suggest values as low as $9^\circ - 11^\circ$ for avalanches bigger than 10^5 m^3 and even lower ones for debris flows.

This way we could limit the friction coefficient between 7° and 16° , using data from literature and the measured overall slope gradient. The decision landed on 10° and we selected this value before doing a single simulation. We do not "compare the computational results with the observation" as mentioned in the manuscript. We would choose the same value again and also for similar events and we also do not see this parameter as an optimized value.

Anyway, we reduced the strength of these statements so that they are now more in line with our processes.

L416: "terrain before the event in the deposition area and the terrain after the event" also here information not available prior to the event are required.

We assume here a scenario where the failing layer is known, yes. Weak layers are often known before the actual collapse of a slope because of e.g. large deformations (see, e.g. a suspicious area in Karrat Fjord, Paris et al., 2019) and these areas are often monitored (e.g. Aknes rock slide in Norway, Oppikofer et al., 2009). Also the Vajont disaster was foreshadowed by large formations in the slope that would later collapse. Weak layers and failure zones can also be identified by geological explorations or estimated with the continuum (solid) mechanics simulations. For this work, we had to limit the scope to modelling the landslide dynamics. The slope stability and fracture process would make this an unprecedentedly large task and would require another study.

L519: The access data 20.01.2020 is rather a long time ago. Please check and update this date if possible.

We will update the url and access date in the final publication.

Typos and minor issues: L17: Please write "Lake Askja" rather than "lake Askja" (in general, the writing style is not consistent in the manuscript (lake versus Lake)). Caption Figure 1: Write "from" rather than "form". Fig. 2: Write "saturated" rather than "sat.". There is also an unnecessary abbreviation used in Table 1. Fig. 4: Add free spaces between the parameter and the units on all axes. Caption Fig. 6: Write "in Figure 5" rather than "in the figure before". Caption Fig. 7: Add "Lake" in front of "Askja" and write "black" rather than "back". L451: There are issues. L467: Write "Italian" rather than "italian". L102: Remove the full stop. L106: Write "signals" rather than "signal". L113: Add "the" ahead of "landslide". L202: Add a comma after

“however”. L308: Add “a” in front of “part” and “the” in front of “home”. L462: Add a comma after the first author. L486: Write OpenFOAM rather than openfoam. L487: Write the name Lovholt correctly; L510: The capital letters in the article title are not consistent with other titles. L565: Please use capital letters for the Journal name.

Fixed.

General comment: The Authors use American (e.g. L96 “visualized”) rather than British English (visualised). Nature Communications may prefer the latter.

We try to use British English and checked the paper.

Movie supplement: Please write “Lake Askja” rather than “Askja” only (5 times).

Fixed.

Supplementary material: Caption Fig. 2: Please drop one of the “cases”. Also replace rho with the symbol.

Fixed.

Captions Figs. 3 and 7: Please write “Lake Askja” rather than “lake Askja” or “Askja”.

Fixed.

Caption Figs. 8 to 11: Please add free spaces between the parameters and the units on all axes. The text in the figures is too larger compared to the main text (e.g. decrease the figure size).

Fixed.

2nd paragraph under section 3: Please write listing rather than lst. (several times). There is not reason to abbreviate this word.

Fixed.

References: Heller, V., Spinneken, J. (2013). Improved landslide-tsunami predictions: effects of block model parameters and slide model. *Journal of Geophysical Research-Ocean* 118(3):1489-1507 Kessler, M., Heller, V., Turnbull, B. (2020). Grain Reynolds

number scale effects in dry granular slides. *Journal of Geophysical Research-Earth Surface* 125(1):1-19. Shan, T., Zhao, J. (2014) A coupled CFD-DEM analysis of granular flow impacting on a water reservoir. *Acta Mechanica* 225:2449-2470. Walder, J.S., Watts, P., Sorensen, O.E., Janssen, K., (2003). Tsunamis generated by subaerial mass flows. *Journal of Geophysical Research* 108(B5):2236(2). Zijlema, M., Stelling, G., Smit, P. (2011). SWASH: an operational public domain code for simulating wave fields and rapidly varied flows in coastal waters. *Coastal Engineering* 58 (10):992–1012.

References

- S. M. Abadie, J. C. Harris, S. T. Grilli, and R. Fabre. Numerical modeling of tsunami waves generated by the flank collapse of the Cumbre Vieja volcano (La Palma, Canary Islands): Tsunami source and near field effects. *Journal of Geophysical Research*, 117 (C05030), 2012. doi: 10.1029/2011JC007646.
- T. Barker, M. Rauter, E. Maguire, C. Johnson, and J. Gray. Coupling rheology and segregation in granular flows. *Journal of Fluid Mechanics*, 909, 2021. doi: 10.1017/jfm.2020.973.
- L. Clous and S. Abadie. Simulation of energy transfers in waves generated by granular slides. *Landslides*, 16(9):1663–1679, 2019. doi: 10.1007/s10346-019-01180-0.
- B. Di Paolo, J. L. Lara, G. Barajas, and Í. J. Losada. Wave and structure interaction using multi-domain couplings for navier-stokes solvers in openfoam®. part i: Implementation and validation. *Coastal Engineering*, 164:103799, 2021a.
- B. Di Paolo, J. L. Lara, G. Barajas, and Í. J. Losada. Waves and structure interaction using multi-domain couplings for navier-stokes solvers in openfoam®. part ii: Validation and application to complex cases. *Coastal Engineering*, 164:103818, 2021b.
- J. Fannin. Karl terzaghi: from theory to practice in geotechnical filter design. *Journal of geotechnical and geoenvironmental engineering*, 134(3):267–276, 2008.
- S. S. Gylfadóttir, J. Kim, J. K. Helgason, S. Brynjólfsson, Á. Höskuldsson, T. Jóhannesson, C. B. Harbitz, and F. Løvholt. The 2014 Lake Askja rockslide-induced tsunami: Optimization of numerical tsunami model using observed data. *Journal of Geophysical Research: Oceans*, 122(5):4110–4122, 2017. doi: 10.1002/2016JC012496.
- L. Hoße, M. Rauter, and F. Løvholt. Numerical simulation of impulse waves using primitive flow models. In *Geophysical Research Abstracts*, volume 21, 2019. URL https://www.researchgate.net/publication/332423112_Numerical_Simulation_of_Impulse_Waves_using_Primitive_Flow_Models_Preliminary_Results.
- M. Kessler, V. Heller, and B. Turnbull. Grain reynolds number scale effects in dry granular slides. *Journal of Geophysical Research: Earth Surface*, 125(1):e2019JF005347, 2020.

- F. Løvholt, G. Pedersen, and G. Gisler. Oceanic propagation of a potential tsunami from the La Palma Island. *Journal of Geophysical Research: Oceans*, 113(C9), 2008. doi: 10.1029/2007JC004603.
- F. Løvholt, S. Glimsdal, and C. B. Harbitz. On the landslide tsunami uncertainty and hazard. *Landslides*, 17:2301–2315, 2020. doi: 10.1007/s10346-020-01429-z.
- A. Lucas, A. Mangeney, and J. P. Ampuero. Frictional velocity-weakening in landslides on earth and on other planetary bodies. *Nature communications*, 5(1):1–9, 2014. doi: 10.1038/ncomms4417.
- F. Mintgen and M. Manhart. A bi-directional coupling of 2D shallow water and 3D Reynolds-averaged Navier–Stokes models. *Journal of Hydraulic Research*, 56:771–785, 2018. doi: 10.1080/00221686.2017.1419989.
- L. Moretti, A. Mangeney, Y. Capdeville, E. Stutzmann, C. Huggel, D. Schneider, and F. Bouchut. Numerical modeling of the mount steller landslide flow history and of the generated long period seismic waves. *Geophysical Research Letters*, 39(16), 2012. doi: 10.1029/2012GL052511.
- T. Oppikofer, M. Jaboyedoff, L. Blikra, M.-H. Derron, and R. Metzger. Characterization and monitoring of the Aknes rockslide using terrestrial laser scanning. *Natural Hazards and Earth System Sciences*, 9(3):1003–1019, 2009.
- A. Paris, E. A. Okal, C. Guérin, P. Heinrich, F. Schindelé, and H. Hébert. Numerical modeling of the June 17, 2017 landslide and tsunami events in Karrat Fjord, West Greenland. *Pure and Applied Geophysics*, 176(7):3035–3057, 2019. doi: 10.1007/s00024-019-02123-5.
- M. Rauter. The compressible granular collapse in a fluid as a continuum: validity of a Navier-Stokes model with $\mu(J), \phi(J)$ -rheology. *Journal of Fluid Mechanics*, 915, 2021. doi: 10.1017/jfm.2021.107.
- M. Rauter, L. Hoße, R. Mulligan, A. Take, and F. Løvholt. Numerical simulation of impulse wave generation by idealized landslides with OpenFOAM. *Coastal Engineering*, 165:103815, 2021. doi: 10.1016/j.coastaleng.2020.103815.
- A. E. Scheidegger. On the prediction of the reach and velocity of catastrophic landslides. *Rock Mechanics and Rock Engineering*, 5(4):231–236, 1973. doi: 10.1007/BF01301796.
- P. Si, H. Shi, and X. Yu. A general numerical model for surface waves generated by granular material intruding into a water body. *Coastal Engineering*, 142:42–51, 2018. doi: 10.1016/j.coastaleng.2018.09.001.
- S. Viroulet, A. Sauret, O. Kimmoun, and C. Kharif. Granular collapse into water: toward tsunami landslides. *Journal of Visualization*, 16(3):189–191, 2013. doi: 10.1007/s12650-013-0171-4.

S. Viroulet, A. Sauret, O. Kimmoun, and C. Kharif. Tsunami waves generated by cliff collapse: comparison between experiments and triphasic simulations. In *Extreme Ocean Waves*, pages 173–190. Springer, 2016. doi: 10.1007/978-3-319-21575-4_10.

Reviewer #3:

Remarks to the Author:

Summary:

This is the 2nd time that I review this article. I would like to congratulate the Authors once more for this excellent study representing a step forward in landslide-tsunami research. The Authors invested a significant amount of time, effort and work to revise the manuscript and submitted a significantly improved version based on many additional simulations (e.g. mesh refinement study, 3D effects, sensitivity analysis), a quantitative analysis, a refined reference list, a clearer and more balanced communication of the findings, etc. This addresses nearly all of my main concerns of the previous version (convergence tests, inclusion of DEM, etc.). However, I wonder if there is a better way to quantify the goodness of fit than with [mm] or [m], e.g. through a % relative to the maximum wave amplitude, or similar. Below are a number of further points, but they are all minor. I recommend another revision where the points below are addressed/considered.

Detailed points:

L15: It may need a hyphen for the words "small-scale" and "full-scale" (compared with the writing style on L144).

L18/L47: ", reminiscent of a large scale laboratory," can be removed.

L22: "plunging" is somehow associated with a vertical fall and could be replaced with a more precise word, e.g. "impacting".

L27: Please write "from 1904-1936".

L38: The word "impossible" is very strong and should be revised, as there are surely other ways to advance the field.

L23/L49: "Krakatoa" versus "Krakatau", are these the same areas? Please use a consistent writing style throughout the manuscript.

L41: The rheology parameters should be defined for the understanding of the general reader.

Fig. 2: "sat." should not be abbreviated, there is enough space for the full term. In the caption, it should read "stress models" or it needs an article in singular form.

Table 1: I feel that the description should be added next to the parameter, and the abbreviation (par.) should be avoided.

L94: "water surface wave" should be replaced by "tsunami".

Fig. 3: The font sizes of the text within this figure is too large compared to the text on the axes.

Fig. 4: The parameters x and z should be written in italic. On the y -axes, there should be free spaces between the unit and the numbers, and the subscripts should be written lower.

L105: Please replace "like" with "such as" (a more formal term).

L132/L219/L231: I like that a quantification of the error between measurements and simulations has been introduced. The error in [m] or [mm] is useful for a comparison with other configurations within this study. It would be good to express the error also e.g. as a % of the maximum wave amplitude, rather than in mm, as this would be a more general quantification.

Fig. 6: In the caption, please replace "the figure before" with "Figure 5".

L205: The statement that the deposition is unlikely to be used as a reliable indicator of the tsunamigenic potential opens up some questions and should be weakened in my view, e.g. "deposition alone...". Some Readers may criticize that, of course, also the bathymetry needs to be

considered in relation to the slide run-out distance.

L210: Please write "the slide" rather than "slide" only.

L218: There is an inconsistency in the manuscript about the "Oxford comma", here no comma is used, elsewhere a comma is used after "and".

L227: Please replace "big" with "significant" to use a more scientific term.

L321: I feel that the statement of "our model performs well" does not match the maximum error of 32.24 m, i.e. it over-sells the results.

L310/1: Please replace "at" with "from" (twice).

L318: "We thank.... the support of the editor" is Grammatically incorrect, it is not possible to thank the support.

L332ff: I believe x should be written in bold to indicate that this is a vector, also in the supplementary material.

Eq. (4)ff: Please double-check that all parameters used in these equations are defined in the text.

L356: "low" may be replaced by "small".

L399: Please replace "sand" with "granular".

L413: Please write "Herein" rather than "In here".

Refs. [4], [5], [13], [39], [44], [53] and [58]: Please write the names with capital letters, also the journal names (e.g. "Reports").

Ref. [64]: Please do not use abbreviations.

Movie Supplement:

On the 2nd last line it should read "black line" rather than "back line".

Supplementary material:

Fig. 3: "Outline" should be written with a lower case letter. In the caption write "and its surrounding terrain" rather than "the terrain surrounding it".

Caption Fig. 8: Please write "Lake Askja" rather than only "Askja" and drop one of the "in the" on the 2nd last line.

Ahead of Eq. (38): Please add a comma after "First".

Table 1: Please replace "max." with "maximum".

Table 2: In the caption, please write $^{\circ}$ as a superscript.

Table 3: In the caption please write "shallow" rather than "Shallow", and there is a typo in "Boussinsq", both in the caption and in the table.

P12, L3 from the bottom: There is a free space missing ahead of [5].

Before Table 5: The statement "although the quantitative results should be taken with a grain of

sand." remains unclear, please rephrase.

Fig. 11: "Measurement" is written with an upper case letter, in contrast to other figures in the manuscript where lower case letters are used. See also Fig. 37.

In the captions of Tabs. 6 to 13 the unit (m) for the slope angle is incorrect.

Section 5.8: On line 5 please drop the 2nd "on".

After Eq. (47): Please revise the sentence "The macroscopic wave is stronger damped and after the first wave much smoother at the first gauge than the simulation without the turbulence model and the experiment." And 4 lines further down please drop "on".

Fig. 31: In the legend the text should not be written in italic. Please also use lower case letters for "Equations". I do not think the abbreviations "NS" and "RANS" have been introduced, so please either introduce them or use the full terms.

Fig. 32: The same as in the previous comment applies to the headings of this figure.

Fig. 36: In this caption, please write the text not in italic and add free spaces.

Sec. 7: The first sentence can be linked to the 2nd paragraph to avoid a paragraph consisting of one sentence only.

Ref. [3]: Please do not use abbreviations.

Answers to reviewer 2 on "Granular porous landslide tsunami modelling – the 2014 Lake Askja flank collapse"

The original authors

December 8, 2021

The review is shown in blue, answers in black.

I commend the authors for significantly revising and improving their manuscript, in both form and content, and for providing detailed responses to all my comments or concerns (as well as those of other reviewers). I also appreciate that they did some significant additional work in support of their modeling (such as reported in supplementary material). The correction of the error on the viscosity value, in particular, makes some results much more interesting and conclusive. Likewise with the comparison with Geoclaw.

Although we may still have some differences in views and interpretation of some of the existing modeling studies and what they represent, I find that my most important comments or concerns have been adequately addressed in the revised manuscript. It is not the place here to have a debate regarding these interpretations provided the paper fairly and sufficiently completely presents the state of the art of previous modeling efforts, including their strength and weaknesses in contrast to the present work, for the readers to make up their own opinion and judgment. I find the revised manuscript to be better balanced in this respect.

I indicated in my first review, that I thought this work had merit and warranted publication after revision. While I still think that, this being a new type of modeling, a wider scope and more detailed paper, with more benchmarks particularly including inundation and runup, and 3D subaerial slide benchmarks, could have been published first, I am sure the authors will want to do that in future work and for this reason I recommend publication in Nature Communication of the revised manuscript.

We would like to thank the referee again for his detailed review and for spotting some critical issues. His criticism and attention for details was essential to bring the paper to its final form. We appreciated the fulminant discussion and hope to continue discussing this topic in the future.

Answers to reviewer 3 on "Granular porous landslide tsunami modelling – the 2014 Lake Askja flank collapse"

The original authors

December 8, 2021

The review is shown in blue, answers in black.

This is the 2nd time that I review this article. I would like to congratulate the Authors once more for this excellent study representing a step forward in landslide-tsunami research. The Authors invested a significant amount of time, effort and work to revise the manuscript and submitted a significantly improved version based on many additional simulations (e.g. mesh refinement study, 3D effects, sensitivity analysis), a quantitative analysis, a refined reference list, a clearer and more balanced communication of the findings, etc. This addresses nearly all of my main concerns of the previous version (convergence tests, inclusion of DEM, etc.). However, I wonder if there is a better way to quantify the goodness of fit than with [mm] or [m], e.g. through a % relative to the maximum wave amplitude, or similar. Below are a number of further points, but they are all minor. I recommend another revision where the points below are addressed/considered.

Thank you again for your detailed review. Your constructive comments and suggestions improved the paper considerably for which we are very grateful.

Detailed points:

L15: It may need a hyphen for the words "small-scale" and "full-scale" (compared with the writing style on L144). Fixed.

L18/L47: ", reminiscent of a large scale laboratory," can be removed. Fixed.

L22: "plunging" is somehow associated with a vertical fall and could be replaced with a more precise word, e.g. "impacting". Fixed.

L27: Please write "from 1904-1936". Fixed.

L38: The word "impossible" is very strong and should be revised, as there are surely other ways to advance the field. Fixed.

L23/L49: “Krakatoa” versus “Krakatau”, are these the same areas? Please use a consistent writing style throughout the manuscript. Fixed.

L41: The rheology parameters should be defined for the understanding of the general reader.

There are no rheology parameters near this line. The $\mu(I)$ -rheology is mentioned here, however this is solely the name of a model (we have to agree that this is a confusing name). The rheological parameters are introduced in section 2.1 and table 1, where they are described shortly. A longer description can be found in the method section 3.1.

Fig. 2: “sat.” should not be abbreviated, there is enough space for the full term. In the caption, it should read “stress models” or it needs an article in singular form. Fixed.

Table 1: I feel that the description should be added next to the parameter, and the abbreviation (par.) should be avoided. Fixed.

L94: “water surface wave” should be replaced by “tsunami”. Fixed.

Fig. 3: The font sizes of the text within this figure is too large compared to the text on the axes.

All text sizes in figures have been set to pt.7 following the artwork guidelines in the new version.

Fig. 4: The parameters x and z should be written in italic. On the y -axes, there should be free spaces between the unit and the numbers, and the subscripts should be written lower. Fixed.

L105: Please replace “like” with “such as” (a more formal term). Fixed.

L132/L219/L231: I like that a quantification of the error between measurements and simulations has been introduced. The error in [m] or [mm] is useful for a comparison with other configurations within this study. It would be good to express the error also e.g. as a % of the maximum wave amplitude, rather than in mm, as this would be a more general quantification.

This is, in theory simple to do, however, we have to choose a reference length for each of the errors. It is not obvious which reference length to choose, especially when speaking of average errors along the coastline or over time t (for example, the error in the inundation can be related to the mean inundation height, the maximum inundation height or the inundation height at a specific place).

Therefore we added the relative error where it was straightforward (error in wave crest and through, maximum error in inundation) and kept the absolute values elsewhere. We hope that you can understand this decision.

Fig. 6: In the caption, please replace “the figure before” with “Figure 5”. Fixed.

L205: The statement that the deposition is unlikely to be used as a reliable indicator of the tsunamigenic potential opens up some questions and should be weakened in my view, e.g. “deposition alone...”. Some Readers may criticize that, of course, also the bathymetry needs to be considered in relation to the slide run-out distance. Fixed.

L210: Please write “the slide” rather than “slide” only. Fixed.

L218: There is an inconsistency in the manuscript about the “Oxford comma”, here

no comma is used, elsewhere a comma is used after “and”.

We try to use the Oxford comma where appropriate. We added it here.

L227: Please replace “big” with “significant” to use a more scientific term. Fixed.

L231: I feel that the statement of “our model performs well” does not match the maximum error of 32.24 m, i.e. it over-sells the results.

We think that our model performs well, at least in comparison to the current state of the art. However, we agree that the results are not perfect, so we have toned down the sentence and replaced it with "Our model performs overall quite well [...] with some local outliers".

L310/1: Please replace “at” with “from” (twice). Fixed.

L318: “We thank... the support of the editor” is Grammatically incorrect, it is not possible to thank the support. Removed.

L332ff: I believe x should be written in bold to indicate that this is a vector, also in the supplementary material. The x is written in bold. It was difficult to see with the chosen font.

q. (4)ff: Please double-check that all parameters used in these equations are defined in the text. Done.

L356: “low” may be replaced by “small”. Fixed.

L399: Please replace “sand” with “granular”. Fixed.

L413: Please write “Herein” rather than “In here”. Fixed.

Refs. [4], [5], [13], [39], [44], [53] and [58]: Please write the names with capital letters, also the journal names (e.g. “Reports”). Fixed.

Ref. [64]: Please do not use abbreviations. Fixed.

Movie Supplement:

On the 2nd last line it should read “black line” rather than “back line”. Fixed.

Supplementary material:

Fig. 3: “Outline” should be written with a lower case letter. In the caption write “and its surrounding terrain” rather than “the terrain surrounding it”. Fixed.

Caption Fig. 8: Please write “Lake Askja” rather than only “Askja” and drop one of the “in the” on the 2nd last line. Fixed.

Ahead of Eq. (38): Please add a comma after “First”. Fixed.

Table 1: Please replace “max.” with “maximum”. Fixed.

Table 2: In the caption, please write as a superscript. Fixed.

Table 3: In the caption please write “shallow” rather than “Shallow”, and there is a typo in “Boussinsq”, both in the caption and in the table. Fixed.

P12, L3 from the bottom: There is a free space missing ahead of [5]. Fixed.

Before Table 5: The statement “although the quantitative results should be taken with a grain of sand.” remains unclear, please rephrase. Fixed.

Fig. 11: “Measurement” is written with an upper case letter, in contrast to other figures in the manuscript where lower case letters are used. See also Fig. 37.

We use lower case names in labels in supplementary materials because the majority of the figures were formatted this way.

In the captions of Tabs. 6 to 13 the unit (m) for the slope angle is incorrect. Fixed.

Section 5.8: On line 5 please drop the 2nd “on”. Fixed.

After Eq. (47): Please revise the sentence “The macroscopic wave is stronger damped and after the first wave much smoother at the first gauge than the simulation without the turbulence model and the experiment.” And 4 lines further down please drop “on”. Fixed.

Fig. 31: In the legend the text should not be written in italic. Please also use lower case letters for “Equations”. I do not think the abbreviations “NS” and “RANS” have been introduced, so please either introduce them or use the full terms. Fixed.

Fig. 32: The same as in the previous comment applies to the headings of this figure.

We fixed the font where it was possible with acceptable effort. However, since the plotting library and LaTeX differ in some details, not all details could be fixed.

Fig. 36: In this caption, please write the text not in italic and add free spaces. Fixed.

Sec. 7: The first sentence can be linked to the 2nd paragraph to avoid a paragraph consisting of one sentence only. Fixed.

Ref. [3]: Please do not use abbreviations. Fixed.